**Article** https://doi.org/10.1038/s41467-024-48620-7

# The chromatin landscape of pathogenic transcriptional cell states in rheumatoid arthritis

Kathryn Weinand [1,2,3,4,5], Saori Sakaue [1,2,3,4,5], Aparna Nathan [1,2,3,4,5], Anna Helena Jonsson [1], Fan Zhang [1,2,3,4,5,6], Gerald F. M. Watts[1], Majd Al Suqri [1,2,3,5], Zhu Zhu[1], Accelerating Medicines Partnership Program: Rheumatoid Arthritis and Systemic Lupus Erythematosus (AMP RA/SLE) Network*, Deepak A. Rao [1], Jennifer H. Anolik [7], Michael B. Brenner [1], Laura T. Donlin [8,9], Kevin Wei[1] & Soumya Raychaudhuri [1,2,3,4,5,10] ✉

Synovial tissue inflammation is a hallmark of rheumatoid arthritis (RA). Recent work has identified prominent pathogenic cell states in inflamed RA synovial tissue, such as T peripheral helper cells; however, the epigenetic regulation of these states has yet to be defined. Here, we examine genome-wide open chromatin at single-cell resolution in 30 synovial tissue samples, including 12 samples with transcriptional data in multimodal experiments. We identify 24 chromatin classes and predict their associated transcription factors, including a *CD8 + GZMK+* class associated with EOMES and a lining fibroblast class associated with AP-1. By integrating with an RA tissue transcriptional atlas, we propose that these chromatin classes represent 'superstates' corresponding to multiple transcriptional cell states. Finally, we demonstrate the utility of this RA tissue chromatin atlas through the associations between disease phenotypes and chromatin class abundance, as well as the nomination of classes mediating the effects of putatively causal RA genetic variants.

Rheumatoid arthritis (RA) is a chronic autoimmune disease that affects ~1% of people in North America and Northern Europe[1]. In RA, the synovial joint tissue is infiltrated by immune cells that interact with stromal cells to sustain a cycle of inflammation. Untreated, RA can lead to joint destruction, disability, and a reduction in life expectancy[2]. The heterogeneous clinical features of RA, including differences in cyclic citrullinated peptide antibody autoreactivity[3], underlying genetics[4,5], and response to targeted therapies[6–10], render

it challenging to construct generic treatment plans that will be effective for most patients.

Recent studies have taken advantage of single-cell technologies to define key cell populations that are present and expanded in RA tissue inflammation[11–14], demonstrating both the heterogeneous nature of tissue inflammation and the promise to identify novel targeted therapeutics for RA. Our recent Accelerating Medicines Partnership Program: Rheumatoid Arthritis (AMP-RA) reference study[14]

[1]Division of Rheumatology, Inflammation, and Immunity, Department of Medicine, Brigham and Women's Hospital and Harvard Medical School, Boston, MA, USA. [2]Center for Data Sciences, Brigham and Women's Hospital and Harvard Medical School, Boston, MA, USA. [3]Division of Genetics, Department of Medicine, Brigham and Women's Hospital and Harvard Medical School, Boston, MA, USA. [4]Department of Biomedical Informatics, Harvard Medical School, Boston, MA, USA. [5]Broad Institute of MIT and Harvard, Cambridge, MA, USA. [6]Department of Medicine Division of Rheumatology and Department of Biomedical Informatics, University of Colorado School of Medicine, Aurora, CO, USA. [7]Division of Allergy, Immunology and Rheumatology, Department of Medicine, University of Rochester Medical Center, Rochester, NY, USA. [8]Hospital for Special Surgery, New York, NY, USA. [9]Weill Cornell Medicine, New York, NY, USA. [10]Versus Arthritis Centre for Genetics and Genomics, Centre for Musculoskeletal Research, Manchester Academic Health Science Centre, The University of Manchester, Manchester, UK. *A list of authors and their affiliations appears at the end of the paper. ✉e-mail: soumya@broadinstitute.org

comprehensively classified pathogenic transcriptional cell states within synovial joint tissue using single-cell CITE-seq[15], which simultaneously measures mRNA and surface protein marker expression at the single-cell level. Within 6 broad cell types (B/plasma, T, NK, myeloid, stromal [fibroblast/mural], and endothelial), the study defined 77 fine-grain cell states. Many of these cell states have been previously shown to be associated with RA pathology: for example, CD4+ T peripheral helper cells (TPH)[11,12], HLA-DR[hi] sublining fibroblasts[11], proinflammatory *IL1B*+ monocytes[11], and autoimmune-associated B cells (ABC)[11,16]. However, we have a limited understanding of the chromatin accessibility profiles that underlie these pathogenic synovial tissue cell states.

Open chromatin at critical *cis*-regulatory regions allows essential transcription factors (TFs) to access DNA and epigenetically regulate gene expression[17]. Chromatin accessibility is a necessary, but not sufficient, condition for RNA polymerases to produce transcripts at gene promoters[18]. Therefore, one possibility is that each transcriptional cell state has its own unique chromatin profile[19], which we will denote as a chromatin class. Alternatively, multiple transcriptional cell states could share a chromatin class if the cell states were dynamically transitioning from one to another in response to external stimuli without altering the chromatin landscape[19]. In RA, those external stimuli could

be cytokines that activate TFs to induce the expression of key genes and drive pathogenic cell states[20]. For example, NOTCH3 signaling propels transcriptional programs coordinating the transformation from perivascular fibroblasts to inflammatory sublining fibroblasts[21]. Similarly, exposure to TNF and interferon-γ promotes the differentiation of monocytes into inflammatory myeloid cells[22].

Here, we characterize synovial cells from patients with RA or osteoarthritis (OA) using unimodal single-cell ATAC-seq (scATAC-seq) and multimodal single-nucleus ATAC-seq (snATAC-seq) and RNA-seq (snRNA-seq) technologies to compare chromatin classes to transcriptional cell states (Fig. 1a). Our results support a model of open chromatin superstates shared by multiple fine-grain transcriptional cell states. We show these superstates may be regulated by key TFs and associated with clinical and genetic factors in the pathology of RA (Fig. 1a).

## Results

### Quality control of unimodal scATAC-seq and multimodal snATAC-seq synovial tissue datasets

We obtained synovial biopsy specimens from 25 people with RA and 5 with OA and disaggregated cells using well-established protocols from the AMP-RA/SLE consortium[23] (Methods). We conducted unimodal

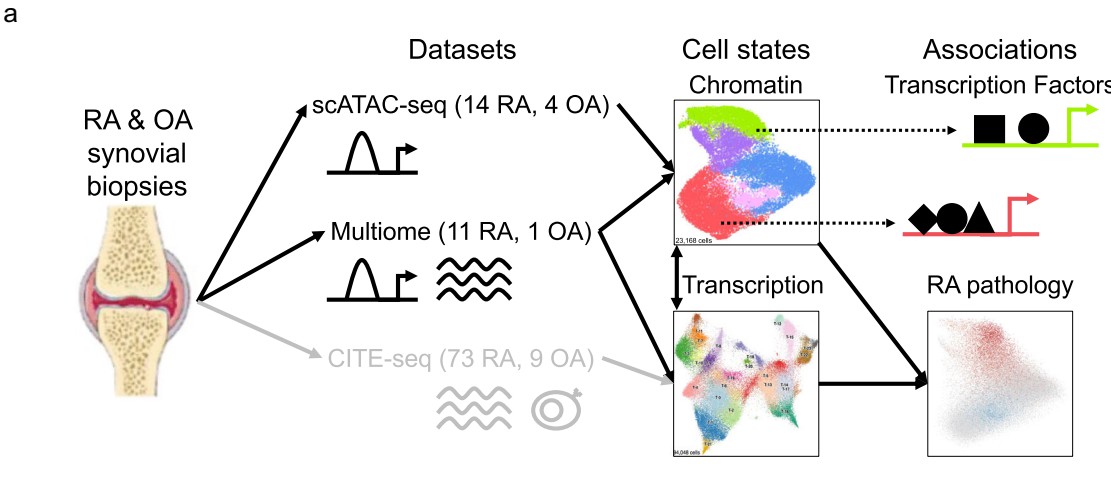

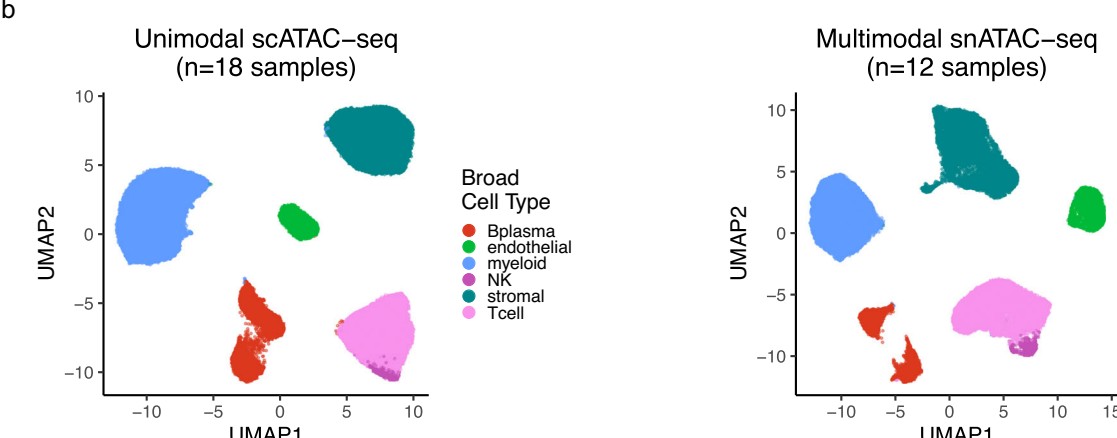

**Fig. 1 | Study overview and open chromatin broad cell type identification. a** Study overview. Synovial biopsy specimens from RA and OA patients were utilized for unimodal scATAC-seq and multimodal snATAC-seq + snRNA-seq experiments. CITE-seq on similar specimens was performed in the AMP-RA reference study[14]. We defined chromatin classes using the unimodal scATAC-seq and multimodal snATAC-seq data and compared them with AMP-RA transcriptional cell states[14] classified onto the multiome cells. We further defined transcription factors likely regulating these chromatin classes and found putative links to RA pathology by associating the classes to RA clinical metrics, RA subtypes, and putative RA risk variants. **b** Open chromatin broad cell type identification in unimodal scATAC-seq datasets (left) and multimodal snATAC-seq datasets (right) visualized on a UMAP, processed separately. Parts of Fig. 1a were generated using Servier Medical Art, provided by Servier, licensed under a Creative Commons Attribution 3.0 unported license.

scATAC-seq on samples from 14 RA patients and 4 OA patients and multimodal snATAC-/snRNA-seq on samples from 11 RA patients and 1 OA patient (Supplementary Table 1). Applying stringent quality control to the open chromatin modality, we retained cells with >10,000 reads, >50% of those reads falling in peak neighborhoods, >10% of reads in promoter regions, <10% of reads in the mitochondrial chromosome, and <10% of reads falling in the ENCODE blacklisted regions[24] (Methods; Supplementary Figs. 1a, b and 2a, b; Supplementary Table 2). We further required that cells from the multimodal data pass quality control for the snRNA-seq modality (Methods; Supplementary Figs. 1b and 2c). After additional QC within individual cell types combining both technologies, the final dataset contained 86,994 cells from 30 samples (median of 2990 cells/sample) (Supplementary Figs. 1c, d and 2d, e). For consistency, we called a set of 132,520 consensus peaks from the unimodal scATAC-seq data to be used for all analyses (Methods). We observed that 95% of the called peaks overlapped ENCODE candidate *cis*-regulatory elements (cCREs)[25] and 17% overlapped promoters[26], suggesting highly accurate peak calls (Supplementary Fig. 2f).

## Defining RA broad cell types by clustering unimodal and multimodal datasets

To assign each cell to a broad cell type, we clustered the unimodal scATAC-seq and multimodal snATAC-seq datasets independently (Methods). In both instances, we characterized six cell types that we annotated based on the chromatin accessibility of "marker peaks," defined as peaks in cell-type-specific marker gene promoters (Methods; Fig. 1b). We identified T cells (*CD3D* and *CD3G*), NK cells (*NCAM1* and *NCR1*), B/plasma cells (*MS4A1* and *TNFRSF17*), myeloid cells (*CD163* and *C1QA*), stromal cells (*PDPN* and *PDGFRB*), and vascular endothelial cells (*VWF* and *ERG*) (Supplementary Fig. 2g–j). In the multimodal data, we observed consistent peak accessibility and gene expression for marker genes in these cell types (Supplementary Fig. 2k–m).

We combined cells from unimodal and multimodal chromatin technologies and then created datasets for each of the broad cell types. For cell types with more than 1500 cells, we applied Louvain clustering to a shared nearest neighbor graph based on batch corrected[27] principal components (PCs) of chromatin accessibility to define fine-grain chromatin classes (Methods).

## RA T cell chromatin classes

We first examined the accessible chromatin for 23,168 T cells across unimodal and multimodal datasets. Louvain clustering defined 5 T cell chromatin classes, denoted as $T_A$ for T cell ATAC, across 30 samples (Fig. 2a; Supplementary Fig. 3a). In the $T_A$−2: CD4+ PD-1+ TFH/TPH chromatin class, we observed high promoter accessibility and gene expression for PD-1 (*PDCD1*) and *CTLA4*, marker genes for T follicular helper (TFH)/TPH cells (Fig. 2b; Supplementary Fig. 3b). A known expanded pathogenic cell state in RA, TFH/TPH cells help B cells respond to inflammation[11,12]. The $T_A$−3: CD4+ IKZF2+ Treg cluster had high accessibility and expression for *IKZF2* (Helios), which can stabilize the inhibitory activity of regulatory T cells[28] (Treg) (Fig. 2b). We also observed open chromatin regions at both the *FOXP3* transcription start site (TSS) as well as the downstream Treg-specific demethylated region[29] (TSDR) specifically for $T_A$−3 (Supplementary Fig. 3c); *FOXP3* was also expressed exclusively in $T_A$−3 cells (Supplementary Fig. 3b). We found one more predominantly CD4+ T cell class, $T_A$−1: CD4+ IL7R+, with high expression and accessibility for *IL7R*, encoding the CD127 protein. This marker is typically lost with activation, suggesting that $T_A$−1 is a population of naive or central memory T cells, as further evidenced by *SELL* and *CCR7* expression (Fig. 2b; Supplementary Fig. 3b). The $T_A$−0: CD8A+ GZMK+ cluster was marked by *GZMK* and *CRTAM* peak accessibility and gene expression (Fig. 2b; Supplementary Fig. 3b); a similar population has been shown to be expanded in RA and a major producer of inflammatory cytokines[11,30]. We found another

primarily CD8+ group of T cells, the $T_A$−4: CD8A+ PRF1+ cytotoxic cluster, which had high accessibility for the *PRF1* promoter and expression for the *PRF1*, *GNLY*, and *GZMB* genes, suggesting an effector memory phenotype (Fig. 2b; Supplementary Fig. 3b).

Since T cells are primarily defined as CD4 and CD8 lineages that are not thought to cross-differentiate[31], we next examined whether the chromatin classes were strictly segregated by CD4 or CD8 promoter peak accessibility. We observed that each chromatin class, while largely showing accessibility for only one lineage's promoter, also included some cells with accessibility for the other lineage's promoter (Supplementary Table 3). For example, cytotoxic T cells in $T_A$−4 were more likely to have an accessible CD8A promoter, but also included a minority of cells with accessibility at the CD4 promoter. Therefore, we assessed which promoter peaks were associated with a specific lineage. While accounting for chromatin class, sample, and fragment count, we ran a logistic regression model over all T cells relating each promoter peak's openness to CD4/CD8A promoter peak accessibility status: 1 for open CD4 and closed CD8A, −1 for open CD8A and closed CD4, or 0 otherwise (Methods). We only found 93 out of 16,383 promoter peaks open in T cells significantly associated with a lineage's promoter accessibility, with 29 associating to CD4 and 64 to CD8A, at FDR < 0.20 (Supplementary Data 1). This indicated that T cell lineage is important for a small subset of genes' local promoter chromatin environment, such as *IL6ST* in CD4 T cells and *CRTAM* in CD8 T cells, and those lineage-specific loci segregate by chromatin class as expected (Methods; Supplementary Fig. 3d). However, the majority of promoters appeared to be more specifically accessible within their chromatin classes across lineages. This might suggest that the corresponding gene's function was critical for the class definition, as highlighted by functional genes such as *PRF1* with expression in both cytotoxic CD4 and CD8 T cells[32] as well as the homing gene *CCR7* that acts across both lineages[33].

We next identified the TFs potentially regulating these T cell chromatin classes by calculating TF motif enrichments[34] in class-specific peaks[35] whose TFs were at least minimally expressed within that class (Methods). In the primarily CD8+ classes, $T_A$−0: CD8A + GZMK+ and $T_A$−4: CD8A+ PRF1+ cytotoxic, we found EOMES ($p_{adj}$ = 7.44e-99, 8.12e-44, respectively) and T-bet (*TBX21*) ($p_{adj}$ = 4.92e-90, 2.75e-38, respectively) motifs enriched (Fig. 2c); the corresponding TFs are known to drive memory and effector CD8+ cell states[36]. *EOMES* had significantly higher gene expression in $T_A$−0 cells compared to all other T cells (Wilcoxon FDR = 1.92e-84; Supplementary Data 2). Furthermore, we found both motifs in the promoter of *KLRG1*, a gene expressed in CD8+ effector T cells that might participate in the effector-to-memory transition[37] (Fig. 2d). The cytotoxic $T_A$−4 class was also enriched for RUNX3[38] motifs ($p_{adj}$ = 5.81e-13) (Fig. 2c). Within the $T_A$−2: CD4+ PD-1+ TFH/TPH class, we observed high enrichments for AP-1 motifs, especially BATF ($p_{adj}$ = 3.31e-103; Fig. 2d), which promotes expression of key programs in TFH cells[39] and had higher gene expression in this class's cells (Wilcoxon FDR = 3.10e-125; Supplementary Data 2). We found TCF7 and LEF1 motifs[40] within the non-activated $T_A$−1: CD4+ IL7R+ cluster ($p_{adj}$ = 1.14e-10, 3.97e-13, respectively; Fig. 2d).

## RA stromal chromatin classes

Next, we analyzed 24,307 stromal cells (Methods). With Louvain clustering, we partitioned the cells into 4 open chromatin classes: lining fibroblasts ($S_A$−1) along the synovial membrane, sublining fibroblasts ($S_A$−0, $S_A$−2) filling the interstitial space, and mural cells ($S_A$−3) adjacent to blood vessels[41] (Fig. 3a; Supplementary Fig. 4a). The most abundant sublining cluster, $S_A$−0: CXCL12+ HLA-DR$^{hi}$ sublining fibroblasts, was a proinflammatory cluster marked by *CXCL12*, *HLA-DRA*, and *CD74* accessibility and expression; $S_A$−0 also expressed *IL6*, which is an established RA drug target[7,8] (Fig. 3b; Supplementary Fig. 4b). The $S_A$−2: CD34+ MFAP5+ sublining fibroblast class had accessible promoter peaks, where available, for the expressed *CD34*, *MFAP5*, *PI16*, and

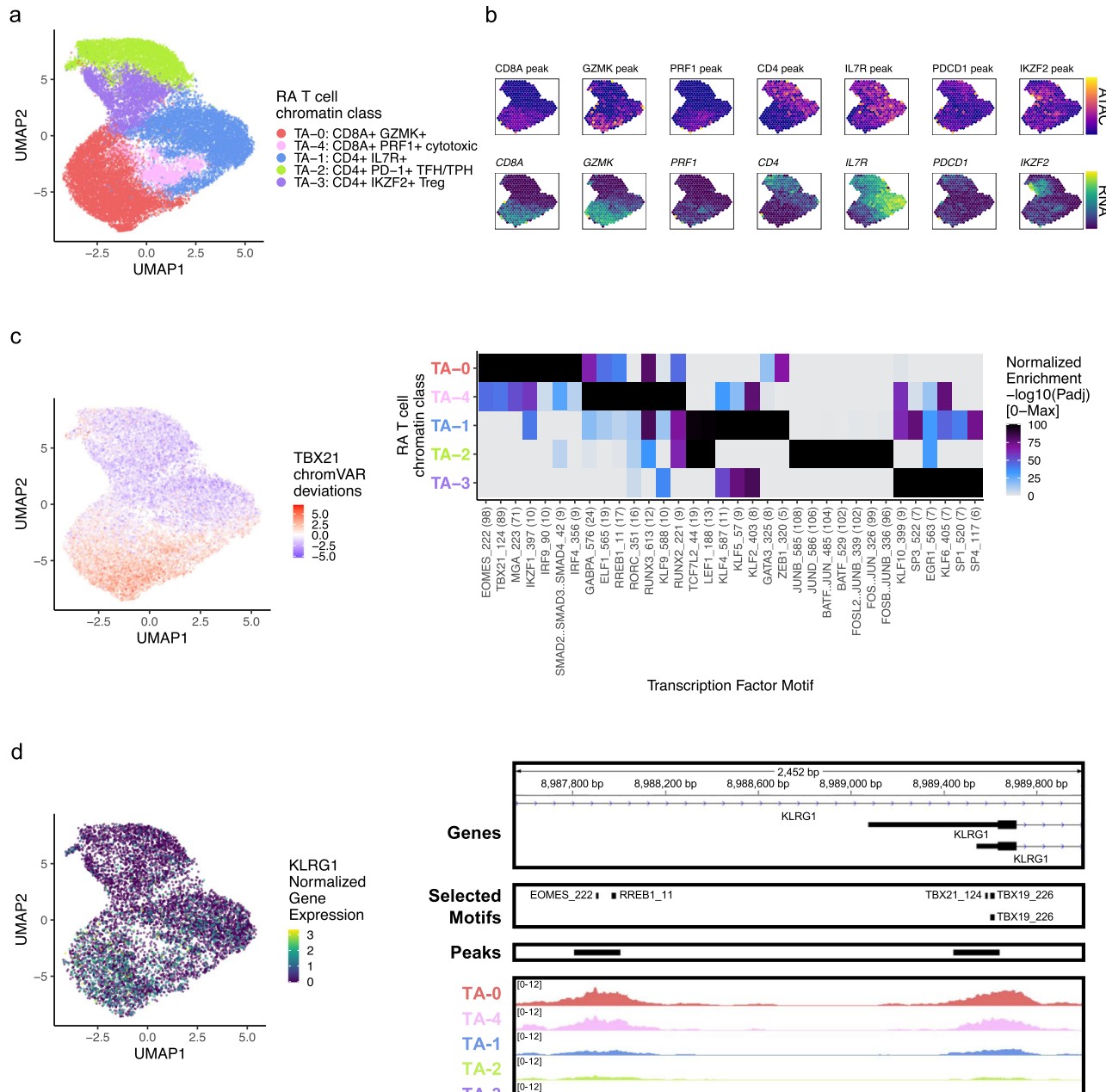

**Fig. 2 | RA T cell chromatin classes. a** UMAP colored by 5 T cell chromatin classes defined from unimodal scATAC-seq and multimodal snATAC-seq cells. **b** Mean binned normalized marker peak accessibility (top; yellow (high) to purple (low)) and gene expression (bottom; yellow (high) to blue (low)) for multimodal snATAC-seq cells on UMAP. **c** UMAP colored by chromVAR[34] deviations for the TBX21 motif (left). Most significantly enriched motifs in class-specific peaks per T cell chromatin class (right). To be included per class, motifs had to be enriched in the class above a minimal threshold, and corresponding TFs had to have at least minimal expression in snRNA-seq. Color scale normalized per motif across classes with max −log10($p_{adj}$) value shown in parentheses in motif label. *P* values were calculated via hypergeometric test in ArchR[35]. **d** UMAP colored by *KLRG1* normalized gene expression in multiome cells (left). KLRG1 locus (chr12:8,987,550–8,990,000) with selected gene isoforms, motifs, open chromatin peaks, and chromatin accessibility reads from unimodal and multiome cells aggregated by chromatin class and scaled by read counts per class (right).

*DPP4* genes, previously reported to represent a progenitor-like fibroblast state shared across tissue types[42–44] (Fig. 3b; Supplementary Fig. 4b). The S$_A$−1: PRG4+ lining fibroblast chromatin class was characterized with high accessibility and expression of *PRG4* and *CRTAC1* (Fig. 3b; Supplementary Fig. 4b). We also observed high expression of *MMP1* and *MMP3*, matrix metalloproteinases responsible for extracellular matrix (ECM) destruction[45], within S$_A$−1 (Supplementary Fig. 4b). Finally, we found a mural cell class, S$_A$−3: MCAM+ mural, with both gene expression and promoter peak accessibility for *MCAM* and *NOTCH3* (Fig. 3b; Supplementary Fig. 4b). In RA, NOTCH3 signaling

from the endothelium acts primarily on mural cells, which in turn stimulate sublining fibroblasts along a spatial axis[21] as seen in the decreasing NOTCH3 gene expression from S$_A$−3, S$_A$−0, S$_A$−2, to S$_A$−1 in the multiome cells (Supplementary Fig. 4b). Knockout of *NOTCH3* has been shown to reduce inflammation and joint destruction in mouse models[21].

DNA methylation and chromatin accessibility work in tandem to define cell-type-specific gene regulation through silencing CpG-dense promoters and repressing methylation-sensitive TF binding[46]. Methylation changes have been previously described between cultured

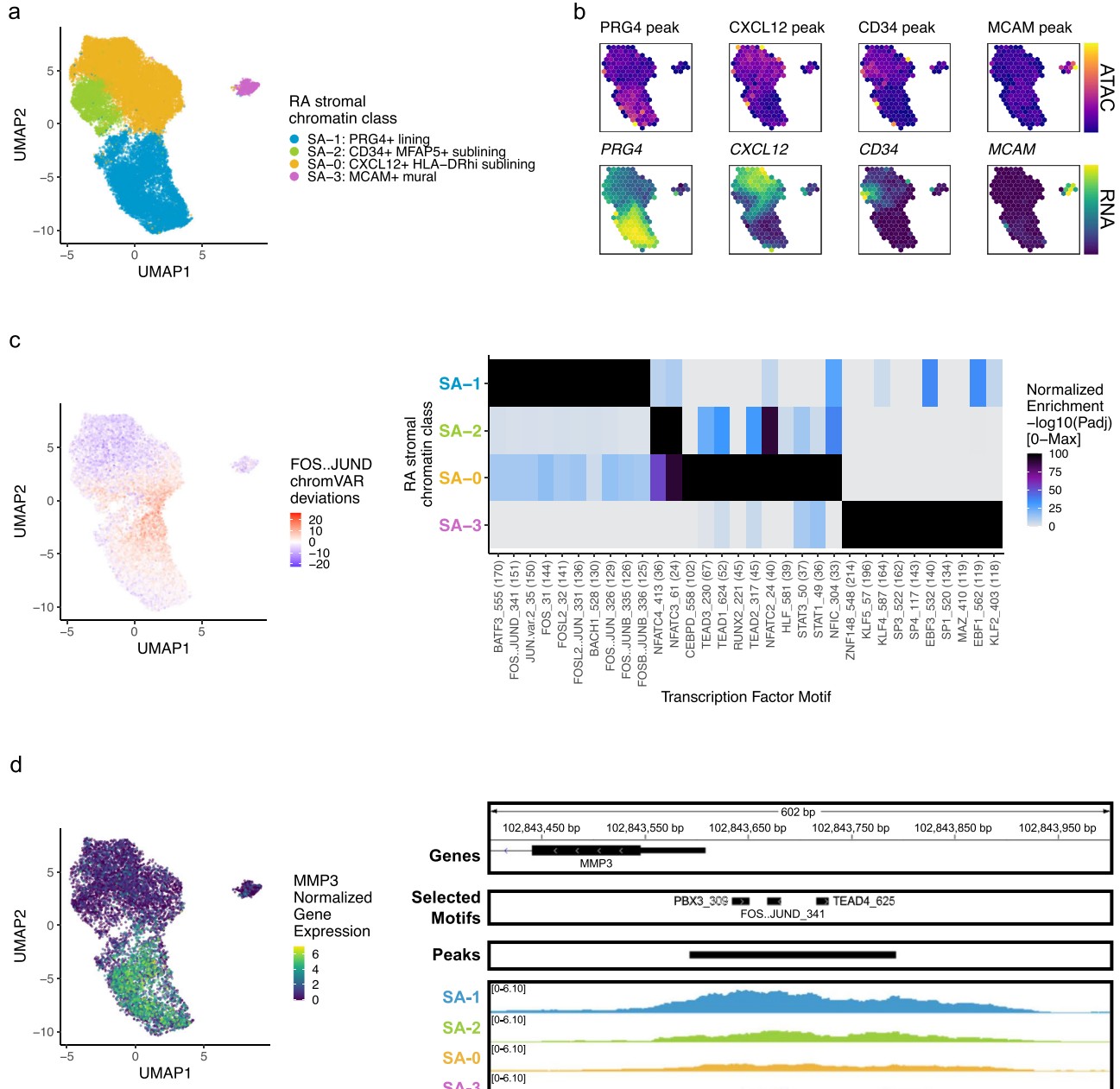

**Fig. 3 | RA stromal chromatin classes. a** UMAP colored by 4 stromal chromatin classes defined from unimodal scATAC-seq and multimodal snATAC-seq cells. **b** Mean binned normalized marker peak accessibility (top; yellow (high) to purple (low)) and gene expression (bottom; yellow (high) to blue (low)) for multimodal snATAC-seq cells on UMAP. **c** UMAP colored by chromVAR[34] deviations for the FOS::JUND motif (left). Most significantly enriched motifs in class-specific peaks per stromal chromatin class (right). To be included per class, motifs had to be enriched in the class above a minimal threshold, and corresponding TFs had to have at least minimal expression in snRNA-seq. Color scale normalized per motif across classes with max $-\log10(p_{adj})$ value shown in parentheses in motif label. *P* values were calculated via hypergeometric test in ArchR[35]. **d** UMAP colored by *MMP3* normalized gene expression (left). MMP3 locus (chr11:102,843,400–102,844,000) with selected gene isoforms, motifs, open chromatin peaks, and chromatin accessibility reads from unimodal and multiome cells aggregated by chromatin class and scaled by read counts per class (right).

fibroblast cell lines from RA and OA patients[47,48]. Thus, we wondered if a specific subset of fibroblasts might be the source of these differentially methylated regions (DMRs). Using a published set of DMRs for RA versus OA fibroblast-like synoviocyte (FLS) cell lines[47], we defined a per-cell score of peak accessibility associated with hypermethylated (positive) or hypomethylated (negative) loci in RA (Methods). The sublining fibroblasts in $S_A$−0 were enriched for hypomethylated regions (Wilcoxon $S_A$−0 versus other stromal cells one-sided $p < 2.2e$-16), suggesting that the RA synovial fibroblast DMRs were relatively enriched for putatively functional accessible chromatin regions

specifically in sublining fibroblasts (Supplementary Fig. 4c). Furthermore, the genes associated to these FLS DMRs were expressed primarily in tissue $S_A$−0 (Supplementary Fig. 4d, right; Methods) and are crucial to a number of signaling pathways potentially at play in these inflammatory fibroblasts[47]: *STAT3* in IL-6 signaling, *CASP1* in IL-1 signaling, *TRAF2* in TNF signaling, and *TGFB3* in TGFβ signaling. These results proposed the possibility of epigenetic memory retention even after multiple FLS cell line passages[49], as sublining fibroblasts, particularly HLA-DR[hi] and CD34[−] fibroblasts, are expanded in RA relative to OA in synovial tissue samples[11].

We then considered if the retention of DNA methylation after multiple passages extended to a retention of chromatin accessibility or whether that would be lost alongside transcriptional identity[21]. To assess this, we developed two per-cell scores of fibroblast identity comparing tissue lining ($S_A$–1) to sublining ($S_A$–0, $S_A$–2) cells; one score using differentially expressed genes and the other using differentially accessible peaks. Using a multiome dataset of isolated FLS from two RA synovial tissue samples cultured for three passages in a recent RA fibroblast heterogeneity study[44], we compared their per-cell fibroblast identity score to our tissue fibroblast populations in both gene and peak space. Unsurprisingly, we found that differential genes from tissue were able to separate tissue lining and tissue sublining cells, but the cultured FLS did not have discernable lining and sublining populations by the same measure, consistent with previous results[21] (Supplementary Fig. 4e). More surprisingly, we saw similar results using the fibroblast identity peak score (Supplementary Fig. 4f), suggesting that fibroblast peak accessibility, and more broadly chromatin class identity, was not maintained in cell culture after multiple passages. This disconnect between DNA methylation and chromatin accessibility has also been seen previously when assaying both directly using ATAC-Me in the monocyte-to-macrophage cell fate transition[50].

Next, we investigated which TFs were putatively driving these chromatin classes (Fig. 3c). AP-1 motifs such as FOS::JUND were most significantly enriched in the $S_A$–1 lining class ($p_{adj}$ = 9.29e-152; Fig. 3c). These TFs are known to play many roles in RA and specifically regulate *MMP1* and *MMP3* promoters[49,51] (Fig. 3d). The progenitor-like sublining $S_A$–2 class harbored NFATC motifs, such as NFATC4 ($p_{adj}$ = 2.89e-36; Fig. 3c). In the $S_A$–0: CXCL12+ HLA-DR[hi] sublining chromatin class, we found TEAD1[52] ($p_{adj}$ = 2.86e-52; Fig. 3c) and STAT1/3 TF motif enrichments ($p_{adj}$ = 3.34e-37, 4.27e-38, respectively; Fig. 3c), with the latter likely regulating the JAK/STAT pathway responsible for the proinflammatory cytokine activation central to RA clinical activity[9,53]. The gene expression of *TEAD1* and *STAT3* in $S_A$–0 cells was significantly higher than in the other stromal cells (Wilcoxon FDR = 1.05e-27 and 1.65e-17, respectively; Supplementary Data 2). Finally, $S_A$–3: MCAM+ mural cells were enriched for KLF2[54,55] and EBF1[56,57] motifs ($p_{adj}$ = 4.94e-119, 1.83e-119, respectively; Fig. 3c).

## RA myeloid chromatin classes

We classified 25,691 myeloid cells into 5 chromatin classes (Fig. 4a; Supplementary Fig. 5a). The first class, $M_A$–2: LYVE1+ TIMD4+ TRM, had markers for tissue-resident macrophages (TRM) with gene and peak signal at *LYVE1*, a perivascular localization marker[13], and *TIMD4*, a scavenger receptor[13] (Fig. 4b; Supplementary Fig. 5b). We found another TRM class, $M_A$–0: F13A1+ MARCKS+ TRM, with high accessibility and expression at *F13A1* and *MARCKS*, both known to be expressed in macrophages[58,59] (Fig. 4b; Supplementary Fig. 5b). The $M_A$–1: FCN1+ SAMSN1+ infiltrating monocytes had accessibility and expression for *FCN1*, *PLAUR*, *CCR2*, and *IL1B*, similar to an expanded proinflammatory population in a previous RA study[11] (Fig. 4b; Supplementary Fig. 5b). The $M_A$–4: SPP1+ FABP5+ intermediate class likely arose from bone marrow-derived macrophages[60] with its high accessibility and expression for *SPP1* (Fig. 4b); bone marrow-derived macrophages are known be abundant in active RA and induce proinflammatory cytokines/chemokines[13,61]. Finally, we found the $M_A$–3: CD1C+ AFF3+ DC chromatin class with expression markers *CD1C*, *AFF3*, *CLEC10A*, and *FCER1A*, whose corresponding promoter peaks generally showed more promiscuously open chromatin across classes (Fig. 4b; Supplementary Fig. 5b).

We next investigated the TF motifs enriched in the myeloid chromatin classes. $M_A$–2 was enriched for KLF motifs (Fig. 4c), with *KLF4* ($p_{adj}$ = 1.34e-6) known to both establish residency of TRMs and to assist in their phagocytic function[62]. Furthermore, we found a KLF4 motif in the promoter of *C1QB*, whose protein product bridges phagocytes to the apoptotic cells they clear[63] (Fig. 4d). Both the intermediate $M_A$–4 and the infiltrating monocyte $M_A$–1 classes had significant enrichments of AP-1 activation motifs (e.g., JUN $p_{adj}$ = 1.77e-153, 3.65e-136, respectively; Fig. 4c). AP-1 TFs have been shown to function in human classical monocytes along with CEBP TFs[64], also enriched in $M_A$–1 (e.g., CEBPD $p_{adj}$ = 2.10e-26; Fig. 4c). SPI1 (PU.1) is the master regulator of myeloid development[65], including conventional DCs[66]. We found the SPI1 motif most strongly enriched in the DC cluster $M_A$–3 ($p_{adj}$ = 3.24e-55; Fig. 4c), though the related SPIB motif's corresponding TF, known to function in pDCs[67], was more specifically expressed in this class (Wilcoxon FDR = 6.93e-74; Supplementary Data 2).

## RA B/plasma chromatin classes

Next, we clustered 8641 B and plasma cells into 4 MS4A1+ B cell and 2 SDC1+ plasma cell chromatin classes (Methods; Fig. 5a; Supplementary Fig. 6a). We defined a $B_A$–3: FCER2+ IGHD+ naive B class with high accessibility and expression of *FCER2*, encoding naïve marker CD23[68] (Fig. 5b; Supplementary Fig. 6b). We also labeled a $B_A$–4: CD24+ MAST4+ unswitched memory B class (Supplementary Fig. 6b). *IGHD* and *IGHM* expression was lower in $B_A$–2: TOX+ PDE4D+ switched memory B cells, and the TF *TOX* had its highest expression and accessibility within B cells in $B_A$–2 as previously shown in switched memory B cells[69,70] (Fig. 5b; Supplementary Fig. 6b). $B_A$–5: ITGAX+ ABC had high accessibility and expression of *ITGAX*, encoding for CD11c, a key ABC marker[71] (Fig. 5b; Supplementary Fig. 6b). ABCs were shown to be associated with leukocyte-rich RA[11] with a potential role in antigen presentation[72], which was supported here by the expression of *LAMP1* and *HLA-DRA* in $B_A$–5 (Supplementary Fig. 6b). The plasma chromatin class, $B_A$–0: CREB3L2+ plasma, was marked by *CREB3L2*, a known TF in the transition between B and plasma cells[73] (Fig. 5b; Supplementary Fig. 6b). These results suggested tissue in situ B cell activation and differentiation into plasma cells, as we have previously suggested[74]. Finally, $B_A$–1: CD27+ plasma, had the highest accessibility and expression of *CD27* (Fig. 5b; Supplementary Fig. 6b). We note that plasma cells were difficult to define using chromatin accessibility data, with many of the immunoglobulin genes having low signal (Supplementary Fig. 6b).

We then explored the TF motif landscape of B and plasma cells. B cells shared many TF motifs across clusters, with many ETS factors (e.g., SPIB, SPI1, ETS1) as well as EBF1 and NFkB1/2 (Fig. 5c). SPIB and SPI1 work together to regulate B cell receptor signaling[75], which starts its dysregulation in RA at the naive B cell level[76,77] ($p_{adj}$ = 0, 0, respectively; Fig. 5c). Switched memory B cells were enriched for ETS1 motifs ($p_{adj}$ = 9.51e-19; Fig. 5c), whose TF is required for IgG2a class switching in mice[78]. In plasma cells, $B_A$–0 had over-represented motifs such as KLF2[79] and SP3[80] ($p_{adj}$ = 8.94e-105, 3.84e-138, respectively; Fig. 5c, d). $B_A$–1 was enriched for AP-1 factor motifs[81], namely BATF::JUN ($p_{adj}$ = 0; Fig. 5c, d, Supplementary Fig. 6c). Both *BATF* and *JUN* gene expression was higher in $B_A$–1 cells compared to those in other B/plasma classes (Wilcoxon FDR = 9.29e-04 and 1.60e-47, respectively; Supplementary Data 2). In the locus of *PRDM1*, a known plasma cell TF[80], the more $B_A$–0 accessible peak had an SP3 motif while the more $B_A$–1 accessible peaks had BATF::JUN motifs (Fig. 5d), suggesting potentially different regulatory strategies by class.

## RA endothelial chromatin classes

Among the 3809 endothelial cells, we identified 4 chromatin classes (Fig. 6a; Supplementary Fig. 7a). The $E_A$–2: SEMA3G+ arteriolar class had gene and peak markers for signaling-related genes including *SEMA3G*[82], *CXCL12*, and *JAG1* (Fig. 6b; Supplementary Fig. 7b). The NOTCH3 signaling gradient that causes inflammation and joint destruction in RA mouse models likely originates through Notch ligand JAG1 in these arteriolar endothelial cells[21]. We identified the $E_A$–0: SELP+ venular class with markers for leukocyte trafficking to tissue such as

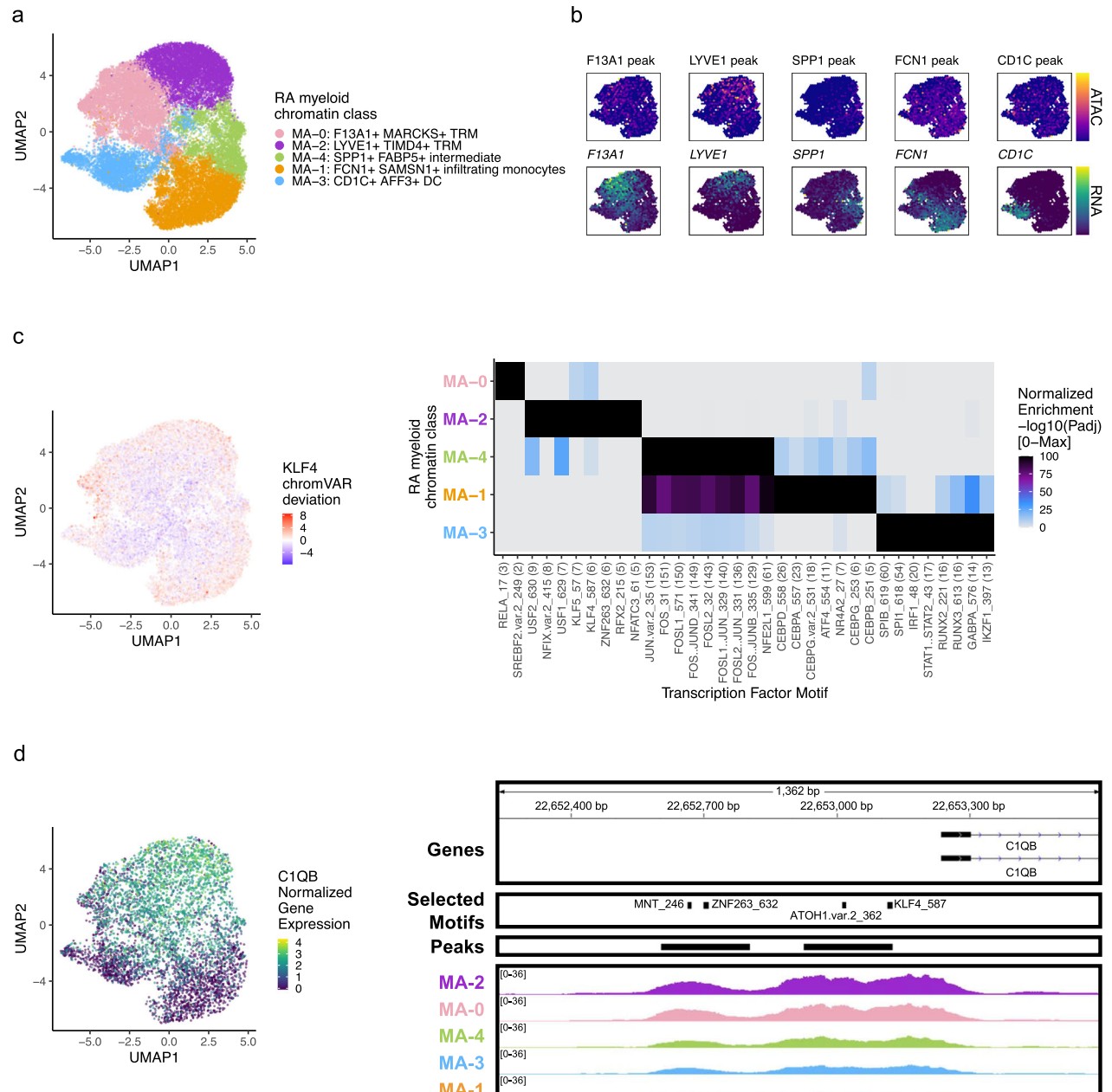

**Fig. 4 | RA myeloid chromatin classes. a** UMAP colored by 5 myeloid chromatin classes defined from unimodal scATAC-seq and multimodal snATAC-seq cells. **b** Mean binned normalized marker peak accessibility (top; yellow (high) to purple (low)) and gene expression (bottom; yellow (high) to blue (low)) for multimodal snATAC-seq cells on UMAP. **c** UMAP colored by chromVAR[34] deviations for the KLF4 motif (left). Most significantly enriched motifs in class-specific peaks per myeloid chromatin class (right). To be included per class, motifs had to be enriched in the class above a minimal threshold, and corresponding TFs had to have at least minimal expression in snRNA-seq. Color scale normalized per motif across classes with max $-\log10(p_{adj})$ value shown in parentheses in motif label. *P* values were calculated via hypergeometric test in ArchR[35]. **d** UMAP colored by *C1QB* normalized gene expression (left). C1QB locus (chr1: 22,652,235–22,653,595) with selected gene isoforms, motifs, open chromatin peaks, and chromatin accessibility reads from unimodal and multiome cells aggregated by chromatin class and scaled by read counts per class (right).

*SELP*[83] as well as inflammatory genes *HLA-DRA* and *CD74* (Fig. 6b; Supplementary Fig. 7b). We also found a capillary class, $E_A$−1: RGCC+ capillary marked by *RGCC*[84] and *SPARC* chromatin accessibility and gene expression (Fig. 6b; Supplementary Fig. 7b). Finally, a small population of $E_A$−3: PROX1+ lymphatic cells had gene expression of and promoter peak accessibility at *PROX1*[85] and *PARD6G* genes (Fig. 6b; Supplementary Fig. 7b).

We identified SOX motifs[86] in $E_A$−2, STAT motifs[87] in $E_A$−0, and AP-1 motifs[88] in $E_A$−1 (Fig. 6c). Sox17 is a crucial intermediary between Wnt and Notch signaling that specifically initiates and maintains endothelial arterial identity in mice[86]. Similarly, we found a SOX17 motif ($p_{adj}$ = 3.27e-8) in the promoter of *NES*[89,90] with its highest accessibility and expression (Wilcoxon FDR = 4.29e-19; Supplementary Data 2) in $E_A$−2 cells (Fig. 6d).

## Chromatin classes are stable irrespective of OA and low-cell-count samples

Our chromatin classes were determined using all samples for maximum power, so we next investigated the contribution of OA and low-cell-count samples to this classification. While we were underpowered

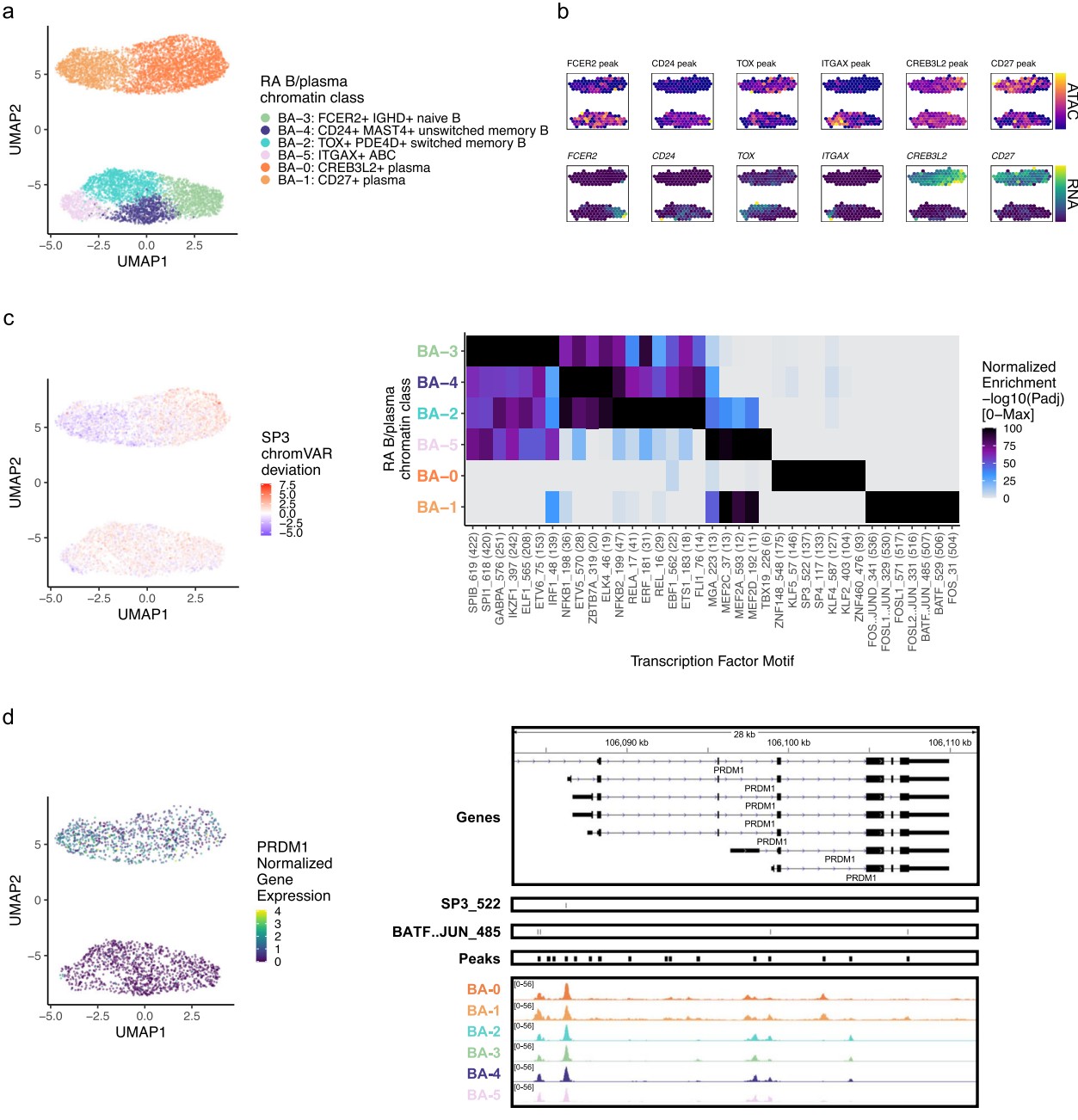

**Fig. 5 | RA B/plasma chromatin classes. a** UMAP colored by 6 B/plasma chromatin classes defined from unimodal scATAC-seq and multimodal snATAC-seq cells. **b** Mean binned normalized marker peak accessibility (top; yellow (high) to purple (low)) and gene expression (bottom; yellow (high) to blue (low)) for multimodal snATAC-seq cells on UMAP. **c** UMAP colored by chromVAR[34] deviations for the SP3 motif (left). Most significantly enriched motifs in class-specific peaks per B/plasma chromatin class (right). To be included per class, motifs had to be enriched in the class above a minimal threshold, and corresponding TFs had to have at least minimal expression in snRNA-seq. Color scale normalized per motif across classes with max $-\log 10(p_{adj})$ value shown in parentheses in motif label. *P* values were calculated via hypergeometric test in ArchR[35]. **d** UMAP colored by *PRDM1* normalized gene expression (left). PRDM1 locus (chr6:106,082,865–106,111,658) with selected gene isoforms, motifs, open chromatin peaks, and chromatin accessibility reads from unimodal and multiome cells aggregated by chromatin class and scaled by read counts per class (right).

to reliably detect differences between RA and OA, we saw that chromatin classes varied in their proportions between these two diseases (Supplementary Table 4). To determine if the chromatin class definitions were robust to the exclusion of OA samples, we removed the 2395 T cells corresponding to OA samples and reclustered the remaining cells. We only observed positive, significant odds ratios (ORs) for cells from a new RA-only cluster belonging to their corresponding original chromatin class, relative to the other classes (Supplementary Fig. 8a). This showed that the same groups of RA T cells cluster together regardless of whether OA T cells were included in the clustering. Since stromal cells had a higher proportion of OA cells, particularly in lining fibroblasts[14,91] (Supplementary Table 4), we also reclustered the stromal cells after removing 4,462 cells from OA samples and found that all four of our original stromal chromatin classes had corresponding RA-only cluster(s) (Supplementary Fig. 8b). Furthermore, we sought to determine if including the low-cell-count samples was impacting the chromatin class definitions, especially for the cell types with lower cell counts overall. To test this, we removed 467 cells across 11 samples with fewer than 100 B/plasma cells and reclustered the remaining cells. We were able to recover all the original

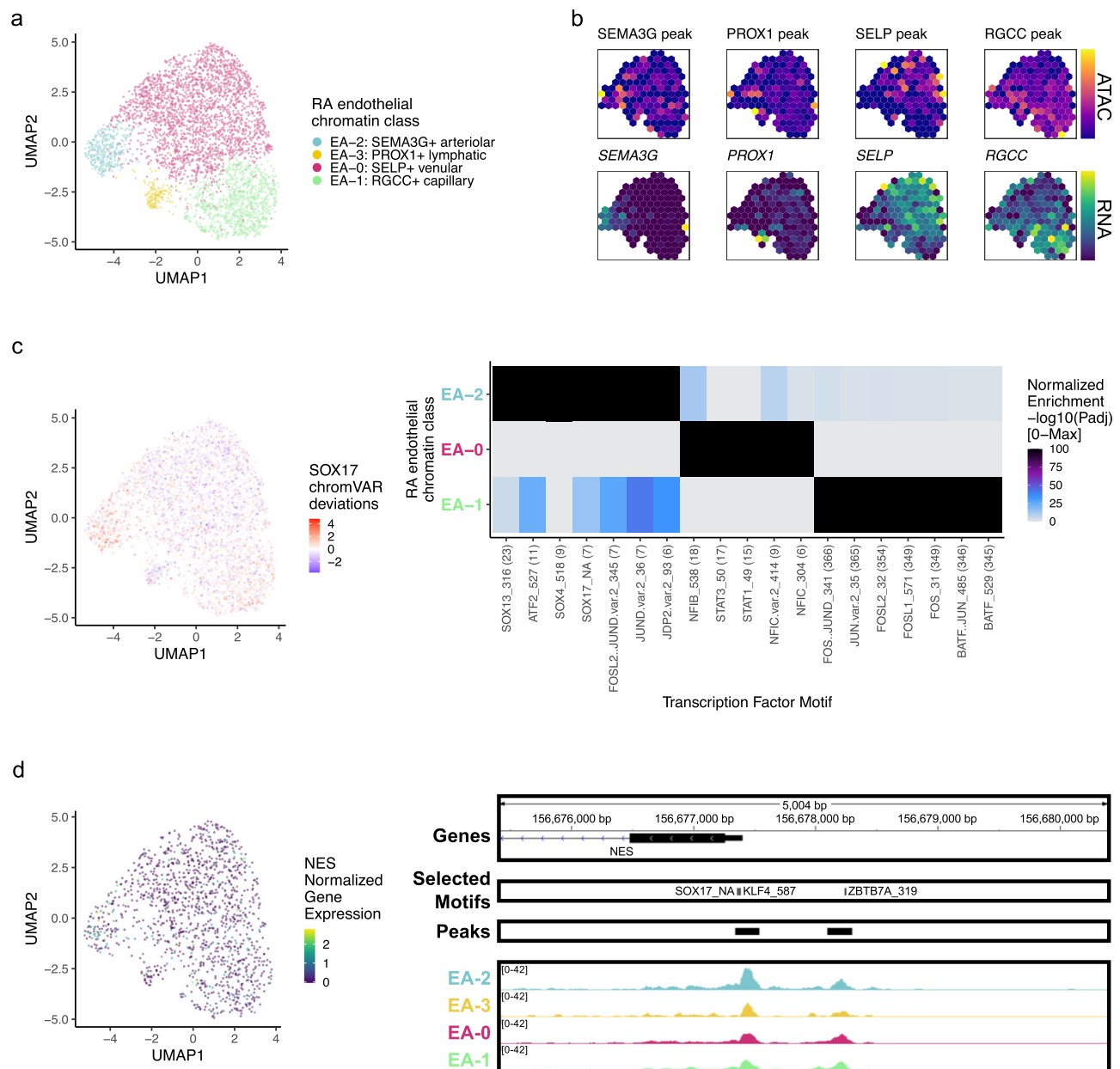

**Fig. 6 | RA endothelial chromatin classes. a** UMAP colored by 4 endothelial chromatin classes defined from unimodal scATAC-seq and multimodal snATAC-seq cells. **b** Mean binned normalized marker peak accessibility (top; yellow (high) to purple (low)) and gene expression (bottom; yellow (high) to blue (low)) for multimodal snATAC-seq cells on UMAP. **c** UMAP colored by chromVAR[34] deviations for the SOX17 motif (left). Most significantly enriched motifs in class-specific peaks per endothelial chromatin class (right). To be included per class, motifs had to be enriched in the class above a minimal threshold, and corresponding TFs had to have at least minimal expression in snRNA-seq. Color scale normalized per motif across classes with max −log10($p_{adj}$) value shown in parentheses in motif label. $P$ values were calculated via hypergeometric test in ArchR[35]. $E_A$−3 is not shown because only 1 class-specific peak was found, likely due to low cell counts. **d** UMAP colored by *NES* normalized gene expression (left). NES locus (chr1: 156,675,398–156,680,400) with selected gene isoforms, motifs, open chromatin peaks, and chromatin accessibility reads from unimodal and multiome cells aggregated by chromatin class and scaled by read counts per class (right).

B/plasma chromatin classes (Supplementary Fig. 8c), suggesting that these low-cell-count samples did not drive our original classes. We saw similar results in endothelial cells after removing 954 cells across 19 samples (Supplementary Fig. 8d). These analyses suggested our chromatin classes were robust to the inclusion of both OA and low-cell-count samples.

**Synovial tissue is key to identifying pathogenic RA chromatin classes**

To determine if the chromatin classes identified in RA tissue were comparable with the known peripheral blood chromatin landscape, we clustered the tissue cells with those from a published healthy PBMC multiome dataset[92,93] (Supplementary Fig. 9). To determine the similarity between the PBMC and tissue chromatin classes, we calculated the OR between the newly defined clusters and the original blood and tissue labels; overall, there was good concordance. For example, the PBMC Treg cells and $T_A$−3: CD4+ IKZF2+ Treg cells were both enriched in T cell combined cluster 5 (OR = 12 and 85, respectively) (Supplementary Fig. 9a) and PBMC cDC2 and pDC associated with $M_A$−3: CD1C+ AFF3 + DC in myeloid combined cluster 4 (OR = 45, 78, and 100, respectively) (Supplementary Fig. 9b). However, there were some tissue chromatin classes that did not have clear counterparts in PBMCs,

such as $T_A$−2: CD4+ PD-1+ TFH/TPH, $M_A$−2: LYVE1+ TIMD4+ TRM, $M_A$−4: SPP1+ FABP5+ intermediate, and $B_A$−5: ITGAX + ABC (Supplementary Fig. 9). With the current dataset, we cannot conclusively determine whether these disparities reflect tissue and blood or RA and healthy differences. However, prior studies have shown both that these cell states are tissue-enriched[12,71,94] and implicated in RA pathogenesis[11–13,16,61], suggesting that the study of disease tissue is necessary for well-powered analyses of these populations.

## Chromatin classes are epigenetic superstates of transcriptional cell states

To understand how these chromatin classes corresponded to transcriptionally defined cell states, we used Symphony[95] to map the RA multimodal snRNA-seq profiles into the well-annotated AMP-RA cell type references[14]. After embedding the multimodal snRNA-seq profiles into the AMP-RA reference data, we annotated each multimodal cell by the most common cell state of its five nearest reference neighbors. 70% of T cells (24 states), 96% of stromal cells (10 states), 96% of myeloid cells (15 states), 87% of B/plasma cells (9 states), and 99% of endothelial cells (5 states) mapped well (i.e., at least 3/5 neighbors had the same cell state annotation). We also observed that the proportion of each cell state in the AMP-RA reference and the multimodal query datasets was consistent, suggesting that the reference and query datasets had comparable cell state distributions despite different technologies (Supplementary Fig. 10a−e).

We then sought to understand the correspondence between the mapped transcriptional cell states and chromatin classes. We calculated an OR for each combination of state and class to measure the strength of association and used a Fisher's exact test to assess significance. We observed that each transcriptional cell state generally corresponded to a single chromatin class (Fig. 7a–c; Supplementary Fig. 10g, h). In contrast, a single chromatin class represented a superstate encompassing multiple transcriptionally defined cell states. For example, cells in the $T_A$−0: CD8A+ GZMK+ chromatin class were more likely to be labeled in the T-5: CD4+ GZMK+ memory, T-13: CD8+ GZMK/B+ memory, or T-14: CD8+ GZMK+ transcriptional cell states across CD4/CD8 lineages (OR = 11, 12, 11, respectively; Fig. 7a); the high *GZMK* promoter accessibility and expression shared by these states may have contributed to this categorization (Supplementary Fig. 10f). We saw examples of this model in every cell type: $S_A$−1 linked to F-0/F-1 and $S_A$−0 to F-6/F-5/F-3/F-8 in stromal cells; $M_A$−1 to M-7/M-11 and $M_A$−4 to M-3/M-4 in myeloid cells; $B_A$−4 to B-1/B-3 in B/plasma cells; and $E_A$−0 to E-1/E-2 in endothelial cells (Fig. 7b, c; Supplementary Fig. 10g, h; Supplementary Data 3). In all cell types, the transcriptional cell state classification was more accurate within cells whose transcriptional cell state and chromatin class were concordant (e.g., T-14 and $T_A$−0), supporting our class-to-state mapping (Supplementary Fig. 10i).

Indeed, when we aggregated the snATAC-seq reads by states, we observed shared openness between transcriptional cell states within the same class (i.e., superstate), as seen with the cytotoxic $T_A$−4 grouped cell states T-12/T-15 at the cytotoxicity-associated[32] *FGFBP2* gene, lining fibroblast $S_A$−1 grouped cell states F-0/F-1 at the lining-associated[11] *CLIC5* gene, and intermediate myeloid $M_A$−4 grouped cell states M-3/M-4 at bone marrow-derived macrophage-associated[60] *SPP1* gene (Supplementary Fig. 11). Furthermore, we found very few differential promoter peaks between transcriptional states in the same chromatin class even after pseudobulking by sample and state to decrease sparsity (Supplementary Fig. 12a). $T_A$−1: CD4+ IL7R+ had one of the higher numbers of differential peaks within a class, but still only found 1.3% of the peaks tested as differential at FDR < 0.10. Among those was the expected *CD4* and *CD8A* promoter peaks since both the T-4: CD4+ naive state and T-16: CD8+ CD45ROlow/naive state corresponded to $T_A$−1 (Supplementary Fig. 12b; Fig. 7a). These populations likely mapped together since they shared naïve T cell transcriptional

profiles, consistent with a highly accessible *SELL* promoter peak. This contrasted sharply to the number of differential peaks found between states across classes within a cell type (median of 8717 within a cell type vs 23 within a single class; Supplementary Fig. 12a), suggesting that the chromatin landscape in states within a class is much more homogeneous than across classes, as proposed by our superstate model.

We next asked if evidence for chromatin superstates was sensitive to clustering resolution. We observed that the class and state relationships largely replicated when we increased the open chromatin clustering resolution (Supplementary Fig. 13). To further support the superstate hypothesis, we trained a linear discriminant analysis (LDA) model to predict the transcriptional cell state between each pair of states from the chromatin PCs, upon which the chromatin classes were defined. Generally, transcriptional cell states belonging to the same chromatin class were difficult to distinguish using chromatin accessibility data alone (Supplementary Fig. 14). As an example, transcriptional states T-14 and T-13 both belonged to chromatin class $T_A$−0, and thus chromatin PCs could not easily discriminate between them (AUC = 0.61); on the other hand, T-14 and T-3 belonged to classes $T_A$−0 and $T_A$−2, respectively, and LDA nearly perfectly distinguished them (AUC = 0.98) (Supplementary Fig. 14a). In all cell types, the mean AUC between states within the same chromatin class was less than that of states across different chromatin classes. For instance in T cells, the mean AUC was 0.77 within the same classes and 0.88 across different classes, suggesting there was a limit to how well the chromatin accessibility data could differentiate between transcriptional cell states.

Finally, to more thoroughly investigate the validity of the chromatin superstate model, we profiled the chromatin accessibility and transcriptomes of select cell states known to be functionally distinct and defined by well-characterized surface markers[12,96]. We generated a multiome dataset of sorted RA PBMC subsets via fluorescence-activated cell sorting (FACS) of four populations spanning two chromatin classes and four transcriptional states: $CD4^+CD127^-CD25^{hi}$ Treg, $CD4^+CD127^-CD25^{int}$ Treg, $CD4^+CD25^-PD1^+CXCR5^+$ TFH, and $CD4^+CD25^-PD1^+CXCR5^-$ TPH (Supplementary Fig. 15a). We performed quality control steps in all three modalities and identified FACS cell state labels before doing any downstream analysis for the remaining 2,998 cells (Supplementary Fig. 15b). When we de novo clustered the chromatin accessibility data of the combined PBMC and tissue cells (Supplementary Fig. 15c), we found that the sorted RA PBMC TFH/TPH cells were most enriched in combined cluster 2 (OR = 4), which was most highly enriched for RA tissue TFH/TPH cells (OR = 32). Similarly, sorted RA PBMC Tregs were most enriched for combined cluster 4 (OR = 3), which was most highly enriched for RA tissue Tregs (OR = 24). This confirmed that our tissue class annotations agreed with well-known subclasses of T cells sorted using established protein markers.

We also wanted to assess whether the two cell states within a chromatin class defined via cell surface proteins (e.g., $CD4^+CD25^-PD1^+CXCR5^+$ TFH and $CD4^+CD25^-PD1^+CXCR5^-$ TPH) were transcriptionally distinct. By clustering the cells from the four sorted populations based on gene expression, we successfully distinguished between the pairs of transcriptomic states from each chromatin class (Supplementary Fig. 15d). Moreover, we observed that each gold-standard FACS-defined population had a distinct mRNA cluster identity. Next, we calculated the differentially expressed genes and differentially accessible promoter peaks between the transcriptional states within the same class. While we found significant transcriptional differences, we largely did not observe similar accessibility differences in the corresponding genes' promoter peaks (Fig. 7d, e). This was consistent with the model of transcriptional cell states from a common superstate sharing open chromatin landscapes. For example, the *PDE4D* gene, which encodes an RA treatment target[97], had significantly more expression in TPH than TFH cells (unadjusted $P$ = 4.64e-19), but a

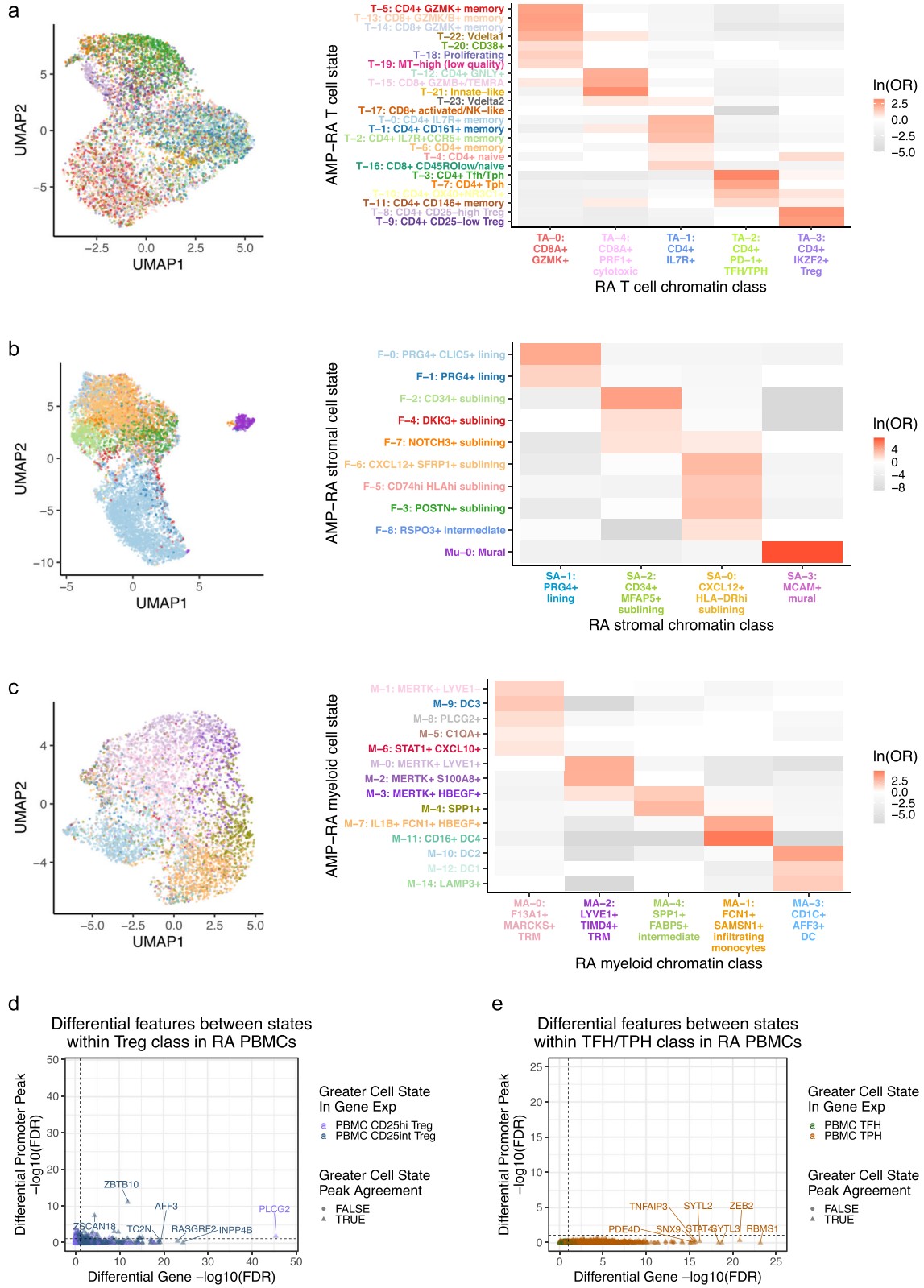

**Fig. 7 | A chromatin class encompassed multiple transcriptional cell states in proposed superstate model. a–c** For **a** T, **b** stromal, and **c** myeloid cells, chromatin class UMAP colored by the classified AMP-RA reference transcriptional cell states for multiome cells (left) and the natural log of the odds ratio between the chromatin classes and transcriptional cell states (right). On the right, non-significant values (FDR < 0.05) are white, and the colors of the y axis labels correspond to the colors in the UMAPs on the left. In **c**, the M-13: pDC transcriptional cell state was excluded as fewer than 10 cells were classified into it. **d**, **e** Using genes and promoter peak pairs with at least minimal signal, the two-sided Wilcoxon −log10(FDR) of normalized gene expression (x axis) and the logistic regression LRT −log10(FDR) of binary promoter peak accessibility (y axis) between **d** RA PBMC CD25hi and CD25int Treg populations (n = 7208 pairs) and **e** RA PBMC TFH and TPH populations (n = 5264 pairs). Color was determined by the state with the higher gene expression and the shape denotes whether the state with the higher chromatin accessibility agreed. The dotted lines correspond to FDR = 0.10, calculated separately within modalities.

non-significant change in the promoter peak accessibility (unadjusted $P = 0.913$) (Supplementary Fig. 15e). On the other hand, *ZBTB10*, a telomere-associated TF[98], was a rare example where the chromatin accessibility and gene expression concurred across Treg states (Supplementary Fig. 15f). However, globally, the lack of these examples likely contributed to the lack of fully distinguished state-specific chromatin classes.

## Cell neighborhood associations with histological metrics and cell state proportions

Next, we sought to investigate associations between the RA chromatin classes and RA clinical metrics using the larger AMP-RA reference dataset with clinical measurements for 82 RA or OA patients. Per cell type, we classified[95] each cell from the AMP-RA reference dataset, now the query, into the RA chromatin classes based on the five nearest multimodal snRNA-seq neighbors, now the reference. To validate this annotation, we compared the relative proportions of chromatin classes between the unimodal scATAC-seq cells and the classified AMP-RA scRNA-seq cells for donors in both studies. We observed a generally high correlation between the two technologies (Fig. 8a; Supplementary Fig. 16a). We then investigated RA clinical associations calculated via Co-varying Neighborhood Analysis (CNA)[99]. In brief, CNA tests associations between sample-level attributes, such as clinical metrics, and cellular neighborhoods, which are small groups of cells that reflect granular cell states. We used the previously described CNA associations defined in the AMP-RA reference cells and re-aggregated them by their chromatin classes. For example, we found an association between myeloid cells and histology characterized by lymphoid infiltration density ($p = 0.005$). Specifically, the increase in lymphocyte populations was positively associated with the $M_A$−4: SPP1+ FABP5+ intermediate class, whose inflammatory cytokines/chemokines production may be responsible for lymphocyte homing[100], and negatively associated with $M_A$−2: LYVE1+ TIMD4+ TRM, whose gene markers were found more often expressed in synovial TRMs from healthy and remission RA than active RA patients[13] (Fig. 8b). Additionally, we observed an association between T cells and the histological Krenn inflammation score ($p = 0.02$), with $T_A$−2: CD4+ PD-1+ TFH/TPH positively[101] and $T_A$−4: CD8A+ PRF1+ cytotoxic negatively correlated (Supplementary Fig. 16b). These results were consistent with the original transcriptional cell state findings[14] and suggested that the connections between RA pathology and cell state may begin before transcription.

One of the key findings from the AMP-RA study was the identification of six Cell Type Abundance Phenotypes (CTAPs), which characterized RA patients into subtypes based on the relative proportions of their broad cell type abundances in synovial tissue[14]. For instance, CTAP-TB has primarily T and B/plasma cells. Specific cell neighborhoods within cell types were expanded or depleted in these CTAPs as defined by CNA associations in the AMP-RA reference cells. We recapitulated some of these transcriptional associations by re-aggregating the CNA results within the chromatin classes; for example, the RA T cell class $T_A$−2 was positively associated with CTAP-TB compared to other T cell classes, likely reflecting the role of TFH/TPH cells in B cell inflammation response[11,12], while $T_A$−4 was negatively associated ($p = 0.046$; Fig. 8c). Furthermore, in stromal cells, we saw the $S_A$−1: PRG4+ lining class positively associated with CTAP-F, a primarily fibroblast CTAP ($p = 0.0027$; Supplementary Fig. 16c). This indicated that the most expanded type of fibroblasts in CTAP-F individuals was predominantly from the synovial lining layer, which was consistent with lining marker CLIC5 protein having high staining in the lining fibroblasts and being expressed in the highest proportion of cells from high-density fragments of CTAP-F samples (ANOVA $p_{adj} = 4.92\text{e-}03$ between CTAPs)[14]. Therefore, we could meaningfully replicate the RA pathological associations of both clinical metrics and phenotypic subtypes to transcriptional cell states using their related chromatin class superstate, suggesting that the epigenetic regulation underlying the transcriptional cell states may be mined for further pathological insights into RA.

## Chromatin classes prioritize RA-associated SNPs

We next asked whether RA risk variants overlapped the chromatin classes to help define the function of putatively causal variants, genes, and pathways at play in RA pathology[102–106]. Using an RA multi-ancestry genome-wide association meta-analysis study[107], we overlapped fine-mapped non-coding variants with posterior inclusion probability (PIP) greater than 0.1 with the 200 bp open chromatin peaks and assessed peak accessibility across the 24 chromatin classes (Fig. 8d; Supplementary Table 5). For six loci, putatively causal variants overlapped a peak accessible in predominantly one cell type, such as rs11209051 in peak chr1: 67,333,106–67,333,306 in T cells (Wilcoxon T versus non-T class one-sided $p = 4.17\text{e-}04$) near the *IL12RB2* gene and rs4840568 in peak chr8:11,493,501–11,493,701 in B/plasma cells (Wilcoxon $p = 1.49\text{e-}05$) near the *BLK* gene. In the other loci, variants overlapped with chromatin classes from two cell types, with most combinations involving T cells. There were four SNPs overlapping peaks accessible in the $T_A$−2: CD4+ PD-1+ TFH/TPH class, which was the most targeted class within T cells and known to be important for RA pathogenesis[11,12].

As an example, we observed the putatively causal SNP rs798000 (PIP = 1.00) overlapped with peak chr1: 116,737,968–116,738,168, accessible primarily in T cells (Wilcoxon $p = 2.35\text{e-}05$) with $T_A$−2 as its most accessible class ($z = 3.03$) (Fig. 8d, e, top). In a previous study[93], we linked active chromatin regions to their target genes, which suggested *CD2* was the causal gene in this locus. *CD2* is a co-stimulatory receptor primarily expressed in T and NK cells[108], which likely explains why it was only accessible in our T cell chromatin classes among the five cell types investigated (Fig. 8e, bottom). Intriguingly, rs798000 overlaps a STAT1/2 binding site at a high information content half site position (Fig. 8e, top, position 8 in JASPAR[109] motif MA0517.1), suggesting a potential direct link to TF regulation of the JAK/STAT pathway commonly upregulated in RA[53].

We also discovered SNP rs9927316 (PIP = 0.54) in myeloid-specific peak chr16:85,982,638–85,982,838 (Wilcoxon $p = 4.17\text{e-}04$), downstream of *IRF8*, one of the master regulator TFs of myeloid and B cell fates[110–112] (Supplementary Fig. 17a). The SNP disrupts a KLF4 motif[52], one of the TRM TFs highlighted earlier (Supplementary Fig. 17a; Fig. 4c, d). Furthermore, we observed SNP rs734094 (PIP = 0.41) overlapping peak chr11:2,301,916–2,302,116 with its most accessible classes in T and myeloid cells: $T_A$−4: CD8A+ PRF1+ cytotoxic and $M_A$−3: CD1C+ AFF3+ DC ($z = 1.94, 1.65$, respectively) (Fig. 8d; Supplementary Fig. 17b). While existing in the promoters of both *TSPAN32* and *C11orf21* gene isoforms (Supplementary Fig. 17b), we[93] proposed the causal gene as Lymphocyte-specific Protein 1 (*LSP1*), shown to negatively regulate T cell migration and T cell-dependent inflammation in arthritic mouse models[113].

For each of these loci, we also aggregated chromatin accessibility reads by classified transcriptional cell state and saw that the multiple states underlying each class had similar patterns, such as rs734094 having some of the strongest signal in $T_A$−4 associated classes T-12, T-21, and $M_A$−3 associated classes M-10, M-14 (Supplementary Fig. 18). This both reaffirmed our chromatin class superstate model and suggested that the classes are useful functional units that simplify mapping risk loci to affected cell states. The RA tissue chromatin classes can help prioritize putative cell states of action for non-coding RA risk variants to assist in their functional characterization within disease etiology.

## Discussion

In this study, we described 24 chromatin classes across 5 broad cell types in 30 synovial tissue samples assayed with unimodal

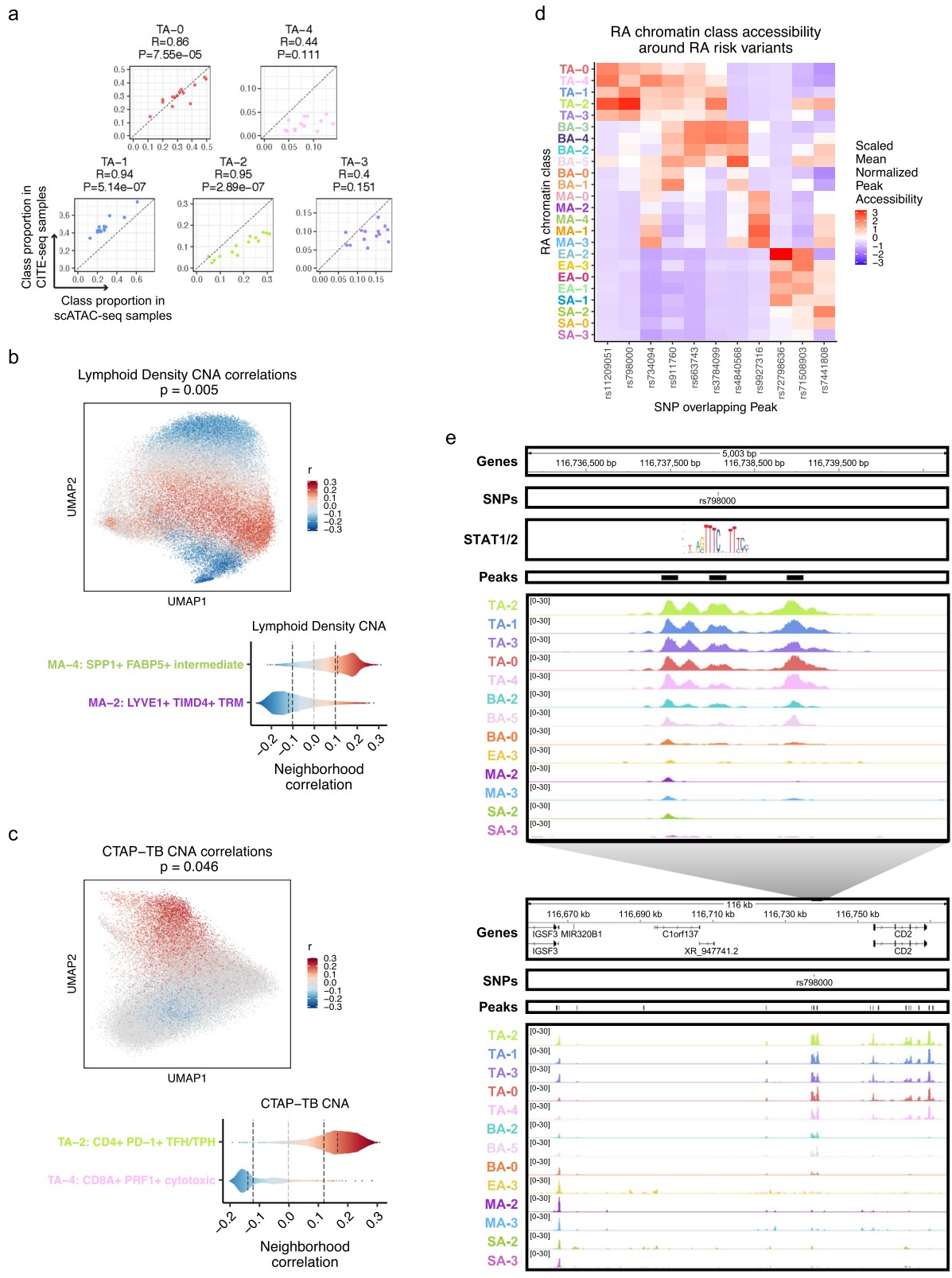

scATAC-seq and multimodal snATAC-seq along with the TFs potentially regulating them. Based on our observation that cells from the same chromatin class corresponded to multiple transcriptional cell states, we proposed that these chromatin classes were putative superstates of related transcriptional cell states. Finally, we assessed these chromatin classes' relationship to RA clinical metrics, subtypes, and genetic risk variants. Our main findings are summarized in Supplementary Table 6 and Supplementary Data 4.

Chromatin accessibility is a key piece in the puzzle of gene regulation. It determines which regions of the genome may participate in regulatory events such as TF binding or may be impacted by noncoding genetic variants. Accessible TF motifs are not guaranteed to be bound, in contrast to the regions identified in gold-standard TF ChIP-

**Fig. 8 | Linking RA chromatin classes to RA pathology. a** For each donor of the 14 donors shared between the unimodal scATAC-seq and AMP-RA reference studies with at least 200 T cells, the Pearson correlation coefficient (R) and two-sided p value (P) between the relative proportions of T cell chromatin classes defined in the unimodal scATAC-seq datasets (x axis) and classified into in the CITE-seq datasets through the multiome cells (y axis). **b** CNA correlations between myeloid cell neighborhoods and lymphoid density in AMP-RA reference myeloid cells visualized on chromatin class UMAP (top; two-sided global P = 0.005) and aggregated by classified myeloid chromatin classes (bottom). On the top, cells not passing the FDR threshold were colored grey. On the bottom, FDR thresholds shown in dotted black lines. **c** CNA correlations between T cell neighborhoods and CTAP-TB in AMP-RA reference T cells visualized on chromatin class UMAP (top; two-sided global

P = 0.046) and aggregated by classified T cell chromatin classes (bottom). On the top, cells not passing the FDR threshold were colored grey. On the bottom, FDR thresholds shown in dotted black lines. **d** Scaled mean normalized chromatin accessibility for peaks that overlap putatively causal RA risk variants across chromatin classes in unimodal and multimodal datasets. Additional information is in Supplementary Table 5. **e** rs798000 locus, zoomed in (chr1: 116,735,799–116,740,800) (top) and zoomed out (chr1: 116,658,581–116,775,106) (bottom) with selected gene isoforms, SNPs, open chromatin peaks, and chromatin accessibility reads aggregated by chromatin class and scaled by read counts per class. STAT1/2 motif was downloaded from JASPAR[109] ID MA0517.1 and is not to scale, but it is aligned to the SNP-disrupting motif position.

seq[114] or CUT&RUN[115]. However, chromatin accessibility datasets are not TF-specific or dependent on antibodies, so they can capture potential regulatory sites for a broader set of factors. At a small scale, the regulation of key loci can be interrogated using scATAC-seq. For example, we found accessible AP-1 motifs in the differentially accessible promoter peak of *MMP3*, a key driver of RA extracellular matrix destruction[51], in lining fibroblasts compared to other stromal cells (Fig. 3c, d). Multiple drugs (e.g., CKD-506, T-5224, Roflumilast) are under investigation to disrupt this specific interaction of AP-1 at the *MMP3* promoter, and AP-1 signaling targets more broadly, in models of arthritis as well as clinical trials of RA patients[116]. At a large scale, these TF-gene interactions can be linked together to form gene regulatory networks in silico[117,118] to interrogate the more widespread effects of disrupting signaling cascades. Furthermore, as ~90% of disease causal genetic variants fall in non-coding regions[119], chromatin accessibility can prioritize where to look for functional effects of putatively causal RA genetic variants, particularly for those that disrupt TF motifs. Our analyses suggested that the likely causal SNP rs798000 may disrupt STAT binding in a TFH/TPH regulatory region reported to act on *CD2*, an important T cell co-stimulatory gene[120,121]. Therefore, our study underscores the value of chromatin accessibility studies in disease-specific transcriptional regulation.

Simultaneous chromatin accessibility and gene expression measurements in the multiome cells were essential to test the relationship between marker peaks and genes. Across cell types, the correlations between scaled marker peak accessibility and gene expression across our chosen markers varied. T cells had higher correlation (R = 0.92; Supplementary Fig. 3b) while myeloid cells had lower correlation (R = 0.76; Supplementary Fig. 5b), potentially due to more heterogeneous subpopulations such as TRMs, infiltrating monocytes, and dendritic cells. Furthermore, when we did not see class correspondence between chromatin accessibility and gene expression on the individual gene level, we observed more class-specific gene expression in the context of promiscuous chromatin accessibility. This suggested a poised chromatin state that depends on the presence of a specific TF or extracellular signal to give rise to a particular transcriptional outcome. For example, the promoter peak of *RTKN2* was accessible in all CD4 T cells, but the gene was primarily expressed in Tregs (Supplementary Fig. 3b), likely because it is a direct target of the Treg master regulator FOXP3[122]. *CCL2* in stromal fibroblasts had an accessible promoter peak in both subliningr populations, but was primarily expressed in the inflammatory subset (Supplementary Fig. 4b), likely due to stimulation by TNF/INFγ[44,123].

Indeed, when expanding genome-wide, we saw a similar pattern of class-specific transcriptional cell states but chromatin classes encompassing multiple related states in our proposed superstate model (Fig. 7a–c; Supplementary Fig. 10g, h). To validate this model, we conducted an RA PBMC multiome experiment of FAC-sorted populations. While we saw differentially expressed genes between transcriptional cell states within a chromatin class, there was an almost complete lack of differentially accessible promoter peaks corresponding to those genes (Fig. 7d, e). Biologically, open chromatin is

necessary but not sufficient for gene expression[18], so it is reasonable to expect related cell states to have similar open chromatin landscapes with further specificity coming from TFs among other epigenetic regulators. Technically, the robustness of the observed class-state relationships across multiple clustering resolutions mitigated concerns that this proposed model was an artifact (Supplementary Fig. 13). Even in the absence of clusters, classifiers based on continuous chromatin PCs also demonstrated the lack of resolution chromatin accessibility has to distinguish between similar transcriptional states (Supplementary Fig. 14).

Defining the relationship between transcriptional cell states and chromatin classes may have important therapeutic implications. One effective RA treatment strategy is the deletion of a pathogenic cell state: the use of B cell-depleting antibodies (e.g., rituximab[10]) is an example. However, if one chromatin class corresponds to multiple transcriptional cell states, then deleting very specific pathogenic populations may be ineffective as other non-pathogenic states may transition into the pathogenic state in response to the same pathogenic tissue environment. As an example, a recent study[124] of ILCs in a mouse model of psoriasis showed chromatin accessibility in a disease-relevant population of ILC3s even before disease induction using IL-23, particularly at ILC3 TFs, that then increased further after induction. In that case, altering the environment or removing exogenous factors (e.g., TFs, cytokines) might be a more effective treatment. Within RA, the $S_A-0$: CXCL12+ HLA-DR$^{hi}$ sublining fibroblast class, with its four related transcriptional states in our superstate model, may merit further investigation in this regard. $S_A-0$ accessible peaks were enriched for STAT motifs, suggesting potential regulation by the JAK/STAT signaling pathway. Indeed, JAK inhibition via tofacitinib and upadacitinib has been shown to prevent pro-inflammatory HLA-DR induction in RA synovial fibroblasts[125]. Additional experiments would be required to determine if the F-3: POSTN+ sublining transcriptional cell state could transform into the RA-expanded[14] F-5: CD74$^{hi}$HLA$^{hi}$ sublining or F-6: CXCL12+ SFRP1+ sublining fibroblast populations under JAK/STAT stimulation.

More broadly, the results presented here suggest some interesting next steps. First, our chromatin class superstate model indicated that certain transcriptional cell states were more closely linked, but further experimentation would be required to ascertain whether these related cell states have a plastic enough chromatin landscape to allow for cross-differentiation or whether they are more broadly grouped by function. Second, to better understand whether the more pathogenic chromatin classes such as $T_A-2$: CD4+ PD-1+ TFH/TPH and $M_A-1$: FCN1+ SAMSN1+ infiltrating monocytes are indeed only in tissue, a RA PBMC scATAC-seq study may be warranted. While we saw a general consensus between the chromatin landscapes of RA tissue class $T_A-2$ and our small population of RA blood TFH/TPH cells, a larger PBMC study would be better powered to determine if the chromatin environment in blood may be a proxy for the environment in tissue that gives rise to pathogenic transcriptional populations. Third, even though we did not see large effects of OA and low-cell-count samples on our chromatin classes, a larger study with a more even distribution

of RA and OA samples with higher cell counts would be better able to distinguish between RA- and OA-specific chromatin variation.

In conclusion, we presented an atlas for RA tissue chromatin classes that will be a useful resource for linking chromatin accessibility to gene expression and the interpretation of genetic information.

# Methods

## Patient recruitment

Fourteen RA and 4 OA patients were recruited by the Accelerating Medicines Partnership (AMP) Network for RA and SLE to provide samples for use in the unimodal scATAC-seq experiments. Separately, synovial tissue samples from 11 RA patients and 1 OA patient were collected from Brigham and Women's Hospital (BWH) and the Hospital for Special Surgery (HSS) for use in the multimodal ATAC + Gene Expression experiments. Histologic sections of RA synovial tissue were examined, and samples with inflammatory features were selected in both cases.

Patients were recruited from Brigham and Women's Hospital, Columbia University, Hospital for Special Surgery, Queen Mary University of London UK, University of Birmingham UK, University of California San Diego, University of Pittsburgh, University of Rochester. All sites obtained approval for this study from their Institutional Review Boards. All patients gave written informed consent. We have complied with all relevant ethical regulations.

## Synovial tissue collection and preparation

Synovial tissue samples from 14 RA patients and 4 OA patients were collected and cryopreserved as part of a larger study cohort by the AMP Network for RA and SLE, as previously described[14]. Synovial tissue samples were thawed and disaggregated as previously described[14,23]. The resulting single-cell suspensions were stained with anti-CD235a antibodies (clone 11E4B-7-6 (KC16), Beckman Coulter, 1:100 dilution) and Fixable Viability Dye (FVD) eFlour 780 (eBioscience/Thermo-Fisher). Live non-erythrocyte (i.e., FVD− CD235−) cells were collected by fluorescence-activated cell sorting (BD FACSAria Fusion). The sorted live cells were then re-frozen in Cryostor and stored in liquid nitrogen. The cells were later thawed and processed as described above for droplet-based scATAC-seq according to manufacturer's protocols (10X Genomics). For the multimodal experiments, the 11 RA and 1 OA synovial tissue samples were collected and cryopreserved before being thawed, disaggregated, and FAC-sorted, as described above.

## Unimodal scATAC-seq experimental protocol

Unimodal scATAC-seq experiments were performed by the BWH Center for Cellular Profiling. Each sample was processed separately in the cell capture step. Nuclei were isolated using an adaptation of the manufacturer's protocol (10X Genomics). Approximately ten thousand nuclei were incubated with Tn5 Transposase. The transposed nuclei were then loaded on a Chromium Next GEM Chip H and partitioned into Gel Beads in-emulsion (GEMs), followed by GEM incubation and library generation. The ATAC libraries were sequenced to an average of 30,000 reads per cell with the recommended number of cycles according to the manufacturer's protocol (Single Cell ATAC V1.1, 10X Genomics) using Illumina Novaseq. Samples were initially processed using 10x Genomics Cell Ranger ATAC 1.1.0, which included barcode processing and read alignment to the hg38 reference genome.

## Multiome experimental protocol

Multiome experiments were performed by the BWH Center for Cellular Profiling. Each sample was processed separately in the cell capture step. Nuclei were isolated as above. Approximately ten thousand transposed nuclei were loaded on Chromium Next GEM Chip J followed by GEM generation. 10x Barcoded DNA from the transposed DNA (for ATAC) and 10x Barcoded, full-length cDNA from poly-adenylated mRNA (for Gene Expression) were produced during GEM

incubation. The ATAC libraries and Gene Expression libraries were then generated separately. Both library types were sequenced to an average of 30,000 reads per cell on different flow cells with the recommended sequencing cycles according to the manufacturer's protocol (Chromium Next GEM Single Cell Multiome ATAC + Gene Expression, 10X Genomics) using Illumina Novaseq. Samples were initially processed using 10x Genomics Cell Ranger ARC 2.0.0, which included barcode processing and read alignment to the hg38 reference genome, for both ATAC and GEX information.

## Computational methods

Supplementary Fig. 1 shows an overview of the computational methodology for cell type/state identification, as many of the methods were reused in different contexts. In the following sections, we explain the core methodology the first time it is used, and then only the ways in which the methodology differs in the different contexts afterwards.

## ATAC read QC

Reads were quality controlled from the Cell Ranger BAM files via a new cell-aware strategy that removes likely duplicate reads from PCR amplification bias within a cell while keeping reads originating from the same positions but from different cells. For unimodal scATAC-seq data, duplicate reads from the same cell were called based on read and mate start positions and CIGAR scores, but the multimodal snATAC-seq data only used start positions since Cell Ranger ARC did not provide a mate CIGAR score (MC:Z flag). Reads that were not properly mapped within a pair, had a $MAPQ < 60$, did not have a cell barcode, or were overlapping the ENCODE blacklisted regions[24] of 'sticky DNA' were also removed. Using the deduplicated BAM files, we converted them to fragment BED files using BEDOPS[126] bam2bed while accounting for the 9-bp Tn5 binding site.

## ATAC peak calling

Peaks were called twice on the unimodal scATAC-seq cells, before and after "ATAC cell QC", to first provide general peak information to be used in the cell QC step and then afterwards on the post QC cells to provide the final, refined peak set. Individual sample unimodal scATAC-seq BAM files were converted to MACS2[127] BEDPE files using macs2 randsample, concatenated across samples, and then used to call consensus peaks with macs2 callpeak --call-summits using a control file[128] where ATAC-seq was done on free DNA to account for Tn5's inherent cutting bias. Each sub-peak was trimmed to 200 bp (summit ± 100 bp) to localize the signal and avoid confounding any statistical analysis with peak length. Any overlapping peaks were removed iteratively, keeping the best sub-peak, as determined by q-value, to avoid double counting. For consistent analysis, we used the post cell QC unimodal scATAC-seq trimmed consensus peaks for all downstream analyses unless otherwise stated. We wanted to confirm that these unimodal scATAC-seq consensus peaks were reasonable to use for the multimodal snATAC-seq datasets, beyond just that the datasets were done on the same tissue type. Therefore, we called peaks, as done above, on the individual sample multimodal snATAC-seq BAM files and found that an average of 75% ($n = 12$ samples; range: 66–83%) of the 200 bp trimmed multimodal snATAC-seq sample-specific peaks overlapped the unimodal scATAC-seq consensus peaks. Furthermore, we used the 5x full consensus peak neighborhoods in the cell QC step for multiome datasets as an added safeguard. We also confirmed our peaks' quality by seeing good overlap with ENCODE SCREEN v3 candidate cis-regulatory elements (cCREs)[25] and the GENCODE v28[26] promoter annotations via bedtools[129] intersectBed (Supplementary Fig. 2f).

## ATAC cell QC

We kept cells with more than 10,000 reads with at least 50% of those reads falling in peak neighborhoods (5x full peak size), at least 10% of

reads in promoter regions, not more than 10% of reads called in the mitochondrial chromosome, and not more than 10% of pre-deduplication reads falling in the ENCODE backlisted regions[24]. The genome annotation we used to define promoters was GENCODE v28 basic[26] as was done for Cell Ranger ATAC read mapping; we defined promoter regions for the QC step as 2 kb upstream of HAVANA protein coding transcripts that we subsequently merged to avoid double counting. The fragments from the post QC cells were quantified within the 200 bp trimmed consensus peaks (see "ATAC peak calling") via GenomicRanges::findOverlaps[130] into a peaks x cells matrix.

## ATAC clustering

We did multiple rounds of clustering with different inputs. Generally, we did: binarize peaks x cells matrix, log(TFxIDF) normalization using Seurat::TF.IDF[131], most variable peak feature selection using Symphony::vargenes_vst[95], center/scale features to mean 0 and variance 1 across cells using base::scale, PCA dimensionality reduction using irlba::prcomp_irlba, batch correction by sample using Harmony::HarmonyMatrix[27], shared nearest neighbor creation using RANN::nn2 and Seurat::ComputeSNN[131], Louvain clustering using Seurat::RunModulatrityClustering[131], and cluster visualization using UMAP coordinates via umap::umap. For the unimodal scATAC-seq feature selection, we chose peaks that had at least one fragment in at least five percent of cells and TFxIDF normalization using Seurat::TF.IDF[131] before continuing in the above steps. We used 20 PCs for the broad cell type clustering and 10 PCs for the chromatin class clustering since there was less variation within a cell type.

For cluster identification, we used marker peaks, defined as peaks overlapping the promoters of marker genes; if there were multiple peaks overlapping a gene's promoter or multiple isoforms of a gene, the peak that best tracked with the gene's expression in the multiome cells was chosen. The broad cell type marker peaks we used are in Supplementary Fig. 2g–j and the chromatin class marker peaks in panel b of Supplementary Figs. 3–7.

## ATAC doublet cluster removal

Within the unimodal scATAC-seq and multimodal snATAC-seq separately, we then did an initial round of ATAC clustering using all post cell QC cells to find doublet clusters. We removed doublet clusters with multiple cell-type-specific marker peaks, intermediate placement between broad cell type clusters in PC space, high fragment counts, and high doublet scores determined per cell per sample by ArchR[35]. Note that this does not necessarily preclude doublets of the same cell type.

## RNA cell QC

Multimodal snRNA-seq cells had to pass Cell Ranger ARC cell filtering and have at least 500 genes and <20% of mitochondrial reads. The Cell Ranger ARC filtered genes x cells matrix was subsetted to only these cells passing cell QC.

## RNA clustering

To cluster genes x cells matrices, we did: log normalization to 10,000 reads using Seurat::NormalizeData[131], most variable gene feature selection using a variance stabilizing transformation (VST)[131], center/scale features to mean 0 and variance 1 across cells using base::scale, PCA dimensionality reduction using irlba::prcomp_irlba, batch correction by sample via Harmony::HarmonyMatrix[27], shared nearest neighbor creation using RANN::nn2 and Seurat::ComputeSNN[131], Louvain clustering using Seurat::RunModulatrityClustering[131], and cluster visualization using UMAP coordinates via umap::umap. We used 20 PCs for the broad cell type clustering and 10 PCs for the sorted RA PBMC mRNA clustering since there was less variation within a cell type.

For cluster identification, we used marker genes seen in Supplementary Fig. 2l, m for the broad cell types and in panel b of Supplementary Figs. 3–7 for the chromatin classes.

## RNA doublet cluster removal

After doing an initial round of RNA clustering on the post cell QC cells, we removed doublet clusters with multiple cell-type-specific genes, intermediate placement between broad cell type clusters in PC space, high UMI counts, and high doublet scores determined per cell per sample by Scrublet[132]. Note that this does not necessarily preclude doublets of the same cell type.

## Symphony classification of transcriptional identity

To determine the RA transcriptional cell types/states within our multimodal data, we used Symphony[95] to map the multimodal snRNA-seq profiles into the AMP-RA reference synovial tissue transcriptional cell types/states[14] (Supplementary Fig. 1b, d). We used one Symphony reference object from that study for the broad cell types together and one for each broad cell type we tested (T cell, stromal, myeloid, B/plasma, and endothelial) for the fine-grain cell state identities. The broad cell types and lymphocyte states were defined using both gene and surface protein expression while the others were defined using gene expression only. In each case, we mapped the multimodal snRNA-seq gene x cells matrix into the appropriate Symphony reference object using the mapQuery function, accounting for sample as a batch variable. Using the knnPredict function with $k = 5$, each multiome cell was classified into a reference transcriptional cell type/state by the most common annotation of its five nearest AMP-RA reference neighbors in the harmonized embedding. We considered it a high confidence mapping if at least 3 out of the 5 nearest reference neighbors were the same cell type/state, though the number of cell types/states will affect this as more cell types/states means more boundary regions between cell types/states.

## Broad cell type clustering

For non-doublet cells passing cell QC, we subsetted the feature x cells matrices and performed broad cell type clustering within modalities as described above in "ATAC clustering" for the unimodal scATAC-seq and multimodal snATAC-seq datasets separately and "RNA clustering" for the multimodal snRNA-seq datasets (Supplementary Fig. 1a, b). We also classified the multimodal snRNA-seq cells into the AMP-RA CITE-seq study[14] broad cell types using Symphony[95] (see "Symphony classification of transcriptional identity"). The small minority of cells (2%) with discordant cell types defined in the snATAC-, snRNA-, and CITE-seq modalities for the multiome datasets were removed (Supplementary Fig. 1b). Here, as in all analyses unless otherwise stated, we included OA samples to increase cell counts, but we did not make any OA versus RA comparisons due to low power.

## Fine-grain chromatin class clustering

To define chromatin classes within broad cell types (Supplementary Fig. 1c), we made peaks x cells matrices for each broad cell type concatenating unimodal scATAC-seq and multimodal snATAC-seq cells of that type across the consensus peaks. Since peaks were called on all unimodal scATAC-seq cells regardless of cell type, we first subset each consensus peaks x broad cell type cells matrix by "peaks with minimal accessibility" (PMA). We defined minimal accessibility as consensus peaks that had a fragment in at least 0.5% of cells of that type, except for endothelial cells which we increased to a minimum of 50 cells. After subsetting the matrix by PMA peaks, we ran the same clustering pipeline detailed in "ATAC clustering". For endothelial cells, due to small cell counts, we batch-corrected on both sample and assay and updated Harmony's sigma parameter to 0.2. We did another round of QC to exclude cells that clustered primarily due to relatively fewer total fragments per cell and fewer peaks with at least one 1 fragment per cell,

and then re-clustered. We tried a number of clustering resolutions (see Supplementary Fig. 13 for a subset) and chose the resolution at which known cell-state-specific gene markers' promoter peak chromatin accessibility and gene expression largely respected cluster boundaries, such as *PRF1* in $T_A$−4: CD4+ PRF1+ cytotoxic (Fig. 2b) or *SPP1* in $M_A$−4: SPP1+ FABP5+ intermediate (Fig. 4b).

To label chromatin classes, we used the first letter of the broad cell types (T - T cell; S - stromal; M - myeloid; B - B/plasma; E - endothelial), a subscript A for accessibility, a cluster number (ordered by number of cells, with the biggest cluster named 0). To give biological context, we took advantage of both the peak accessibility and gene expression profiles. We chose a class's markers based on a number of factors: (1) the class-specificity of the marker gene's expression, (2) the class-specificity of the marker peak associated to that gene's promoter, (3) previous reports of that gene as a cell type marker in the literature, and (4) corroboration with our well-annotated AMP-RA tissue CITE-seq dataset[14] via reference mapping[95] (Figs. 2–6b, 7a–c; Supplementary Figs. 1d, 3–7b, 10g, h; Supplementary Data 3, 4). We proposed a cell identity based on known markers in the field; for example, *PDCD1* and *CXCL13* in TFH/TPH[12] or *PRG4* and *CD55* in lining fibroblasts[21]. We further supported the proposed identity by the correspondence to the transcriptional cell state annotation from our well-annotated AMP-RA reference of synovial tissue CITE-seq data[14] (Fig. 7a–c; Supplementary Fig. 10g, h; Supplementary Data 3).

## T cell lineage analysis

We used a logistic regression model to investigate how promoter peaks align with the CD4 and CD8 lineage distinction ('lineage') across T cells beyond their chromatin class identity ('class'), sample identity ('sample'), and overall fragment counts ('nFragments'). The lineage variable was defined as the cell's chromatin accessibility at the promoter peaks of: CD4+ CD8A- (+1), CD4+ CD8A+ or CD4− CD8A− (0), CD4− CD8A+ (−1); cell counts by lineage and class are in Supplementary Table 3. A plus sign (+) signifies that the CD4 or CD8 lineage promoter peak is accessible while a minus sign (−) signifies that it is not. Genome-wide T cell promoter peaks were defined as those T cell PMA peaks that overlapped an ENCODE promoter-like cCRE[25], whose proposed target gene was assessed via overlapping ENSEMBL[133] hg38 release 92 transcript annotations. We note that if there were multiple overlapping transcripts, we selected one gene to annotate the cCREs by excluding lincRNA, miRNA, antisense genes, orfs, and other pseudogenes then selecting one of the remaining genes. We excluded peaks that were uniformly positive or negative after binarizing. For each of these binarized promoter peaks ('peak'), we calculated two logistic regressions using lme4::glmer[134] with a nloptwrap optimizer for speed:

$$\text{Full model} : \text{peak} \sim \text{lineage} + \text{class} + (1|\text{sample}) + \text{scale}(\log 10(\text{nFragments}))$$

$$\text{Null model} : \text{peak} \sim \text{class} + (1|\text{sample}) + \text{scale}(\log 10(\text{nFragments}))$$

A lineage beta in the model is positive if the peak is associated to CD4 and negative if associated to CD8. We calculated significance as a likelihood ratio test (LRT) between the full and null models with multiple hypothesis test correction using FDR < 0.20; significant results are shown in Supplementary Data 1. Furthermore, we defined a lineage score per cell via: (1) subsetting the normalized chromatin accessibility matrix by the lineage-significant peaks; (2) dividing CD4-associated peaks by the number of CD4-associated peaks to normalize; (3) dividing CD8A-associated peaks by the number of CD8A-associated peaks to normalize; (4) multiplying CD8A-associated peaks by −1 to differentiate lineage; (5) summing over peaks by cell to get a cell score. Thus, if a cell's lineage score is positive, that cell is more associated

with CD4 and CD8 if otherwise. We aggregated these cell scores by chromatin class in Supplementary Fig. 3d.

## TF motif analysis

We used ArchR[35] version 1.0.2 for our TF motif analysis. For each cell type's final QC cells, we subsetted each sample's fragments using awk[135], bgzip[136], and tabix[137] before creating arrow files from them using createArrowFiles with all additional QC flags nullified. ArchR removed samples with two or fewer cells, so one sample with only two B/plasma cells was removed in that cell type. From the arrow files, we created an ArchR project via ArchRProject. We added our peak set into the project by addPeakSet and recreated a peaks by cells matrix via addPeakMatrix. We added our chromatin classes to the project's cell metadata with addCellColData. Then, we added motif annotations to our peaks using addMotifAnnotations with the JASPAR2020 motif set version 2, a 4 bp motif search window width, and motif *p* value of 5e-05. We added chromVAR background peaks via addBgdPeaks and then calculated chromVAR deviations using addDeviationsMatrix. Next, we found class-specific peaks for each chromatin class using getMarkerFeatures via a Wilcoxon test and accounting for TSS Enrichment and log10(nFragments). Within those peaks, we found motif enrichment via peakAnnoEnrichment with cutoffs FDR ≤ 0.1 and Log2FC ≥ 0.5. We modeled our heatmap of motif enrichment on plotEnrichHeatmap, but we added some filters. As in the default plotEnrichHeatmap method, we used the −log10(*p*adj), where the *p* value is calculated via a hypergeometric test, as the motif enrichment value. For each chromatin class sorted by maximum motif enrichment value, we chose the top motifs not already chosen that had at least an enrichment value of 5 for that class, had the maximal or within 95% of the maximal enrichment for that class, and whose corresponding TF had at least 0.05 mean-aggregated normalized gene expression for that class. For myeloid cells, the enrichment cutoff was set to 2 to show some motifs for $M_A$−0. In endothelial cells, there were so few $E_A$−3 cells that only 1 class-specific peak was called, resulting in no useful motif information to be shown; we also added a SOX17 motif (JASPAR[109] ID MA0078.1), a prominent arteriolar endothelial TF[86], to the JASPAR2020 motif set for endothelial cells. For the chosen motifs, we plotted the percentage of the max enrichment value across classes with the max value in parentheses in the motif label as in plotEnrichHeatmap.

For the TFs associated with the top class-specific accessible motifs, we used a one-sided Wilcoxon test to compare the normalized gene expression for the TF between cells in that chromatin class and the other cells within that cell type, with the alternative hypothesis being "greater" and multiple hypothesis test correction within cell types using FDR (Supplementary Data 2).

## Loci visualization

To visualize the chromatin accessibility read buildups by chromatin class or transcriptional cell state (class/state), we first subsetted the deduplicated BAM files for each sample by the cells in the specific state/class using an awk[135] command looking for the samtools CB:Z (i.e., cell barcode) flag; a BAM index file was made for each BAM file for region subsetting purposes later. Then for each class/state at each locus, we subsetted each sample's BAM file for that region using samtools view, merged the BAM files across samples using samtools merge, converted the BAM files to bedgraph files using bedtools[129] genomecov, and then divided the bedgraph counts by the total read count (by 1e7 reads) in that class/state to allow for comparison between classes/states. The bedgraph files were then imported to IGV[138] and the data range for each class/state was set to the maximum value across classes/states. Tracks were colored by their class/state. We did not always show all classes/states for space reasons, but we picked representatives that were similar in the locus shown. Peaks (see "ATAC peak calling"), motifs (see "TF motif analysis"), and SNPs (see "Genetic variant analysis") were imported into IGV as BED files. We

could not label all motifs found in these loci for space reasons, so we picked the enriched motif we were highlighting and a few other enriched motifs. We also could not always show all the gene isoforms for all loci for space reasons, but we did always show a representative isoform for those that looked similar in the locus shown.

### Stromal DNA methylation analysis

We downloaded 1859 DM loci for RA versus OA synovial fibroblast cell lines from Nakano et al., 2013[47]. We converted the 1 bp DM regions from hg19 to hg38 reference genomes using liftOver[139]; 1 region did not map. Next, we overlapped these DM loci with our 200 bp stromal PMA peaks using intersectBed[129] to get 152 DM loci, with 67 associated eith hypermethylation and 85 to hypomethylation. We defined a per-cell score as in the "T cell lineage analysis" section, but with positive scores corresponding to hypermethylation and negative scores to hypomethylation. We calculated a one-sided Wilcoxon test $p$ value of DNA methylation cell scores between the 11,733 cells in $S_A-0$ and the 12,574 stromal cells not in $S_A-0$ to get significance of $S_A-0$ enrichment for hypomethylated regions.

We used the genes assigned to the DM loci from the original paper[47]. For the genes related to hypermethylated DM and hypomethylated DM accessible loci separately, we plotted their scaled mean normalized gene expression within fibroblast classes $S_A-0$, $S_A-1$, and $S_A-2$ to assess fibroblast class preferences.

### Cultured fibroblast datasets

We obtained two cultured unstimulated FLS multiome datasets from Smith et al.[44]. We downloaded their genes x cells matrices from Immport accession ID SDY2213 and fragment files from the authors. We subset these files by their QCed cells found in Immport file adata_scatac_chromVAR_motif_cultured.968213.h5; there were 19,573 QC cells across the two samples. We overlapped this subsetted fragment file by our peaks to create a peaks x cells matrix. We saw good overlap in that matrix with 99.99% of our peaks having at least 1 cell represented and all cells having overlapping fragments with at least a few hundred peaks. For both gene and peak matrices individually, we concatenated the two samples and normalized as above.

### Fibroblast identity analysis

We subsetted our stromal tissue datasets to only include fibroblast populations ($S_A-0$, $S_A-1$, $S_A-2$). We calculated differentially expressed genes between tissue lining ($S_A-1$) and sublining ($S_A-0$, $S_A-2$) populations in the normalized gene expression matrix using presto::wilcoxauc and adjusted $p$ values using FDR. We created gene sets of 382 lining and 254 sublining genes using the cutoffs: FDR < 0.1, logFC > 0.25, and AUC > 0.6. We then calculated a per-cell score as in the "T cell lineage analysis" section, but with positive scores corresponding to lining fibroblasts and negative scores to sublining fibroblasts. Using the tissue-defined gene sets, we calculated this per-cell fibroblast identity gene score in the normalized cultured fibroblast gene expression matrix (see "Cultured fibroblast datasets"). We used a two-sided Wilcoxon test of fibroblast identity gene scores between all pairs of fibroblast sources to determine significance via ggpubr::compare_means. We did the same analysis with differentially accessible peaks in the normalized chromatin accessibility matrix using cutoffs FDR < 0.1, logFC > 0.1, and AUC > 0.58 to get 248 lining peaks and 294 sublining peaks.

### Tissue and blood analysis

We downloaded a publicly available 10x Single Cell Multiome ATAC + Gene Expression dataset[92] of healthy donor (female, age 25) PBMCs with granulocytes removed through cell sorting as part of our sister study[93] ('Public PBMC' dataset). The PBMC cell labels were generated using the processing defined in that study. No further quality control was done on the fragment file downloaded from the 10x website

(https://cf.10xgenomics.com/samples/cell-arc/2.0.0/pbmc_granulocyte_sorted_10k/pbmc_granulocyte_sorted_10k_atac_fragments.tsv.gz). For each cell type (B, T, and myeloid), we subset the fragment file by that cell type's cells and then overlapped them with our peaks to get a peaks x cells matrix as done in "ATAC quality control". We concatenated this matrix to our RA tissue's peaks x cells matrix for each corresponding cell type and then re-clustered using the same PMA and variable peaks chosen for tissue and harmonizing by sample. We chose the resolution that best mirrored the RA tissue chromatin classes. The odds ratio for each individual biological source's cell label and the combined tissue and blood cluster label was calculated as in "Class/state odds ratio". We replicated this analysis using the RA PBMCs for TFH/TPH and Treg FACS populations and the 5 RA tissue chromatin classes.

### Class/state odds ratio

For each combination of chromatin class and transcriptional cell state within a cell type, we constructed a $2 \times 2$ contingency table of the number of cells belonging or not to the class and/or state. For cell states that had >10 cells, we then calculated the odds ratio (OR) and $p$ value via stats::fisher.test. We did multiple hypothesis test correction via stats::p.adjust using FDR < 0.05. We displayed the natural log of the OR via base::log, and if the value was infinite, we capped it at 1 plus the ceiling of the non-infinite max absolute value of logged OR for display purposes; negative infinity was the negative capped number. All the ORs and $p$ values for all class/state combinations from Fig. 7a–c and Supplementary Fig. 10g, h are in Supplementary Data 3.

We defined the accuracy of the class/state correspondence as the percentage of multiome cells with perfect mapping (i.e., all 5 nearest neighbors in the reference had the same cell state) within each group of 'concordant' (i.e., cells whose class and state agreed as determined by the odds ratio) or 'discordant' (i.e., cells whose class and state disagreed) cells per cell type. For example, cells mapping to class $T_A-0$: CD8A+ GZMK+ and state T-14: CD8+ GZMK+ memory would be 'concordant' cells while cells mapping to class $T_A-2$: CD4+ PD-1+ TFH/TPH and state T-14: CD8+ GZMK+ memory would be 'discordant' cells.

### ATAC pseudobulk differential peak analysis

For T, stromal, and myeloid cell types, we summed the non-binary peaks x cells matrix by sample and transcriptional cell state combinations across cells. We subset the summed matrix to include only samples with more than 150 cells, states with more than 130 cells, and combinations with more than 10 cells. For the within-class analysis, we split the matrix by the transcriptional cell states that belonged to the same chromatin class (e.g., 5 T cell matrices); we excluded any class with only 1 state passing our QC thresholds. We also kept the full matrix per cell type for the across-classes analysis. We subset peaks by each cell type's promoter PMA peaks (see "T cell lineage analysis") that had at least 5 reads across the pseudobulks within that analysis. For each peak for each set of states (either within or across classes), we calculated two negative binomial models of that peak's sample/state pseudobulk distribution using MASS::glm.nb, accounting for covariates of sample identity ('sample') and the number of fragments ('nFragments') in the sample and cell state combination and differing by the inclusion of transcriptional cell state ('cell state'):

$$\text{Full model} : \text{peak} \sim \text{cell state} + \text{sample} + \text{scale}(\log 10(\text{nFragments}))$$

$$\text{Null model} : \text{peak} \sim \text{sample} + \text{scale}(\log 10(\text{nFragments}))$$

Cell state and sample were represented by 1-hot encoded matrices. We calculated an ANOVA log-likelihood ratio test (LRT) $p$ value between these two models and reconciled multiple hypothesis test

correction within each analysis separately via FDR. Peaks were considered differential if they had FDR < 0.10.

## Linear discriminant analysis

We used LDA to determine how well knowing the chromatin harmonized principal component (hPC) information helped predict the mRNA fine-grain cell states for each pairwise combination of states. We specifically use pairwise combinations instead of 1 versus all comparisons to assess the chromatin accessibility data's ability to give rise to one or multiple transcriptional cell states. For each pair of transcriptional cell states within a broad cell type, we subset all data structures by those cells and remade the cell state vector into a 1-hot encoding. If either cell state of the pair had <50 cells, we excluded it from further analysis. We used the 10 chromatin hPCs from the fine-grain chromatin class clustering (see "Fine-grain chromatin class clustering"). Covariates of sample (1-hot encoded for 12 samples) and scaled logged number of fragments (nFragments) were used since both can affect cell type identity. We trained an LDA model using MASS::lda on 75% of cells across the pair of states, verifying that the training and testing sets had cells from both states:

$$\text{LDA model} : \text{cell state} \sim \text{chromatin hPCs} + \text{sample}$$
$$+ \text{scale}(\log 10(\text{nFragments}))$$

We tested the model using stats::predict for the 25% of held-out data and quantified the discriminative value of the model using an area under the curve AUC metric from ROCR[140] library functions ROCR::prediction and ROCR::performance. Pairs of distinct clusters were only calculated once; the square matrices of results have the triangles mirrored. If the cell states were the same and a model was not run (identity line) or the model between pairs of clusters had a constant variable due to samples with too few cells (non-identity line), the box is greyed out.

## Superstate FACS protocol

From pooled PBMC samples from 4 RA patients, we enriched for CD4 T cells using the MACS protocol and sorted for 4 populations using FACS (CD4$^+$CD127$^-$CD25$^{hi}$ Tregs, CD4$^+$CD127$^-$CD25$^{int}$ Tregs, CD4$^+$CD25$^-$PD1$^+$CXCR5$^+$ TFH, and CD4$^+$CD25$^-$PD1$^+$CXCR5$^-$ TPH). FACS sequential gating plots can be found in Supplementary Fig. 15a. We used the following antibodies: CD3-FITC, CD4-BV421, CD25-PE-Cy7, CD127-BV650, CXCR5-PE, PD1-APC. All antibodies were purchased by BioLegend and used at one microliter per million cells. The Live/Dead dye 7-AAD was purchased from ThermoFisher Scientific and used at five microliters per million cells. After nuclei isolation, each sorted population was tagged with a nuclear hashing antibody before pooling across populations. Total-Seq™-A hashtag antibodies (A0451-A0454) were purchased from BioLegend and used at a 1:40 dilution.

## Superstate multiome experimental protocol

We performed a multiome experiment as described in "Multiome experimental protocol", with the additional step of producing cDNA from Hashtag oligos (for Protein Antibody Hashtags) during GEM incubation, generating the Hashtag library alongside the Gene Expression library. The Hashtag library was sequenced at approximately five thousand reads per cell.

## Superstate multiome quality control

Quality control steps for the superstate multiome experiment were the same as the RA tissue multiome experiments, up to and not including the doublet step in both modalities (Supplementary Fig. 1b). To better account for doublets between these very similar cell states, we only included cells with a single identity determined by running Seurat::HTODemux[131] on the normalized hashtag library. Those cell state identities were strictly used as a label. Cells needed to pass QC in all three modalities to be included in the downstream analysis. We kept 402 CD4$^+$CD127$^-$CD25$^{hi}$ Tregs, 1690 CD4$^+$CD127$^-$CD25$^{int}$ Tregs, 535 CD4$^+$CD25$^-$PD1$^+$CXCR5$^+$ TFH, and 371 CD4$^+$CD25$^-$PD1$^+$CXCR5$^-$ TPH cells.

## Single-cell differential peak analysis

We used a logistic regression model to determine differential promoter peaks across chromatin class identity. We did this at the single cell level for the combined unimodal scATAC-seq and multimodal snATAC-seq cells and took into account the sample's sample ('sample') and overall fragment counts ('nFragments') as covariates. Genome-wide promoter peaks were defined per cell type as in "T cell lineage analysis". For each peak and class combination, we calculated two logistic regressions using lme4::glmer[134] with a nloptwrap optimizer for speed:

$$\text{Full model} : \text{peak} \sim \text{class} + (1|\text{sample}) + \text{scale}(\log 10(\text{nFragments}))$$

$$\text{Null model} : \text{peak} \sim (1|\text{sample}) + \text{scale}(\log 10(\text{nFragments}))$$

The log2FC was determined as the cell type beta. We calculated significance as a LRT between the full and null models with multiple hypothesis test corrections using FDR. The top 5 peaks per class, defined as having log2FC > 0.5 and −log10(FDR) > 5, ordered by FDR, are shown in Supplementary Data 4.

## Single-cell differential gene analysis

For the multiome cells only, we calculated differentially expressed genes between chromatin class identities within a cell type via a two-sided Wilcoxon test using a normalized gene expression matrix input to presto::wilcoxauc. The top 5 genes per class, defined as having logFC > 0.5 and −log10(FDR) > 5, ordered by FDR and logFC, are shown in Supplementary Data 4. We selected one peak of potentially multiple that overlapped the annotated gene based on the differential peak's significance in the corresponding class.

## TFH/TPH/Treg differential feature analysis

For the sorted RA PBMCs, we determined differential genes and peaks between each pair of states within one chromatin class: (1) CD4$^+$CD127$^-$CD25$^{hi}$ Tregs and CD4$^+$CD127$^-$CD25$^{int}$ Tregs; (2) CD4$^+$CD25$^-$PD1$^+$CXCR5$^+$ TFH and CD4$^+$CD25$^-$PD1$^+$CXCR5$^-$ TPH. We calculated differential genes as in "Single-cell differential gene analysis". Differential promoter peaks were calculated similarly to "Single-cell differential peak analysis", but we excluded sample as a covariate since there was a single pooled RA PBMC sample and used stats::glm instead of lme4::glmer since we removed the random effect of sample, thus negating the need for a mixed effect model. If a gene had multiple promoter peaks, we chose the peak with the max normalized peak accessibility summed across cells in that pair of states. Furthermore, we only included peak/gene pairs with at least 1 fragment/UMI in greater than 50 cells in that pair of states. We corrected $p$ values using FDR separately within modalities.

## Symphony classification of chromatin class

To utilize the richer clinical information in the more abundant AMP-RA reference datasets, we classified each AMP-RA reference cell into a chromatin class. We used the same shared transcriptional spaces by cell type defined in "Symphony classification of transcriptional identity", but we reversed the reference and query objects in the knnPredict function, such that the multiome cells were in the 'reference' and the AMP-RA reference cells were in the 'query'. We used the most common annotation of the 5 nearest multiome neighbors to classify the chromatin class in the AMP-RA reference cells. We averaged the 5 nearest multiome neighbors' UMAP dimensions to visualize the

classified chromatin classes in the AMP-RA reference cells on the chromatin class UMAPs.

### Unimodal scATAC-seq and AMP-RA CITE-seq shared donor analysis

There were different samples that came from the same donors in the unimodal scATAC-seq and AMP-RA reference CITE-seq datasets. We expected similar, but not the same, chromatin class proportions for samples coming from the same donor's tissue but put through different experimental protocols and class assignment methods. First, we filtered out any donors that did not have at least 200 scATAC-seq or CITE-seq cells in all cell types except endothelial, in which we lowered the threshold to 100 cells. We then calculated the proportion of each sample's cells coming from each chromatin class for each technology and plotted the CITE-seq proportion by scATAC-seq proportion for each donor, faceted by chromatin class in Fig. 8a and Supplementary Fig. 16a. We calculated the Pearson correlation and two-sided $p$ value for each chromatin class by stats::cor.test.

### Co-varying neighborhood analysis

We used the significant CNA[99] correlations between AMP-RA reference cell neighborhoods and sample-level covariates from our AMP-RA reference study[14]. We re-plotted the AMP-RA reference cell CNA correlations on the chromatin class UMAPs and re-aggregated them by classified chromatin class calculated in "Symphony classification of chromatin class". In Supplementary Table 6, clinical metrics and CTAPs were listed if the median abundance correlation of the AMP-RA reference cells within their Symphony-classified chromatin class was more extreme than the FDR threshold for that patient attribute[14]. Classes were considered significantly expanded if that class's cells were positively correlated with that attribute's per-sample class abundance within a cell type and depleted if negatively correlated.

### Genetic variant analysis

We used the set of RA-associated non-coding SNP locations and statistically fine-mapped PIPs from our previously published RA multi-ancestry genome-wide association meta-analysis study[107]. We subsetted the SNPs by PIP > 0.1 and overlapped their locations with our 200 bp trimmed peaks using intersectBed[129]. For the overlapping peaks, we plotted their normalized chromatin accessibility mean-aggregated by chromatin class and scaled in Fig. 8d with more description in Supplementary Table 5. To determine broad cell type specificity of a peak's accessibility, we calculated a Wilcoxon test one-sided "greater" $p$ value between the normalized, mean aggregated, scaled peak accessibility in the broad cell type's classes versus those classes in the other broad cell types. Classes were considered accessible for that peak if the scaled mean normalized peak accessibility over 24 classes and 11 peaks, z, >1. We plotted example loci in Fig. 8e and Supplementary Fig. 17 as described in "Loci visualization"; we excluded some chromatin classes for space, but we kept the most accessible chromatin classes and at least one chromatin class from each cell type at each locus. The TF motif logos in Fig. 8e and Supplementary Fig. 17 were downloaded from the JASPAR motif database[109] for accession IDs MA0517.1 (STAT1::STAT2), MA0039.4 (KLF4), and MA1483.1 (ELF2); they were not to scale, but the motif position the SNP disrupts was aligned to the SNP. We further aggregated multimodal snATAC-seq reads by transcriptional cell state for visualization purposes in Supplementary Fig. 18.

### Computational versions used

Specific software versions are listed here, but more information about how they were used within this study can be found in the appropriate Methods sections.

Flow cytometry data was analyzed using FlowJo (v10.7.2 for tissue samples and v10.8.1 for blood samples).

We used R v3.6.1 for most analyses with the following packages: argparse v2.0.3, aricode v1.0.0, BiocGenerics v0.30.0, class v7.3-17, data.table v1.12.8, dplyr v1.0.2, GenomeInfoDb v1.20.0, GenomicRanges v1.36.1, ggbeeswarm v0.6.0, ggplot2 v3.3.0, ggpubr v0.4.0, ggrastr v0.2.3, ggrepel v0.8.2, ggthemes v4.2.0, gplots v3.0.1.1, gridExtra v2.3, gtools v3.8.2, harmony v1.0, IRanges v2.18.3, irlba v2.3.3, lattice v0.20-41, lme4 v1.1-21, magrittr v1.5, MASS v7.3-51.6, Matrix v1.2-18, Matrix.utils v0.9.7, matrixStats v0.56.0, patchwork v1.1.0.9000, pheatmap v1.0.12, plyr v1.8.6, presto v1.0.0, RANN v2.6.1, RColorBrewer v1.1-2, rcompanion v2.4.1, Rcpp v1.0.4.6, RcppCNPy v0.2.10, repr v1.0.1, reticulate v1.13, Rmisc v1.5.1, ROCR v1.0-7, rstatix v0.7.0, S4Vectors v0.22.1, scales v1.1.1, Seurat v3.2.0, Signac v1.1.0, stringr v1.4.0, symphony v1.0, tibble v3.0.1, tidyr v1.0.3, umap v0.2.3.1, uwot v0.1.8, viridis v0.5.1, viridisLite v0.3.0.

For ArchR analyses, we used R v4.2.0 with the following packages: ArchR v1.0.2, argparse v2.1.6, Biobase v2.56.0, BiocGenerics v0.42.0, Biostrings v2.64.1, BSgenome v1.64.0, BSgenome.Hsapiens.UCSC.hg38 v1.4.4, chromVARmotifs v0.2.0, data.table v1.14.4, GenomeInfoDb v1.32.4, GenomicRanges v1.48.0, ggplot2 v3.3.6, gridExtra v2.3, gtable v0.3.1, gtools v3.9.3, IRanges v2.30.1, JASPAR2016 v1.24.0, JASPAR2018 v1.1.1, JASPAR2020 v0.99.10, magrittr v2.0.3, Matrix v1.5-1, MatrixGenerics v1.8.1, matrixStats v0.62.0, plyr v1.8.7, Rcpp v1.0.9, rhdf5 v2.40.0, rtracklayer v1.56.1, S4Vectors v0.34.0, stringr v1.4.1, SummarizedExperiment v1.26.1, TFBSTools v1.34.0, tidyr v1.2.1, XVector v0.36.0.

We also used python v3.7.3, scrublet v0.2.3, samtools v1.9, bedtools v2.28.0, bedops v2.4.36, GNU Awk 3.1.7, jupyter v4.4.0.

## Data availability

The raw FASTQs files generated in this study have been deposited in the dbGaP database under accession code phs003417.v2.p1. These data are available under restricted access as patient-identifiable data; access can be requested from dbGaP. The processed data files generated in this study have been deposited in Synapse under accession code syn53650034[141]. Source data are provided with this paper. Symphony references from ref. 14 are available in Synapse under accession code syn52297840[142]. Cultured unstimulated FLS multiome datasets from ref. 44 are available in Immport accession ID SDY2213. JASPAR motifs from ref. 109 are available in JASPAR under accession codes MA0517.1, MA0039.4, MA1483.1, and MA0078.1.

## Code availability

The code used to generate the results presented herein can be found on GitHub (https://github.com/immunogenomics/RA_ATAC_multiome/).

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

## Acknowledgements

This work was supported by the Accelerating Medicines Partnership (AMP) in Rheumatoid Arthritis and Lupus Network. AMP is a public-private partnership (AbbVie Inc., Arthritis Foundation, Bristol-Myers Squibb Company, Foundation for the National Institutes of Health, GlaxoSmithKline, Janssen Research and Development, LLC, Lupus Foundation of America, Lupus Research Alliance, Merck Sharp & Dohme Corp., National Institute of Allergy and Infectious Diseases, National Institute of Arthritis and Musculoskeletal and Skin Diseases, Pfizer Inc., Rheumatology Research Foundation, Sanofi and Takeda Pharmaceuticals International, Inc.) created to develop new ways of identifying and validating promising biological targets for diagnostics and drug development. Funding was provided through grants from the National Institutes of Health (UH2-AR067676, UH2-AR067677, UH2-AR067679, UH2-AR067681, UH2-AR067685, UH2-AR067690, UH2-AR067691, UH2-AR067694, and UM2-AR067678). Accelerating Medicines Partnership and AMP are registered service marks of the U.S. Department of Health and Human Services. This work is supported in part by funding from the National Institutes of Health (1UH2AR067677-01, U01HG009379, UC2AR081023). We also acknowledge support by NIH NHGRI T32HG002295 and NIAMS T32AR007530 (to K. Weinand and A.N.); Arthritis National Research Foundation (to F.Z.); NIAID T32AR007258 (to K.S.); and NIH NIAMS R01AR078769 (to D.A.R.). S.S. was in part supported by the Uehara Memorial Foundation and the Osamu Hayaishi Memorial Scholarship. K. Wei is supported by NIH-NIAMS K08AR077037, Burroughs Wellcome Fund Career Awards for Medical Scientists, a Doris Duke Charitable Foundation Clinical Scientist Development Award, and a Rheumatology Research Foundation Innovative Research Award. UK Birmingham is supported by the Versus Arthritis Research Into Inflammatory Arthritis Centre Versus Arthritis (Versus Arthritis grant 22072) and the EU Innovative Medicines Initiative RT CURE. We wish to thank Tiffany Amariuta, Kaitlyn A. Lagattuta, Anika Gupta, Angela Zou, Miles Tran, and Nicholas Sugiarto for the helpful discussion.

## Author contributions

K. Weinand, S.S., and S.R. conceptualized the study. K. Weinand conducted all computational analyses. S.S., A.N., F.Z., and S.R. provided input on statistical analyses and study design. S.S., A.N., A.H.J., D.A.R., J.H.A., M.B.B., K. Wei, and S.R. provided input on cellular analyses and interpretation. S.S. and S.R. supervised the study. AMP RA/SLE Network recruited patients and obtained synovial biopsies for unimodal scATAC-seq samples. L.T.D. and K. Wei recruited patients and obtained synovial biopsies for multimodal samples. K. Wei, A.H.J, G.F.M.W., A.N., and M.B.B. designed and implemented the tissue disaggregation, cell sorting, and single-cell sequencing pipeline. A.H.J., K. Wei, and G.F.M.W supervised and executed the tissue disaggregation pipeline for unimodal scATAC-seq samples. K. Wei, G.F.M.W, and Z.Z. supervised and executed the tissue disaggregation pipeline for multimodal tissue samples. K. Weinand, S.S., G.F.M.W., M.A.S., D.A.R., K. Wei, and S.R. designed the RA PBMC multimodal experiment, L.T.D. provided the samples, and M.A.S. executed the experiment. K. Weinand, S.S., and S.R. wrote the initial manuscript. All authors contributed to editing the final manuscript.

## Competing interests

S.R. is a founder for Mestag Therapeutics, a scientific advisor for Janssen, Sonoma, and Pfizer. D.A.R. reports personal fees from Pfizer, Janssen, Merck, GlaxoSmithKline, AstraZeneca, Scipher Medicine, HiFiBio, and Bristol-Myers Squibb, and grant support from Merck, Janssen, and Bristol-Myers Squibb outside the submitted work. D.A.R. is a co-inventor on the patent for TPH cells as a biomarker of autoimmunity. The remaining authors declare no competing interests.

## Additional information

## Accelerating Medicines Partnership Program: Rheumatoid Arthritis and Systemic Lupus Erythematosus (AMP RA/SLE) Network

Jennifer Albrecht[7], William Apruzzese[11], Nirmal Banda[12], Jennifer L. Barnas[7], Joan M. Bathon[13], Ami Ben-Artzi[14], Brendan F. Boyce[15], David L. Boyle[16], S. Louis Bridges Jr.[8,9], Vivian P. Bykerk[8,9], Debbie Campbell[7], Hayley L. Carr[17],

Arnold Ceponis[16], Adam Chicoine[1], Andrew Cordle[18], Michelle Curtis[1,2,3,4,5], Kevin D. Deane[12], Edward DiCarlo[19], Patrick Dunn[20,21], Andrew Filer[17], Gary S. Firestein[16], Lindsy Forbess[17], Laura Geraldino-Pardilla[13], Susan M. Goodman[8,9], Ellen M. Gravallese[1], Peter K. Gregersen[22], Joel M. Guthridge[23], Maria Gutierrez-Arcelus[1,2,3,4,5,24], Siddarth Gurajala[1,2,3,4,5], V. Michael Holers[12], Diane Horowitz[22], Laura B. Hughes[25], Kazuyoshi Ishigaki[1,2,3,4,5,26], Lionel B. Ivashkiv[8,9], Judith A. James[23], Joyce B. Kang[1,2,3,4,5], Gregory Keras[1], Ilya Korsunsky[1,2,3,4,5], Amit Lakhanpal[8,9], James A. Lederer[27], Zhihan J. Li[1], Yuhong Li[1], Katherine P. Liao[1,4], Arthur M. Mandelin II[28], Ian Mantel[8,9], Mark Maybury[17], Andrew McDavid[29], Joseph Mears[1,2,3,4,5], Nida Meednu[7], Nghia Millard[1,2,3,4,5], Larry W. Moreland[12,30], Alessandra Nerviani[31], Dana E. Orange[8,32], Harris Perlman[28], Costantino Pitzalis[31], Javier Rangel-Moreno[7], Karim Raza[17], Yakir Reshef[1,2,3,4,5], Christopher Ritchlin[7], Felice Rivellese[31], William H. Robinson[33], Laurie Rumker[1,2,3,4,5], Ilfita Sahbudin[17], Dagmar Scheel-Toellner[17], Jennifer A. Seifert[12], Kamil Slowikowski[4,5,34,35], Melanie H. Smith[8], Darren Tabechian[7], Paul J. Utz[33], Dana Weisenfeld[1], Michael H. Weisman[14,33] & Qian Xiao[1,2,3,4,5]

[11]Accelerating Medicines Partnership® Program: Rheumatoid Arthritis and Systemic Lupus Erythematosus (AMP® RA/SLE) Network, New York, NY, USA. [12]Division of Rheumatology, University of Colorado School of Medicine, Aurora, CO, USA. [13]Division of Rheumatology, Columbia University College of Physicians and Surgeons, New York, NY, USA. [14]Division of Rheumatology, Cedars-Sinai Medical Center, Los Angeles, CA, USA. [15]Department of Pathology and Laboratory Medicine, University of Rochester Medical Center, Rochester, NY, USA. [16]Division of Rheumatology, Allergy and Immunology, University of California, San Diego, La Jolla, CA, USA. [17]Rheumatology Research Group, Institute of Inflammation and Ageing, University of Birmingham, NIHR Birmingham Biomedical Research Center and Clinical Research Facility, University of Birmingham, Queen Elizabeth Hospital, Birmingham, UK. [18]Department of Radiology, University of Pittsburgh Medical Center, Pittsburgh, PA, USA. [19]Department of Pathology and Laboratory Medicine, Hospital for Special Surgery, New York, NY, USA. [20]Division of Allergy, Immunology, and Transplantation, National Institute of Allergy and Infectious Diseases, National Institutes of Health, Bethesda, MD, USA. [21]Northrop Grumman Health Solutions, Rockville, MD, USA. [22]Northwell, New Hyde Park, NY, Department of Medicine, Manhasset, NY, USA. [23]Department of Arthritis & Clinical Immunology, Oklahoma Medical Research Foundation, Oklahoma City, OK, USA. [24]Division of Immunology, Department of Pediatrics, Boston Children's Hospital and Harvard Medical School, Boston, MA, USA. [25]Division of Clinical Immunology and Rheumatology, Department of Medicine, University of Alabama at Birmingham, Birmingham, AL, USA. [26]Laboratory for Human Immunogenetics, RIKEN Center for Integrative Medical Sciences, Yokohama, Japan. [27]Department of Surgery, Brigham and Women's Hospital and Harvard Medical School, Boston, MA, USA. [28]Division of Rheumatology, Department of Medicine, Northwestern University Feinberg School of Medicine, Chicago, IL, USA. [29]Department of Biostatistics and Computational Biology, University of Rochester School of Medicine and Dentistry, Rochester, NY, USA. [30]Division of Rheumatology and Clinical Immunology, University of Pittsburgh School of Medicine, Pittsburgh, PA, USA. [31]Centre for Experimental Medicine & Rheumatology, William Harvey Research Institute, Queen Mary University of London, London, UK. [32]Laboratory of Molecular Neuro-Oncology, The Rockefeller University, New York, NY, USA. [33]Division of Immunology and Rheumatology, Institute for Immunity, Transplantation and Infection, Stanford University School of Medicine, Stanford, CA, USA. [34]Center for Immunology and Inflammatory Diseases, Department of Medicine, Massachusetts General Hospital (MGH), Boston, MA, USA. [35]MGH Cancer Center, Boston, MA, USA.

