## [Peer Review File · Nature Communications]

The chromatin landscape of pathogenic transcriptional cell states in rheumatoid arthritisREVIEWER COMMENTS

Reviewer #1 (expert in the epigenetics of rheumatoid arthritis):

Weinand and colleagues present data from an integrated single-cell RNA/ATAC sequencing approach to define cellular states in the synovium. The data is novel and of high scientific interest. The analysis is very well done and presented in a clear way. I have the following suggestions to improve the manuscript.

OA samples were included to increase cell counts, but the analysis would be cleaner if only RA samples were used. It would be best to add a sufficient number of OA samples to separate the data sets and make a valid comparison. Alternatively, at least the cell counts in the supplemental data showing the individual cell clusters should indicate which were OA samples and it should be mentioned how many cells are from OA samples. It is assumed that the T/B cell analysis mainly reflects RA states, whereas a more significant contribution from OA samples is expected in the stromal population.

In Suppl. Fig. 1e, one sample contained very few cells and did not reflect all synovial cells. Also, for cell counts for specific cell types, the different chromatin classes contained less than 100 cells in some samples (especially B/plasma cells and endothelial cells). Here, the analysis is determined by a few samples. Samples with so few cells/patients should be re-examined in detail and possibly removed. At least this should be mentioned as a limitation in the discussion.

The hypothesis of chromatin superstates of cell populations is very compelling. However, to match these cellular superstates with long-known subclasses of cells (e.g., in T cells), it would be helpful to perform the analysis approach from the other end (presorting known cell subclasses and then ATACseq), at least in one subclass. For example, there was good overlap between Tregs as defined by ATACseq in PBMC and in tissues. How do these chromatin profiles compare to a chromatin profile of a PBMC Treg population sorted using established protein markers?

Was the enrichment of transcription factor motifs in a particular chromatin state associated with increased transcription of these transcription factors in the superstates or their corresponding subclasses?

It seems that the correlation between chromatin class and marker gene expression is markedly different between different cell types (best in T cells, worst in myeloid and B cells). Possible explanations for these differences should be discussed.

The enrichment of DNA methylation changes in SA-0 is interesting. Were the genomic sites affected by changes in DNA methylation within the SA-0 cluster random or were they enriched for marker genes important for differentiating this cluster from the other fibroblast clusters? If the latter, one could speculate that changes in DNA methylation stabilize the phenotype of the SA-0 cluster or one of its subclusters.

Since changes in DNA methylation have been detected in cultured cells in fibroblasts, the question arises whether chromatin classes are maintained in cell culture even if the transcriptional profile is lost. Single-cell ATAC sequencing of cultured RA synovial fibroblasts should be performed to clarify the stability of chromatin classes independent of the synovial environment.

Caroline Ospelt

Reviewer #2 (expert in expert in computational biology and single-cell/single-nucleus transcriptomics/genomics):

The manuscript by Weinand et al. entitled "The Chromatin Landscape of Pathogenic Transcriptional

Cell States in Rheumatoid Arthritis” describes an evaluation of cells isolated from synovial tissue of RA patients using scATAC-seq and combined snATAC-seq and snRNA-seq assays to define discrete chromatin states, a comparison of these chromatin states with previously published single cell transcriptional analysis of synovial tissue and PBMCs from RA patients, and a correlation between these chromatin states and their marker genes and clinical metrics of RA, previously defined RA subtypes, and previously identified RA genetic risk variants. The computational methods used for QC and batch correction are appropriate and the data analysis approaches are state-of-the-art for the analysis and interpretation of these types of single cell datasets and their correlation with the clinical and other parameters. This is an excellent manuscript and the results reported will serve as a valuable reference about the cellular and molecular characteristics of this important autoimmune disease.

Several suggested edits would contribute to enhancing the information reported as a reference for RA.

1. An important outcome of these kinds of single cell studies is the definition of discrete cell types using the measure characteristics, in this case chromatin states. In order for information about these cell types to be reusable, it is necessary to assign the cell types with was some kind of unique cell type name/label. The authors use an interesting syntactical approach for naming cell types based on these chromatin characteristics – broad cell class, accessibility (A) subscript, sequential number, marker gene(s), previously-characterized cell class – e.g., TA-2: CD4+ PD-1+ TFH/TPH. From their combined analysis of chromatin promoter accessibility and gene expression, they note that both PD-1 and CTLA4 are expressed and their promoters are accessible. I’m guessing that their analysis reveals several genes with similar expression/accessibility characteristics for each cell type. The authors should clarify how/why a given gene (e.g., PD-1 alone) was selected for the cell type label. It appears that this is based on prior knowledge combined with the analysis results, but there may be some circumstances where a data driven approach would be better than relying on prior knowledge, which may be incomplete and biased.
2. They should also clarify how they connect the chromatin defined cell types with the previously-characterized cell class (e.g., TFH/TPH). Was this done manually? Could it be done statistically/computationally in such a way that captures some quantitative measure of confidence?
3. In the T cell section, 5 chromatin classes were defined. How do these relate to the naïve/central memory/effector memory T cell subsets paradigm?
4. In several of the figures, the authors show examples where the open chromatin state matches the gene expression data for the associated gene (e.g., Fig 2b). This raises the question of if the chromatin analysis has added any value to the transcriptional state analysis that has already been reported. In Fig 7 they make the interesting observation that the same chromatin class is associated with multiple transcriptional classes, but not vice versa. This is an interesting result, but still begs the question of added value of the chromatin analysis. It would be good if the authors added a more detailed discussion of what unique insights are provided by the chromatin analysis above and beyond those provided by the transcriptional state analysis alone.
5. Along these lines, it would be interesting to explore if the difference in transcriptional states could be explained by different transcription factors accessing subsets of the open chromatin regions.
6. The authors use the concordance between chromatin accessibility and gene expression to validate cell type-specific marker genes. But it is also interesting to describe genes that have open chromatin but are not expressed. These may be genes that are poised to respond to changes in their environment in a cell type-specific fashion. These are also very interesting genes and so it would be useful to look for and highlight these genes in the different chromatin cell types.
7. The identification of pathogenic RA chromatin classes is a very important finding from these studies. I would move Supplementary Figure 7 to the main body of the manuscript.
8. The paper is packed with a lot of very valuable information about these chromatin cell types, their gene accessibility biomarkers, with and without correlated gene expression, and their association with clinical metrics of RA, previously defined RA subtypes, and previously identified RA genetic risk variants that is buried throughout the text. I would strongly recommend that the authors compile all of this useful information into a single summary table that could serve as a reference to the community.

Reviewer #3 (expert in computational biology, regulation of gene expression and epigenomics):

This study presents a comprehensive analysis of chromatin landscape of synovial tissues from RA patients at single cell levels using scATAC-seq and multiomic analyses. The data are highly valuable to the community and timely needed for studying RA pathogenesis. Clustering analysis defined 6 broad cell types and 24 chromatin classes were further uncovered within 5 of these cell types, from which marker genes and motifs were uncovered. These chromatin classes were compared with the AMP RA clusters defined from single cell transcriptomic studies and each chromatin class corresponded to multiple transcriptomic classes. The authors therefore referred to the chromatin classes as superstates corresponding to multiple transcriptional cell states. This is an interesting but not surprising observation and this superstate hypothesis needs additional validations.

Overall, this is an interesting study and the authors may want to consider the following points to improve the manuscript. (1) As a data driven study, the scATAC-seq, multiome and CITE-seq data from previous study were analyzed in this work. The clarity of the description on data analysis and integration needs to be improved and workflow chart is recommended to summarize the procedure and elucidate the logic. (2) The superstate hypothesis is intriguing but additional validations are needed.

Detailed comments are the following.

1. As the large dataset is a valuable resource to the community, it would be helpful to have a summary table of QC such as total reads, mapping rate, percentage of reads falling in peak neighborhoods, reads in promoters, mitochondrial reads, reads falling in the blacklisted regions for the final selected cells.
2. Access ID for raw and processed data should be provided.
3. While the consensus open chromatin peaks of scATAC-seq are called from all the cells pooled to a bulk, how are peaks called for snATAC-seq? Line 797-798, "an average of 75% (n=12 datasets; range: 66%-83%) of the 200bp trimmed snATAC-seq donor-specific peaks overlapping the scATAC-seq consensus peaks", how are donor-specific snATAC-seq peaks called? More details need to be provided.
4. "Broad cell type clustering" in Methods does not really describe the procedure. Initial broad cell type clustering is mentioned in "ATAC quality control". Is there any further broad cell type clustering after the initial one? It'd better present a summary of the procedure with a workflow chart to help readers understand how the open chromatin data are clustered to six cell types.
5. scATAC-seq and snATAC-seq data are combined in each broad cell type to define fine-grain chromatin classes within the cell type. Line 841-842, "After subsetting the matrix by PMA peaks, we ran the same clustering pipeline detailed in the broad cell type clustering section with 10 PCs requested." There is no detailed description of the pipeline in the broad cell type clustering section with 10 PCs. Are all the cells from scATAC and snATAC pooled to call consensus peaks for clustering? How to decide the number of clusters (not much discussion on Supplementary Fig. 10)? Need to clarify and provide details.
6. Line 828-831, "We also classified the multiome snRNA cells into the AMP-RA CITE-seq study12 broad cell types using Symphony (see Symphony classification of transcriptional cell state). The small minority of cells (2%) with discordant cell types defined in the snATAC, snRNA, and CITE seq modalities for the multiome datasets were removed.". snRNA-seq were classified to the broad cell types defined by AMP-RA CITE-seq study using Symphony.
7. It is weird to compare the RA tissue with healthy PBMC. It is hard to draw any solid conclusion because the tissue chromatin classes can be due to the difference between synovial tissue and blood.

8. It is common that scRNA-seq identifies more clusters than scATAC-seq. For the "superstate" found from scATAC-seq data that represent multiple transcriptional states, is it possible that it is due to the different coverages of scRNA-seq and scATAC-seq? To rule out this possibility and validate the superstate hypothesis, cells in the same superstate need to be sorted out and their transcriptomic profiles need to be analyzed at single cell level.

9. Another analysis missing from the manuscript is to compare the pooled snATAC profiles of the multiome cells in different AMP RA transcriptional states to which snRNA were assigned. As snRNA cells were assigned to the AMP RA transcriptomic clusters, pooling snATAC in the same cluster can improve the coverage and may detect the differences between the transcriptional states or confirm there is no difference.

10. In Discussion Line 605-608, any evidence to support that non-pathogenic transcriptional cell states are able to transition to pathogenic transcriptional cell state if they correspond to the same chromatin class?

RESPONSE TO REVIEWERS' COMMENTS

We thank all three reviewers for their support of our study and their valuable feedback. We were very encouraged by Reviewers' praise of the novelty of our study and its usefulness as a reference in the field. For example, Reviewer 1 declared "The data is novel and of high scientific interest. The analysis is very well done and presented in a clear way." Reviewer 2 affirmed "This is an excellent manuscript and the results reported will serve as a valuable reference about the cellular and molecular characteristics of this important autoimmune disease."

However, there were some improvements suggested by the Reviewers that we have sought to address by adding four major areas to our manuscript, summarized here. We include more details, to these areas and indeed all comments, in our point-by-point response.

1. **Experimental validation of our proposed superstate model.** Using valuable peripheral blood from four RA patients, we FAC-sorted four populations spanning two chromatin classes and four transcriptional states: CD4⁺CD127⁻CD25^{hi} Treg, CD4⁺CD127⁻CD25^{int} Treg, CD4⁺CD25⁻PD1⁺CXCR5⁺ TFH, and CD4⁺CD25⁻PD1⁺CXCR5⁻ TPH. We then performed a multiome experiment and obtained ATAC, RNA, and surface protein information for each gold standard sorted population. This optimized strategy allowed us to confirm that our RA tissue chromatin classes matched the sorted chromatin classes, our four transcriptional cell states were distinct using FACS, and the differential genes between states within a class did not appear to have corresponding differential promoter peaks. We believe this dataset adds to the robust resource presented in this manuscript and provides a reference dataset for future investigations into putatively pathogenic T cell populations in RA from multiple viewpoints, and will thus be publicly available as well. Please see Reviewer 1 Comment 3 and Reviewer 3 Comment 8 for more details.
2. **Methods section reorganization and workflow figure.** As many of our methods are reused in different contexts, we followed Reviewer 3's useful suggestion of creating a workflow figure (**Supplementary Fig. 1**). We have gone further and have restructured our **Methods** section to follow this new workflow. The result is a more streamlined section that is easier for readers to follow. We augmented this section by adding clarifying details throughout, but especially in the sections describing peak calling and clustering. Please see Reviewer 3 Comments 3-6 for more details.
3. **Details regarding chromatin class identification.** Additionally, we added more details about how we named our chromatin classes. Briefly, we used: 1) the class-specificity of the marker gene's expression, 2) the class-specificity of the marker peak associated with that gene's promoter, 3) cross-referencing literature about the gene as a marker for specific cell states, and 4) reference mapping to a well-annotated RA tissue CITE-seq dataset. Please see Reviewer 2 Comments 1-2 and 8 for more details.
4. **Data access.** We have deposited our data into public, curated databases. Raw unimodal scATAC FASTQs and all processed files are in Synapse (accession ID syn53650034). Raw multimodal snATAC and snRNA FASTQs, including those for the sorted RA PBMC experiment, are in dbGaP (accession ID phs003417.v1.p1). Please see Reviewer 3 Comment 2 for more details.

Reviewer #1 (expert in the epigenetics of rheumatoid arthritis):

Weinand and colleagues present data from an integrated single-cell RNA/ATAC sequencing approach to define cellular states in the synovium. The data is novel and of high scientific interest. The analysis is very well done and presented in a clear way. I have the following suggestions to improve the manuscript.

Response: We thank the reviewer for their endorsements and helpful suggestions.

R1C1. OA samples were included to increase cell counts, but the analysis would be cleaner if only RA samples were used. It would be best to add a sufficient number of OA samples to separate the data sets and make a valid comparison. Alternatively, at least the cell counts in the supplemental data showing the individual cell clusters should indicate which were OA samples and it should be mentioned how many cells are from OA samples. It is assumed that the T/B cell analysis mainly reflects RA states, whereas a more significant contribution from OA samples is expected in the stromal population.

Response: Inspired by the reviewer's insightful comment, we wanted to understand if including OA samples was altering our definition of chromatin classes. To verify that our chromatin classes remain valid when only using RA samples, we removed OA samples and re-clustered remaining cells in the same manner as the original chromatin classes described in **Methods** section **Fine-grain chromatin class clustering**. After removing 5 OA samples with 2,395 total T cells, we found that the RA only T cell clusters had a 1-1 correspondence with our original chromatin classes with a positive, significant odds ratio only between corresponding classes in **Supplementary Fig. 8a**. This analysis validates the assumption that our T cell chromatin classes are mainly driven by RA samples. We also performed a similar analysis with stromal cells by removing the 5 OA samples with 4,462 total stromal cells and re-clustering the remaining cells. While these clusters split slightly differently, all 4 of our original stromal chromatin classes have corresponding RA only cluster(s) in **Supplementary Fig. 8b**. Removing a higher proportion of OA cells in stromal populations did have a larger effect on the clusters generated than in T cells, but our original chromatin classes are maintained.

Additional Supplementary Fig. 8:

Legend: **Supplementary Fig. 8**. Chromatin classes are stable including or excluding OA and low-cell-count samples.

a. UMAP colored by T cell clusters defined from unimodal scATAC and multimodal snATAC T cells in RA samples only (**left**) and the natural log of the Odds Ratio between these clusters and the T cell chromatin classes defined in **Fig. 2** (**right**).

b. UMAP colored by stromal clusters defined from unimodal scATAC and multimodal snATAC stromal cells in RA samples only (**left**) and the natural log of the Odds Ratio between these clusters and the stromal chromatin classes defined in **Fig. 3** (**right**).

We agree with the reviewer that while a comparison between RA and OA would add further insights into the disease pathogenesis of OA, our study is primarily focused on characterizing the chromatin landscapes of RA synovial cells. A detailed analysis of OA synovial cells from a well-characterized OA cohort is warranted in a separate, dedicated study. As the alternative suggested by the reviewer, we highlighted the presence of OA samples in **Supplementary Fig. 2d-e** and panel **a** in **Supplementary Figs. 3-7** with a '>' symbol on the y-axis and outlining their bars in black. We also created **Supplementary Table 6** that lists both the absolute cell counts and relative frequencies of OA and RA cells for each chromatin class. In general, we did see higher OA cell proportions in stromal populations than in T cells, as hypothesized by the reviewer, with the highest proportion (23%) of OA cells in the SA-1: PRG4+ lining cells, as seen in multiple studies (Zhang et al., Nature, 2023; Zhang et al., Nat Immunol, 2019; Mizoguchi et al., Nat Comm, 2018). Plasma cells also had low proportions of OA cells while B cells had higher proportions relative to the other chromatin classes, though with lower cell counts, as this reviewer pointed out in the next comment, proportions are less reliable.

Addition to **Results** [Between endothelial cell type and Tissue vs PBMC comparison]:

Chromatin classes are stable irrespective of OA and low cell count samples

Our chromatin classes were determined using all samples for maximum power, so we next investigated the contribution of OA and low-cell-count samples to this classification. While we were underpowered to reliably detect differences between RA and OA, we saw evidence that chromatin classes varied in their proportions between these two diseases (**Supplementary Table 6**). To determine if the chromatin class definitions were robust to the exclusion of OA samples, we removed the 2,395 T cells corresponding to OA samples and reclustered the remaining cells. We only observed positive, significant odds ratios (ORs) for cells from a new RA-only cluster belonging to their corresponding original chromatin class relative to the other classes (**Supplementary Fig. 8a**). This showed that the same groups of RA T cells cluster together regardless of whether OA T cells were included in the clustering. Since stromal cells had a higher proportion of OA cells, particularly in lining fibroblasts (**Supplementary Table 6**; Zhang et al., Nature, 2023; Mizoguchi et al., Nature Communications, 2018), we also reclustered the stromal cells after removing 4,462 cells from OA samples and found that all four of our original stromal chromatin classes had corresponding RA-only cluster(s) (**Supplementary Fig. 8b**).

Addition to the **Supplementary Figs.**:

Supplementary Fig. 2d-e:

Legend: d.-e. Broad cell type cell counts per sample for (d.) unimodal and (e.) multimodal datasets. OA samples are highlighted with a '>' symbol and black bar outlines.

Supplementary Fig. 3a:

Legend: a. T cell chromatin class cell counts per sample for unimodal and multimodal datasets. OA samples are highlighted with a '>' symbol and black bar outlines.

Supplementary Fig. 4a:

Legend: a. Stromal chromatin class cell counts per sample for unimodal and multimodal datasets. OA samples are highlighted with a '>' symbol and black bar outlines.

Supplementary Fig. 5a:

Legend: a. Myeloid chromatin class cell counts per sample for unimodal and multimodal datasets. OA samples are highlighted with a '>' symbol and black bar outlines.

Supplementary Fig. 6a:

Legend: a. B/plasma chromatin class cell counts per sample for unimodal and multimodal datasets. OA samples are highlighted with a '>' symbol and black bar outlines.

Supplementary Fig. 7a:

Legend: a. Endothelial chromatin class cell counts per sample for unimodal and multimodal datasets. OA samples are highlighted with a '>' symbol and black bar outlines.

Additional Supplementary Table 6:

Chromatin Classes	Cell Counts		Cell Percentages	
	OA	RA	OA	RA
TA-0: CD8A+GZMK+	637	6736	9%	91%
TA-1: CD4+ IL7R+	805	5763	12%	88%
TA-2: CD4+ PD-1+ TFH/TPH	432	3934	10%	90%
TA-3: CD4+ IKZF2+ Treg	357	2736	12%	88%
TA-4: CD8A+ PRF1+ cytotoxic	164	1604	9%	91%
SA-0: CXCL12+ HLA-DRhi sublining	1812	9921	15%	85%
SA-1: PRG4+ lining	2210	7197	23%	77%
SA-2: CD34+ MFAP5+ sublining	371	2261	14%	86%
SA-3: MCAM+ mural	69	466	13%	87%
MA-0: F13A1+ MARCKS+ TRM	1251	5530	18%	82%
MA-1: FCN1+ SAMSN1+ infiltrating monocytes	725	5460	12%	88%
MA-2: LYVE1+ TMD4+ TRM	1140	4866	19%	81%
MA-3: CD11C+ AFF3+ DC	762	3028	20%	80%
MA-4: SPP1+ FABP5+ intermediate	303	2626	10%	90%
BA-0: CREB3L2+ plasma	118	2752	4%	96%
BA-1: CD27+ plasma	61	1590	4%	96%
BA-2: TOX+ PDE4D+ switched memory B	246	1201	17%	83%
BA-3: FCER2+ IGHG+ naive B	237	1053	18%	82%
BA-4: CD24+ MAST4+ unswitched memory B	236	636	27%	73%
BA-5: ITGAX+ ABC	116	395	23%	77%
EA-0: SELP+ venular	238	2146	10%	90%
EA-1: RGCC+ capillary	55	893	6%	94%
EA-2: SEMA3G+ arteriolar	30	305	9%	91%
EA-3: PROX1+ lymphatic	13	129	9%	91%

Supplementary Table 6. The absolute cell counts and relative frequencies of OA and RA cells in each chromatin class.

R1C2. In Suppl. Fig. 1e, one sample contained very few cells and did not reflect all synovial cells. Also, for cell counts for specific cell types, the different chromatin classes contained less than 100 cells in some samples (especially B/plasma cells and endothelial cells). Here, the analysis is determined by a few samples. Samples with so few cells/patients should be re-examined in detail and possibly removed. At least this should be mentioned as a limitation in the discussion.

Response: We thank the reviewer for highlighting the issue of samples and cell types with lower cell counts. To better address this limitation, we did a secondary analysis directly testing this limitation and added it to the main text results and discussion.

To determine the effect of removing these low count samples, especially in the overall lower cell count cell types, we reanalyzed the B/plasma populations after removing 11 samples, including the low cell count sample the reviewer noted in **Supplementary Fig. 2e** (previously **Supplementary Fig. 1e**), with fewer than 100 B/plasma cells for a total of 467 cells removed. All of the original B/plasma chromatin classes were represented in this reclustering analysis (**Supplementary Fig. 8c**), suggesting that these

low cell count samples do not alter chromatin class definitions. Similarly, we repeated this analysis in endothelial cells, removing 954 cells from 19 samples, and saw a strong 1-1 correspondence between the new clusters and our original chromatin classes (**Supplementary Fig. 8d**).

Additional Supplementary Fig. 8:

Legend: c. UMAP colored by B/plasma clusters defined from unimodal scATAC and multimodal snATAC B/plasma cells in high-cell-count (HCC) samples with at least 100 B/plasma cells (**left**) and the natural log of the Odds Ratio between these clusters and the B/plasma chromatin classes defined in **Fig. 5 (right)**.

d. UMAP colored by endothelial clusters defined from unimodal scATAC and multimodal snATAC endothelial cells in HCC samples with at least 100 endothelial cells (**left**) and the natural log of the Odds Ratio between these clusters and the endothelial chromatin classes defined in **Fig. 6 (right)**.

In all **right** panels, non-significant ($FDR > 0.05$) OR values are white and the colors of the x-axis labels correspond to the colors in the UMAPs on the **left**.

We decided on the basis of this analysis to include all samples. We further note that we account for sample differences by correcting batch effects. We felt that including and making as much of this data accessible to the public was valuable.

Addition to Results [continuation of paragraph in R1C1]:

Furthermore, we were curious if including the low-cell-count samples was impacting the chromatin class definitions, especially for the cell types with lower cell counts overall. To test this, we removed 467 cells from 11 samples with fewer than 100 B/plasma cells and reclustered the remaining cells. We were able to recover all the original B/plasma chromatin classes

(**Supplementary Fig. 8c**), suggesting that these low-cell-count samples did not drive our original classes. We saw similar results in endothelial cells after removing 954 cells from 19 samples (**Supplementary Fig. 8d**). These analyses suggested our chromatin classes were robust to the inclusion of both OA and low-cell-count samples.

Addition to **Discussion** [in the next steps paragraph]:

Third, even though we did not see large effects of OA and low cell counts samples on our chromatin classes, a larger study with a more even distribution of RA and OA samples with higher cell counts would be better able to distinguish between RA- and OA-specific chromatin variation.

R1C3. The hypothesis of chromatin superstates of cell populations is very compelling. However, to match these cellular superstates with long-known subclasses of cells (e.g., in T cells), it would be helpful to perform the analysis approach from the other end (presorting known cell subclasses and then ATACseq), at least in one subclass. For example, there was good overlap between Tregs as defined by ATACseq in PBMC and in tissues. How do these chromatin profiles compare to a chromatin profile of a PBMC Treg population sorted using established protein markers?

Response: We thank the reviewer for this wonderful suggestion. With the proposed superstate model being such an important conclusion within our paper, we were very excited to experimentally validate it. Using PBMC samples from four RA patients, we sorted CD4⁺CD127⁻CD25⁺ Tregs and CD4⁺CD25⁺PD1⁺ TFH/TPH populations via FACS and obtained single cell ATAC data. This experiment was also used to address R3C8, where it is presented in additional detail.

To answer your question, we determined that the chromatin profiles of the FACS RA PBMC populations corresponded to our chromatin classes by replicating the healthy PBMC and RA tissue ATAC analysis shown in **Supplementary Fig. 9**. Briefly, we *de novo* clustered the combined tissue and PBMC ATAC profiles into 10 clusters (**Supplementary Fig. 15c**). We found that RA PBMC TFH/TPH cells were most enriched in combined cluster 2 (OR=4), which was most highly enriched for RA tissue TFH/TPH cells (OR=32). Similarly, RA PBMC Tregs were most enriched for cluster 4 (OR=3), which was most highly enriched for tissue Tregs (OR=24). This confirmed that our tissue annotations agreed with long-known subclasses of T cells sorted using established protein markers. Unsurprisingly, since the combined clusters were determined by ATAC clustering, the RA tissue chromatin classes, also determined by ATAC clustering, had better ORs than the protein-sorted hashtags.

Additional **Supplementary Fig. 15**:

Legend: **c.** Clustering RA tissue unimodal scATAC, RA tissue multimodal snATAC, and sorted RA PBMC multimodal snATAC cells together visualized on UMAP (**left**) and the natural log of the Odds Ratio between these clusters and the RA tissue/PBMC labels (**right**). Non-significant (FDR>0.05) OR values are white. The colors of the x-axis labels on the **right** correspond to the colors in the UMAPs on the **left**. On the **right** y-axis, RA tissue chromatin classes are colored in purple and PBMC cell states are colored in red.

Addition to **Results** [last paragraph in superstate section]:

When we *de novo* clustered the ATAC modalities of the combined PBMC and tissue cells (**Supplementary Fig. 15c; Methods**), we found that the sorted RA PBMC TFH/TPH cells were most enriched in combined cluster 2 (OR=4), which was most highly enriched for RA tissue TFH/TPH cells (OR=32). Similarly, sorted RA PBMC Tregs were most enriched for combined cluster 4 (OR=3), which was most highly enriched for RA tissue Tregs (OR=24). This confirmed that our tissue class annotations agreed with well-known subclasses of T cells sorted using established protein markers.

Addition to **Discussion** [in next steps paragraph]:

Second, to better understand whether the more pathogenic chromatin classes such as T_A-2: CD4+ PD-1+ TFH/TPH and M_A-1: FCN1+ SAMS1+ infiltrating monocytes are indeed only in tissue, a RA PBMC scATAC-seq study may be warranted. While we saw a general consensus between the chromatin landscapes of RA tissue and our small population of RA blood TFH/TPH cells, a larger PBMC study would be better powered to determine if the chromatin environment in blood may be a proxy for the environment in tissue that gives rise to pathogenic transcriptional populations.

Addition to **Methods** [in **Tissue and blood analysis** section]:

We replicated this analysis using the RA PBMCs for TFH/TPH and Treg FACS populations and the 5 RA tissue chromatin classes.

R1C4. Was the enrichment of transcription factor motifs in a particular chromatin state associated with increased transcription of these transcription factors in the superstates or their corresponding subclasses?

Response: We thank the reviewer for this interesting question. We note that we only included the transcription factor (TF) in our motif heatmap if the corresponding TF had expression above a minimal

mean threshold within the chromatin class as referenced in the **Transcription Factor motif analysis Methods** section. However, we had not explicitly assessed whether the TF gene expression is higher in the class with that TF motif enriched than the other classes in that cell type.

Therefore, we investigated whether TFs were more highly expressed in the chromatin class in which their motif was among the most accessible. For each top TF gene and chromatin class combination within a cell type, we used a one-tailed Wilcoxon test comparing the normalized gene expression between cells in that chromatin class and cells not in that class (**Supplementary Table 5**). We calculated a False Discovery Rate (FDR) from these nominal p-values within each cell type. Reassuringly, we do see specific instances of known TFs having good correspondence between TF motif accessibility and TF gene expression, such as *EOMES* in T_A-0 (FDR=1.92e-84), *BATF* in T_A-2 (FDR=3.10e-127), *TEAD1* in S_A-0 (FDR=1.05e-27), *STAT3* in S_A-0 (FDR=1.65e-17), *SPIB* in M_A-3 (FDR=6.93e-74), *EBF1* in B_A-2 (FDR=8.59e-49), *JUN* in B_A-1 (FDR=1.60e-47), and *SOX17* in E_A-2 (FDR=4.29e-19). However, we do find some TF/class combinations, such as *TCF7L2* in T_A-1 and *EBF3* in S_A-3, with a lack of concordance. This could be because TFs are generally lowly expressed, TFs do not have to be differentially expressed to be functional, accessible motifs are not always bound by TFs, and motifs can be bound by multiple TFs (Vaquerizas et al., Nature Reviews Genetics, 2009; Lambert et al., Cell, 2018).

Addition to **Results**:

T cell TF paragraph:

In the primarily CD8⁺ classes, T_A-0: CD8A⁺ GZMK⁺ and T_A-4: CD8A⁺ PRF1⁺ cytotoxic, we found *EOMES* ($p_{\text{adj}}=7.44\text{e-}99$, $8.12\text{e-}44$, respectively) and T-bet (*TBX21*) ($p_{\text{adj}}=4.92\text{e-}90$, $2.75\text{e-}38$, respectively) motifs preferentially enriched (**Fig. 2c**); the corresponding TFs are known to drive memory and effector CD8⁺ cell states³⁶. *EOMES* had significantly higher gene expression in T_A-0 cells compared to all other T cells (Wilcoxon FDR=1.92e-84; **Supplementary Table 5**). [...]

Within the T_A-2: CD4⁺ PD-1⁺ TFH/TPH class, we observed high enrichments for AP-1 motifs, especially *BATF* ($p_{\text{adj}}=3.31\text{e-}103$; **Fig. 2d**), which promotes expression of key programs in TFH cells³⁹ and had higher gene expression in this class's cells (Wilcoxon FDR=3.10e-127; **Supplementary Table 5**).

Stromal TF paragraph:

In the S_A-0: CXCL12⁺ HLA-DR^{hi} sublining chromatin class, we found *TEAD1*⁵² ($p_{\text{adj}}=2.86\text{e-}52$; **Fig. 3c**) and *STAT1/3* TF motif enrichments ($p_{\text{adj}}=3.34\text{e-}37$, $4.27\text{e-}38$, respectively; **Fig. 3c**), with the latter likely regulating the JAK/STAT pathway responsible for proinflammatory cytokine activation central to RA clinical activity^{9,53}. The gene expression of *TEAD1* and *STAT3* in S_A-0 cells was significantly higher than in the other stromal cells (Wilcoxon FDR= 1.05e-27 and 1.65e-17, respectively; **Supplementary Table 5**).

Myeloid TF paragraph:

SPI1 (PU.1) is the master regulator of myeloid development⁶⁶, including conventional DCs⁶⁷. We found PU.1 motifs most strongly enriched in the DC cluster M_A-3 ($p_{\text{adj}}=3.24\text{e-}55$; **Fig. 4c**), though the related *SPIB* motif's corresponding transcription factor, known to function in pDCs (Schotte et al., J Exp Med, 2004), was more specifically expressed in this class (Wilcoxon FDR=6.93e-74; **Supplementary Table 5**).

B/plasma TF paragraph:

B_A-1 was enriched for AP-1 factor motifs⁸², namely BATF::JUN ($p_{\text{adj}}=0$; **Fig. 5c-d, Supplementary Fig. 6c**). Both *BATF* and *JUN* gene expression was higher in B_A-1 cells compared to those in other B/plasma classes (Wilcoxon FDR= 9.29e-04 and 1.60e-47, respectively; **Supplementary Table 5**).

Endothelial TF paragraph:

We identified SOX motifs⁸⁷ in E_A-2, STAT motifs⁸⁸ in E_A-0, and AP-1 motifs⁸⁹ in E_A-1 (**Fig. 6c**). *Sox17* is a crucial intermediary between Wnt and Notch signaling that specifically initiates and maintains endothelial arterial identity in mice⁸⁷. Similarly, we found a SOX17 motif ($p_{\text{adj}}=3.27\text{e-}8$) in the promoter of *NES*^{90,91} with its highest accessibility and expression (Wilcoxon FDR=4.29e-19; **Supplementary Table 5**) in E_A-2 cells (**Fig. 6d**).

Addition to **Methods** [in **Transcription Factor motif analysis** section]:

For the TFs associated to the top class-specific accessible motifs, we used a one-tailed Wilcoxon test to compare the normalized gene expression for the TF between cells in that chromatin class and the other cells within that cell type, with the alternative hypothesis being “greater” and multiple hypothesis test correction within cell types using FDR (**Supplementary Table 5**).

Additional **Supplementary Table 5** [full table in Excel]:

Cell Type	Class	TF	Mean In Class	Mean Outside of Class	Wilcoxon W statistic	Wilcoxon P-value	Wilcoxon FDR	Negative Log10 Wilcoxon FDR
T cells	TA-0	EOMES	0.22	0.03	7255465	1.07E-85	1.92E-84	84
T cells	TA-0	TBX21	0.14	0.06	6820252	1.76E-13	1.05E-12	12
T cells	TA-0	MGA	0.34	0.36	6407104	9.80E-01	1.00E+00	0
T cells	TA-0	IKZF1	1.22	1.30	6306649	9.96E-01	1.00E+00	0
T cells	TA-0	IRF9	0.14	0.14	6507778	7.89E-01	1.00E+00	0
T cells	TA-0	SMAD2	0.40	0.42	6402708	9.78E-01	1.00E+00	0
T cells	TA-0	SMAD3	0.51	0.67	5971160	1.00E+00	1.00E+00	0
T cells	TA-0	SMAD4	0.21	0.27	6239646	1.00E+00	1.00E+00	0
T cells	TA-0	IRF4	0.22	0.19	6597680	1.61E-01	3.87E-01	0
T cells	TA-4	GABPA	0.09	0.10	1734528	8.78E-01	1.00E+00	0
T cells	TA-4	ELF1	1.43	1.53	1654266	9.86E-01	1.00E+00	0
T cells	TA-4	RREB1	0.29	0.27	1772595	3.38E-01	7.16E-01	0
T cells	TA-4	RORC	0.09	0.01	1848446	3.38E-25	4.06E-24	23
T cells	TA-4	RUNX3	1.30	0.98	2041356	2.49E-10	1.28E-09	9
T cells	TA-4	KLF9	0.49	0.39	1859484	2.95E-03	9.65E-03	2
T cells	TA-4	RUNX2	0.65	0.41	1974204	8.85E-10	3.98E-09	8
T cells	TA-1	TCF7L2	0.12	0.10	7412912	3.23E-02	9.69E-02	1
T cells	TA-1	LEF1	0.54	0.37	7974308	1.01E-18	9.08E-18	17
T cells	TA-1	KLF4	0.10	0.12	7223257	9.94E-01	1.00E+00	0
T cells	TA-1	KLF5	0.09	0.05	7521055	3.25E-08	1.30E-07	7
T cells	TA-1	KLF2	0.80	0.62	8022034	9.71E-16	6.99E-15	14
T cells	TA-1	GATA3	0.40	0.32	7647066	5.40E-06	1.94E-05	5
T cells	TA-1	ZEB1	1.10	1.18	7047609	9.99E-01	1.00E+00	0
T cells	TA-2	JUNB	2.69	2.63	4769041	1.78E-01	4.00E-01	0
T cells	TA-2	JUND	0.98	0.93	4820405	4.83E-02	1.34E-01	1
T cells	TA-2	BATF	0.90	0.35	6135670	8.60E-127	3.10E-125	125
T cells	TA-2	JUN	0.96	1.42	3699158	1.00E+00	1.00E+00	0
T cells	TA-2	FOSL2	0.36	0.54	4254780	1.00E+00	1.00E+00	0
T cells	TA-2	FOS	1.68	1.82	4277896	1.00E+00	1.00E+00	0
T cells	TA-2	FOSB	1.26	1.68	3588339	1.00E+00	1.00E+00	0
T cells	TA-3	KLF10	0.07	0.11	4044441	9.99E-01	1.00E+00	0
T cells	TA-3	SP3	0.66	0.67	4120574	6.56E-01	1.00E+00	0
T cells	TA-3	EGR1	0.22	0.24	4083220	9.18E-01	1.00E+00	0
T cells	TA-3	KLF6	1.67	1.74	4027640	9.46E-01	1.00E+00	0
T cells	TA-3	SP1	0.06	0.07	4106364	9.25E-01	1.00E+00	0
T cells	TA-3	SP4	0.39	0.36	4221132	8.46E-02	2.18E-01	1

Supplementary Table 5. Wilcoxon test between the normalized gene expression of the TF gene in cells in the specified chromatin class ('In Class') and all other cells in that cell type ('Outside of Class'). The TFs chosen correspond to the top motifs enriched in class-specific accessible chromatin from **Fig. 2-6c, right**. FDRs calculated within cell types.

R1C5. It seems that the correlation between chromatin class and marker gene expression is markedly different between different cell types (best in T cells, worst in myeloid and B cells). Possible explanations for these differences should be discussed.

Response: We thank the reviewer for this thought-provoking comment.

One of the reasons for variable correlation is the set of marker genes chosen for each cell type. Based on Reviewer 2's comments 8 and 1, we formalized our differential peak and gene analyses and

updated our marker gene heatmaps in **Supplementary Figs. 3-7b** as well as added **Supplementary Table 9**. For the chosen markers in **Supplementary Figs. 3-7b**, the peak/gene correlation improved. Overall, the range of correlation values decreased with the relative ordering of cell types staying consistent: T cells had the best correlation at 0.92 and myeloid and B/plasma cells had the worst at 0.76 each.

A potential reason why the correlation between scaled peak accessibility and gene expression for the chromatin classes varies between cell types may be the heterogeneity within a cell type. For instance, the myeloid cell type contains cells resembling tissue resident macrophages, infiltrating monocytes, and dendritic cells; the B/plasma cell type contains multiple classes of B cells and plasma cells; and the stromal cell contains fibroblasts and mural cells. Since the scaling done before correlations are calculated is across classes per gene or peak, these more heterogeneous cell types might be more affected by outliers than a more homogeneous cell type like T or endothelial cells. Furthermore, since peaks were called across broad cell types, some of these smaller populations like dendritic cells have slightly worse representation in their marker peaks (e.g., no peak for *FCER1A*). We also noted in the manuscript on Lines 296-297 that plasma cells were particularly hard to annotate with marker peaks as the immunoglobulin genes in our datasets had a paucity of chromatin accessibility.

Addition to **Discussion**:

Simultaneous chromatin accessibility and gene expression measurements in the multiome cells were essential to test the relationship between **marker peaks and genes**. **Across cell types, the correlations between scaled marker peak accessibility and gene expression across our chosen markers varied**. T cells had higher correlation ($R=0.92$; **Supplementary Fig. 3b**) while myeloid cells had lower correlation ($R=0.76$; **Supplementary Fig. 5b**), potentially due to more **heterogeneous subpopulations such as tissue-resident macrophages, infiltrating monocytes, and dendritic cells**.

R1C6. The enrichment of DNA methylation changes in SA-0 is interesting. Were the genomic sites affected by changes in DNA methylation within the SA-0 cluster random or were they enriched for marker genes important for differentiating this cluster from the other fibroblast clusters? If the latter, one could speculate that changes in DNA methylation stabilize the phenotype of the SA-0 cluster or one of its subclusters.

Response: We thank the reviewer for this interesting next step in the stromal DNA methylation analysis. We used the genes assigned to each differentially methylated region (DMR) by the original paper (Nakano et al., Ann Rheum Dis, 2013) and subsetted them to those corresponding to the DMR in our open peaks. We also subset to the fibroblasts within chromatin classes S_A-0 , S_A-1 , and S_A-2 , as mural cells were not present in the original DMR paper and were not associated to either hypo- or hyper-methylated DMR in **Supplementary Fig. 4c**. We then plotted the scaled mean normalized gene expression for the genes related to hypomethylated DMR. Intriguingly, we saw that the genes associated with hypomethylated DMRs were predominantly expressed in the S_A-0 chromatin class (**Supplementary Fig. 4d, right; Methods**), with many of those genes being prominent members of signaling pathways known to be relevant in RA sublining fibroblasts (Nakano et al., Ann Rheum Dis, 2013; Zhang et al., Nature, 2023): *STAT3* in IL-6 signaling, *CASP1* in IL-1 signaling, *TRAF2* in TNF signaling, and *TGFB3* in TGF β signaling. This could suggest that the hypomethylation in these key

genes allows for their high expression (Nakano et al., Ann Rheum Dis, 2013), thus contributing to the stabilization of the inflammatory sublining fibroblast phenotype as suggested by the reviewer.

Addition to Supplementary Fig. 4:

Legend: d. Scaled mean normalized gene expression across fibroblast chromatin classes in multimodal datasets for genes associated with hypermethylated (**left**) and hypomethylated (**right**) differentially methylated regions (DMR).

As we were intrigued by the results of the hypomethylated DMR, we repeated this analysis in the hypermethylated DMR. We found many more genes had higher expression in SA-1: PRG4+ lining fibroblasts (**Supplementary Fig. 4d, left**), with many of them known to be relevant in RA lining fibroblasts (Zhang et al., Nat Immunol, 2019; Collins, Roelofs et al., Ann Rheum Dis, 2023): *CLIC5*, *CD55*, *HBEGF*, *FOXO1*.

Addition to Results [stromal DNA methylation paragraph]:

DNA methylation and chromatin accessibility work in tandem to define cell-type-specific gene regulation through silencing CpG-dense promoters and repressing methylation-sensitive TF binding⁴⁶. Methylation changes have been previously described between cultured fibroblast cell lines from RA and OA patients^{47,48}. Thus, we wondered if a specific subset of fibroblasts might be the source of these differentially methylated regions (DMRs). Using a published set of DMRs for RA versus OA fibroblast-like synoviocyte (FLS) cell lines⁴⁷, we defined a per-cell score of peak accessibility associated to hypermethylated (positive) or hypomethylated (negative) loci in RA (**Methods**). The sublining fibroblasts in SA-0 were enriched for hypomethylated regions (Wilcoxon SA-0 cells versus rest one-sided $p < 2.2 \times 10^{-16}$), suggesting that the RA synovial fibroblast DMRs were relatively enriched for putatively functional chromatin accessible regions specifically in sublining fibroblasts (**Supplementary Fig. 4c**). Furthermore, the genes associated to these FLS differentially methylated regions were expressed primarily in tissue SA-0 (**Supplementary Fig. 4d, right; Methods**) and are crucial to a number of signaling pathways potentially at play in these inflammatory fibroblasts (Nakano et al., Ann Rheum Dis, 2013): *STAT3* in IL-6 signaling, *CASP1* in IL-1 signaling, *TRAF2* in TNF signaling, and *TGFB3* in TGFβ signaling. These results proposed the possibility of epigenetic memory retention even after multiple FLS cell line passages⁴⁹, as sublining fibroblasts, particularly HLA-DR^{hi} and CD34⁺ fibroblasts, are expanded in RA relative to OA in synovial tissue samples¹¹.

Addition to Methods [in Stromal DNA methylation analysis section]:

We used the genes assigned to the DM loci from the original paper (Nakano et al., Ann Rheum Dis, 2013). For the genes related to hypermethylated DM and hypomethylated DM accessible

loci separately, we plotted their scaled mean normalized gene expression within fibroblast classes S_A-0 , S_A-1 , and S_A-2 to assess fibroblast class preferences.

R1C7. Since changes in DNA methylation have been detected in cultured cells in fibroblasts, the question arises whether chromatin classes are maintained in cell culture even if the transcriptional profile is lost. Single-cell ATAC sequencing of cultured RA synovial fibroblasts should be performed to clarify the stability of chromatin classes independent of the synovial environment.

Response: We thank the reviewer for this intriguing question. To answer it, we requested data from our collaborators in the Donlin Lab from their recent paper, “Drivers of heterogeneity in synovial fibroblasts in rheumatoid arthritis” (Smith et al., Nat Immunol, 2023). Within that paper, they isolated fibroblast-like synoviocytes (FLS) from 2 RA synovial tissue samples, cultured them for three passages, and performed multiome sequencing. From those experiments, we used their genes x cells matrix and their fragments to quantify their cells’ fragment overlap with our peaks. Within our own RA tissue fibroblast cells, we created two fibroblast identity scores defining lining vs sublining: one using the most differentially expressed genes and one using the most differentially accessible peaks, subject to fold change and significance cutoffs. We used a Wilcoxon test to perform the differential testing and generated a per-cell score as done in the T cell lineage and stromal DNA methylation analyses (**Methods**) for both RNA and ATAC scores on our RA tissue fibroblasts and their cultured FLS cells. Unsurprisingly, we found that differential genes from tissue were able to separate tissue lining and tissue sublining cells, but the unstimulated cultured FLS after three passages did not have discernable lining and sublining populations with most scores around 0 (**Supplementary Fig. 4e**). This convergence of transcriptional identity in passaged FLS was also seen in a recent fibroblast study investigating the positional identity of stromal cells along the Notch gradient (Wei, Korsunsky, et al., Nature, 2021). More surprisingly, we saw similar results using the ATAC fibroblast score (**Supplementary Fig. 4f**), suggesting that fibroblast ATAC identity, and more broadly chromatin class identity, was not maintained in cell culture after multiple passages. This disconnect between DNA methylation and chromatin accessibility was also seen previously when assaying both directly using ATAC-Me in the monocyte-to-macrophage cell fate transition (Barnett et al., Mol Cell, 2020). That study found that chromatin accessibility, TF binding, and gene expression all occurred before substantial changes to the DNA methylation in distal regions important for cell fate transitions, suggesting that DNA methylation is more persistent than previously thought. Indeed, some TFs prefer to bind methylated DNA (Yin et al., Science, 2017), a potentially key mechanism to allow gene regulatory networks to persist even within ‘silenced’ regions (de Mendoza et al., Genome Biology, 2022).

Addition to **Supplementary Fig. 4**:

Legend: **e-f.** Fibroblast identity score between lining and sublining fibroblasts by cell based on (**e.**) normalized gene expression for differentially expressed genes and (**f.**) normalized peak accessibility for differentially accessible peaks in tissue (**Methods**), segregated by fibroblast source. The cultured fibroblast-like synoviocyte (FLS) data was obtained from Smith et al., Nat Immunol. 2023⁴⁴. All pairwise combinations of scores by source were significantly different by Wilcoxon test within both RNA and ATAC sets.

Addition to **Results** [after stromal methylation paragraph]:

We then considered if the retention of DNA methylation after multiple passages extended to a retention of chromatin accessibility or whether that would be lost alongside transcriptional identity (Wei, Korsunsky, et al., Nature, 2021). To assess this, we developed two per-cell scores of fibroblast identity comparing tissue lining (S_A-1) to sublining (S_A-0 , S_A-2) cells. The two scores were for RNA and ATAC modalities using differentially expressed genes and differentially accessible peaks, respectively (**Methods**). Using a multiome dataset of isolated FLS from two RA synovial tissue samples cultured for three passages in a recent RA fibroblast heterogeneity study (Smith et al., Nat Immunol, 2023) (**Methods**), we compared their per-cell fibroblast identity score to our tissue fibroblast populations in both RNA and ATAC space. Unsurprisingly, we found that differential genes from tissue were able to separate tissue lining and tissue sublining cells, but the cultured FLS did not have discernable lining and sublining populations by the same measure, consistent with previous results (Wei, Korsunsky, et al., Nature, 2021) (**Supplementary Fig. 4e**). More surprisingly, we saw similar results using the ATAC fibroblast identity score (**Supplementary Fig. 4f**), suggesting that fibroblast ATAC identity, and more broadly chromatin class identity, was not maintained in cell culture after multiple passages. This disconnect between DNA methylation and chromatin accessibility has also been seen previously when assaying both directly using ATAC-Me in the monocyte-to-macrophage cell fate transition (Barnett et al., Mol Cell, 2020).

Addition to **Methods**:

Cultured fibroblast datasets. We obtained two cultured unstimulated fibroblast-like synoviocyte (FLS) multiome datasets from Smith et al., Nat Immunol, 2023. We downloaded their genes x cells matrices from Immpart accession ID SDY2213 and fragment files from the authors. We subset these files by their QCed cells found in Immpart file `adata_scatac_chromVAR_motif_cultured.968213.h5`; there were 19,573 QC cells across the two samples. We overlapped this subsetted fragment file by our peaks to create a peaks x cells

matrix. We saw good overlap in that matrix with 99.99% of our peaks having at least 1 cell represented and all cells having overlapping fragments with at least a few hundred peaks. For both RNA and ATAC matrices individually, we concatenated the two samples and normalized.

Fibroblast identity analysis. We subsetted our stromal tissue datasets to only include fibroblast populations (S_{A-0} , S_{A-1} , S_{A-2}). We calculated differentially expressed genes between tissue lining (S_{A-1}) and sublining (S_{A-0} , S_{A-2}) populations in the normalized gene expression matrix using `presto::wilcoxauc` and adjusted p-values using FDR. We created gene sets of 382 lining and 254 sublining genes using the cutoffs: $FDR < 0.1$, $\log FC > 0.25$, and $AUC > 0.6$. We then calculated a per-cell score as in the **T cell lineage analysis** section, but with positive scores corresponding to lining fibroblasts and negative scores to sublining fibroblasts. Using the tissue gene sets, we calculated this per-cell RNA fibroblast identity score in the normalized cultured fibroblast gene expression matrix (see **Cultured fibroblast datasets**). We used a Wilcoxon test of RNA fibroblast identity scores between all pairs of fibroblast sources to determine significance via `ggpubr::compare_means`. We did the same analysis with differentially accessible peaks in the normalized chromatin accessibility matrix using cutoffs $FDR < 0.1$, $\log FC > 0.1$, and $AUC > 0.58$ to get 248 lining peaks and 294 sublining peaks.

Caroline Ospelt

Reviewer #2 (expert in computational biology and single-cell/single-nucleus transcriptomics/genomics):

The manuscript by Weinand et al. entitled "The Chromatin Landscape of Pathogenic Transcriptional Cell States in Rheumatoid Arthritis" describes an evaluation of cells isolated from synovial tissue of RA patients using scATAC-seq and combined snATAC-seq and snRNA-seq assays to define discrete chromatin states, a comparison of these chromatin states with previously published single cell transcriptional analysis of synovial tissue and PBMCs from RA patients, and a correlation between these chromatin states and their marker genes and clinical metrics of RA, previously defined RA subtypes, and previously identified RA genetic risk variants. The computational methods used for QC and batch correction are appropriate and the data analysis approaches are state-of-the-art for the analysis and interpretation of these types of single cell datasets and their correlation with the clinical and other parameters. This is an excellent manuscript and the results reported will serve as a valuable reference about the cellular and molecular characteristics of this important autoimmune disease.

Several suggested edits would contribute to enhancing the information reported as a reference for RA.

Response: We thank the reviewer for their complimentary words and beneficial edits to make our manuscript an even better RA reference.

R2C1. An important outcome of these kinds of single cell studies is the definition of discrete cell types using the measure characteristics, in this case chromatin states. In order for information about these cell types to be reusable, it is necessary to assign the cell types with some kind of unique cell type name/label. The authors use an interesting syntactical approach for naming cell types based on these chromatin characteristics - broad cell class, accessibility (A) subscript, sequential number, marker gene(s), previously-characterized cell class - e.g., TA-2: CD4+ PD-1+ TFH/TPH. From their combined analysis of chromatin promoter accessibility and gene expression, they note that both PD-1 and CTLA4 are expressed and their promoters are accessible. I'm guessing that their analysis reveals several genes with similar expression/accessibility characteristics for each cell type. The authors should clarify how/why a given gene (e.g., PD-1 alone) was selected for the cell type label. It appears that this is based on prior knowledge combined with the analysis results, but there may be some circumstances where a data driven approach would be better than relying on prior knowledge, which may be incomplete and biased.

Response: We thank the reviewer for this important comment.

We picked genes for chromatin class names based on a number of factors: 1) the class-specificity of the marker gene's expression, 2) the class-specificity of the marker peak associated to that gene's transcriptional start site, and 3) the recognizability of that gene as a marker of a specific cell identity in the field. Since we did not make this information clear enough in the existing **Methods**, we have added it to the **Fine-grain chromatin class clustering** section.

In regards to the number of named makers, we included two markers instead of one in the T cells as CD4 and CD8 are the most common types of T cells, but are insufficiently specific in our case. For example, PD-1 can be expressed by exhausted CD8 T cells, with some expression in T_A-0, even if it was most strongly expressed in our T_A-2 CD4 TFH/TPH cells. We also included two markers if one marker gene did not have a marker peak nearby (e.g., B_A-3: FCER2+ IGHD+ naive B), if a more differential markers is less recognizable to the field (e.g., S_A-2: CD34+ MFAP5+ sublining, M_A-1: FCN1+ SAMSN1+ infiltrating monocytes), or if two markers were equally good and informative (e.g., S_A-0: CXCL12+ HLA-DR^{hi} sublining, M_A-2: LYVE1+ TIMD4+ TRM).

We showed both the named marker genes as well as other genes with similar expression/accessibility characteristics for each cell type in panel **b** of **Supplementary Figs. 3-7**, which we updated and expanded as part of your eighth comment along with **Supplementary Table 9**, which lists the top 5 differential peaks and genes as well as known markers for each class as part of a more data driven approach.

As a further data-driven approach, we used the transcriptional cell state annotations from the large (>300K cells), well-curated AMP-RA reference of synovial tissue CITE-seq data to provide further support for the named markers and proposed class identity in our multiome data (**Fig. 7a-c**; **Supplementary Fig. 10g-h**). We used Symphony to map each multiome cell's gene expression onto the CITE-seq reference, and annotated each multiome cell's state as the most common transcriptional cell state among their 5 nearest neighbors in the reference (for more information, see **Methods** section

Symphony classification of transcriptional identity). The accuracy of this approach is described elsewhere (Kang et al *Nat Comm* 2021). As we noted in the main text, most cells mapped confidently, with at least 3/5 neighbors agreeing. Then, to translate this mapping to classes as a whole, we used an odds ratio to measure the strength of association between cells' membership in each pair of transcriptional cell state and chromatin class; we used a Fisher's exact test to measure significance (for more information, see **Methods** section **Class/state odds ratio**). The original reference paper (Zhang et al., *Nature*, 2023) listed differentially expressed genes for each transcriptional cell state, which we consulted when picking the markers chosen for the multiome class names.

Addition to **Methods** [in **Fine-grain chromatin class clustering** section]:

To label chromatin classes, we used the first letter of the broad cell types (T - T cell; S - stromal; M - myeloid; B - B/plasma; E - endothelial), a subscript A for accessibility, a cluster number (ordered by number of cells, with the biggest cluster named 0). To give biological context, we took advantage of both the peak accessibility and gene expression profiles. We chose a class's markers based on a number of factors: 1) the class-specificity of the marker gene's expression, 2) the class-specificity of the marker peak associated to that gene's promoter, 3) previous reports of that gene as a cell type marker in the literature, and 4) corroboration with a well-annotated RA tissue CITE-seq dataset via reference mapping (**Figs. 2-6b, 7a-c; Supplementary Figs. 1d, 3-7b, 10g-h; Supplementary Tables 7,9; Methods** sections **Single cell differential peak analysis, Single cell differential gene analysis, Symphony classification of transcriptional identity, and Class/state odds ratio**).

R2C2. They should also clarify how they connect the chromatin defined cell types with the previously-characterized cell class (e.g., TFH/TPH). Was this done manually? Could it be done statistically/computationally in such a way that captures some quantitative measure of confidence?

Response: We thank the reviewer for this comment.

As discussed in the response to your previous comment, we first looked at known marker genes and their corresponding promoter peak accessibility to propose a chromatin class identity, such as *PDCD1* and *CXCL13* in TFH/TPH (Rao et al., *Nature*, 2017) or *PRG4* and *CD55* in lining fibroblasts (Wei, Korsunsky, et al., *Nature*, 2021). We further supported the chromatin class identity by reference mapping to the larger RA tissue mRNA/protein atlas and comparing our class to their state annotations, respectively quantified with a mapping score and odds ratio.

In light of this comment, we were motivated to combine those two metrics to calculate the accuracy of the correspondence between class and state annotations. To be confident in our class-to-state mappings, we would expect the highest accuracy to be in cells whose class and state agreed, or in other words, transcriptional cell states that corresponded to a chromatin class based on our odds ratio strategy described above. We denoted those as 'concordant' cells, with an example being a multiome cell that was annotated as both class T_A-0 : CD8A+ GZMK+ and state T-14: CD8A+ GZMK+ memory. We compared them to cells whose class and state disagreed ('discordant'; e.g., T_A-2 : CD4+ PD-1+ TFH/TPH and T-14: CD8+ GZMK+ memory). We defined accuracy as the percentage of multiome cells with perfect mapping (*i.e.*, all 5 nearest neighbors in the reference had the same cell state) within each group of 'concordant' or 'discordant' cells. For each cell type, the 'concordant' cells had a more

confident state annotation than the ‘discordant’ cells, suggesting that cells whose state and class agreed with each other also agreed better with the well-annotated reference cells, giving multiple lines of evidence pointing to that cell identity. We note that T cells were once again artificially hampered in this metric by its 24 reference fine-grained cell states, making perfect agreement harder to achieve.

Addition to **Supplementary Fig. 10**:

Legend: **i**. For each cell type, the percentage of cells with perfect mapping (i.e., all 5 nearest neighbors in the reference had the same cell state) segregated by whether the cell’s classified transcriptional cell state was in the corresponding chromatin class (‘concordant’) or not (‘discordant’), as determined by the OR (**Fig. 7a-c**, **Supplementary Fig. 10g-h**).

Addition to the **Results** [superstate paragraph]:

We observed that each transcriptional cell state generally corresponded to a single chromatin class (**Fig. 7a-c**; **Supplementary Fig. 10g-h**). In contrast, a single chromatin class represents a superstate encompassing multiple transcriptionally defined cell states. For example, cells in the T_A-0 : CD8A+ GZMK+ chromatin class were more likely to be labeled in the T-5: CD4+ GZMK+ memory, T-13: CD8+ GZMK/B+ memory, and T-14: CD8+ GZMK+ transcriptional cell states across CD4/CD8 lineages (OR=11, 12, 11, respectively; **Fig. 7a**); the high GZMK promoter accessibility and expression shared by these states may contribute to this categorization (**Supplementary Fig. 10f**). We saw examples of this model in every cell type: S_A-1 linked to F-0/F-1 and S_A-0 to F-6/F-5/F-3/F-8 in stromal cells; M_A-1 to M-7/M-11 and M_A-4 to M-3/M-4 in myeloid cells; B_A-4 to B-1/B-3 in B/plasma cells; and E_A-0 to E-1/E-2 in endothelial cells as more examples (**Fig. 7b-c**; **Supplementary Fig. 10g-h**; **Supplementary Table 7**). In all cell types, the transcriptional cell state classification was more accurate within cells whose transcriptional cell state and chromatin class were concordant (e.g., T-14 and T_A-0), supporting our class-to-state mapping (**Supplementary Fig. 10i**; **Methods**).

Addition to **Methods** [in **Fine-grain chromatin class clustering** section]:

We proposed a cell identity based on known markers in the field; for example, *PDCD1* and *CXCL13* in TFH/TPH (Rao et al., Nature, 2017) or *PRG4* and *CD55* in lining fibroblasts (Wei, Korsunsky, et al., Nature, 2021). We further supported the proposed identity by the correspondence to the transcriptional cell state annotation from the well-annotated AMP-RA reference of synovial tissue CITE-seq data (**Fig. 7a-c**; **Supplementary Fig. 10g-h**; **Supplementary Table 7**; **Methods** sections **Symphony classification of transcriptional identity**, **Class/state odds ratio**).

Addition to **Methods** [in **Class/state odds ratio** section]:

We defined the accuracy of the class/state correspondence as the percentage of multiome cells with perfect mapping (*i.e.*, all 5 nearest neighbors in the reference had the same cell state) within each group of 'concordant' (*i.e.*, cells whose class and state agreed as determined by the odds ratio) or 'discordant' (*i.e.*, cells whose class and state disagreed) cells per cell type. For example, cells mapping to class T_A-0: CD8A+ GZMK+ and state T-14: CD8+ GZMK+ memory would be 'concordant' cells while cells mapping to class T_A-2: CD4+ PD-1+ TFH/TPH and state T-14: CD8+ GZMK+ memory would be 'discordant' cells.

R2C3. In the T cell section, 5 chromatin classes were defined. How do these relate to the naïve/central memory/effector memory T cell subsets paradigm?

Response: We thank the reviewer for this intriguing comment regarding the principally protein-derived naïve/central memory/effector memory paradigm in T cells.

Regarding naïve cells, we highlight two sentences on Lines 134-138 from the original manuscript:

We found one more predominantly CD4+ T cell class, T_A-1: CD4+ IL7R+, with high expression and accessibility for *IL7R*, encoding the CD127 protein. This marker is typically lost with activation, suggesting that T_A-1 is a population of unactivated naïve or **central** memory T cells, as further evidenced by *SELL* and *CCR7* expression (**Fig. 2b**; **Supplementary Fig. 3b**).

Because cell surface proteins are the gold standard for this classification, we used the well-annotated CITE-seq reference of RA synovial tissue to informally confirm that both naïve transcriptional cell states (T-4: CD4+ naïve and T-16: CD8+ CD45RO_{low}/naïve, as defined by CD45RA protein expression) were associated with the T_A-1 multiome superstate (**Fig. 7a**). As synovial tissue mainly harbors memory populations, as seen by widespread CD45RO expression in the CITE-seq reference, it is perhaps not surprising that we did not see distinct naïve populations in our 5 chromatin classes (**Fig. 2b**) or at higher resolutions (**Supplementary Fig. 13a**).

T_A-1 likely also harbored some central memory T cells since we saw *SELL* and *CCR7* expression and promoter peak accessibility. Other studies have shown a notable overlap in central memory T cell chromatin landscapes with naïve and effector memory cells (Jadhav et al., EBioMedicine, 2022; Giles et al., Immunity, 2022). While we confirmed that memory states containing CD45RO protein expression mapped to T_A-1 (T-0, T-1, T-2 in **Fig. 7a**), we could not ascertain the central memory identity given CD62L and *CCR7* were absent in the CITE-seq panels.

Since both T_A-0: CD8A+ GZMK+ and T_A-4: CD8A+ PRF1+ cytotoxic were depleted for *SELL* and *CCR7* gene expression and enriched for granzymes and perforin (**Supplementary Fig. 3b**), it is likely that they are primarily effector memory cells.

Addition to **Results** [in T cell section first paragraph]:

We found one more predominantly CD4+ T cell class, T_A-1: CD4+ IL7R+, with high expression and accessibility for *IL7R*, encoding the CD127 protein. This marker is typically lost with activation, suggesting that T_A-1 is a population of unactivated naïve or **central** memory T cells, as further evidenced by *SELL* and *CCR7* expression (**Fig. 2b**; **Supplementary Fig. 3b**).

[...]

We found another primarily CD8⁺ group of T cells, the T_A-4: CD8A⁺ PRF1⁺ cytotoxic cluster, which had high accessibility for the *PRF1* promoter and expression for the *PRF1*, *GNLY*, and *GZMB* genes, suggesting an effector memory phenotype (Fig. 2b; Supplementary Fig. 3b).

R2C4. In several of the figures, the authors show examples where the open chromatin state matches the gene expression data for the associated gene (e.g., Fig 2b). This raises the question of if the chromatin analysis has added any value to the transcriptional state analysis that has already been reported. In Fig 7 they make the interesting observation that the same chromatin class is associated with multiple transcriptional classes, but not vice versa. This is an interesting result, but still begs the question of added value of the chromatin analysis. It would be good if the authors added a more detailed discussion of what unique insights are provided by the chromatin analysis above and beyond those provided by the transcriptional state analysis alone.

Response: We thank the reviewer for reminding us to emphasize some of ATAC's unique insights.

We started by identifying correlated promoter peak chromatin accessibility and gene expression as that is the most straightforward sequence of events: the promoter of a required gene is accessible, thus allowing RNA pol II to transcribe it. However, that does not always have to be the case, as we show in response to R2C6, where sometimes the chromatin appears to be more promiscuously open, but the gene is more specifically expressed. For example, the specific expression of *RTKN2* may be dependent on FOXP3 binding and *CCL2* expression may be dependent on TNF or IFN γ signaling. Thus, one added benefit of chromatin accessibility is the ability to determine poised states as a regulatory feature of gene transcription (Guyer et al., Cell Reports, 2023; Yu et al., Cell Reports, 2020).

Another added benefit of ATAC data is the ability to determine where transcription factors might be binding, as most TFs require open chromatin to bind. While it is not as accurate as TF ChIP-seq or CUT&RUN, chromatin accessibility datasets are not factor-specific or dependent on antibodies, so they can capture potential regulatory sites for a broader set of factors. We depicted both (1) enriched motifs per class (Fig. 2-6c) as well as (2) specific examples where a differential motif is likely bound to a differential promoter peak that likely allows for differential gene expression (Fig. 2-6d). This cellular annotation by accessible TFs could help identify which populations are good candidates for a TF-specific ChIP-seq or CUT&RUN experiment. More generally, these interactions can be linked together in gene regulatory networks, where the inclusion of chromatin accessibility data can improve GRN inference (scRNA-seq only SCENIC vs multiome SCENIC+ in Gonzalez-Blas et al., Nature Methods, 2023), though that type of analysis was beyond the scope of our paper.

A third benefit of chromatin accessibility data is the inference of a noncoding variant's likely cell type of action (Fig. 8d). Roughly 90% of disease-causal genetic variants occur in noncoding regions (Farh et al., Nature, 2015), where there is no obvious coding change, thus requiring additional data to infer how the non-coding variant is regulating disease. scATAC-seq data by itself can point to which cell types that variant could be functioning as open chromatin is generally required for TF binding, the most direct way non-coding variants affect downstream processes.

Addition to **Discussion** [after first paragraph]:

Chromatin accessibility is a key piece in the puzzle of gene regulation. It determines which regions of the genome may participate in regulatory events such as TF binding or may be impacted by non-coding genetic variants. Accessible TF motifs are not guaranteed to be bound, in contrast to the regions identified in gold standard TF ChIP-seq (Solomon et al., Cell, 1988) or CUT&RUN (Skene and Henikoff, eLife, 2017). However, chromatin accessibility datasets are not TF-specific or dependent on antibodies, so they can capture potential regulatory sites for a broader set of factors. At a small scale, the regulation of key loci can be interrogated using scATAC-seq. For example, we found accessible AP-1 motifs in the differentially accessible promoter peak of *MMP3*, a key driver of RA extracellular matrix destruction, in lining fibroblasts compared to other stromal cells (**Fig. 3c-d**). Multiple drugs (e.g., CKD-506, T-5224, Roflumilast) are under investigation to disrupt this specific interaction of AP-1 at the *MMP3* promoter, and AP-1 signaling targets more broadly, in models of arthritis as well as clinical trials of RA patients (Balendran et al., Front Immunol., 2023). At a large scale, these TF-gene interactions can be linked together to form gene regulatory networks *in silico* (Gonzalez-Blas et al., Nature Methods, 2023; Kamimoto et al., Nature, 2023) to interrogate the more widespread effects of disrupting signaling cascades. Furthermore, as roughly 90% of disease causal genetic variants fall in non-coding regions (Farh et al., Nature, 2015), chromatin accessibility can prioritize where to look for functional effects of putatively causal RA genetic variants, particularly for those that disrupt TF motifs. Our analyses suggested that the likely causal SNP rs798000 may disrupt STAT binding in a TFH/TPH regulatory region reported to act on *CD2*, an important T cell co-stimulatory gene^{122,123}. Therefore, our study underscores the value of chromatin accessibility studies in disease-specific transcriptional regulation.

R2C5. Along these lines, it would be interesting to explore if the difference in transcriptional states could be explained by different transcription factors accessing subsets of the open chromatin regions.

Response: We thank the reviewer for this suggestion. We do find differences in the accessible TF motifs between different classes (**Fig. 2-6c**); therefore, we expect that the states that do not share a chromatin class will also show those differences. However, when we compared chromatin accessibility at promoter peaks between states within the same class (see R3C9 for more details), we found that very few peaks are differentially accessible (median 23 peaks across classes). Therefore, we were not able to investigate differential TF motifs between states as there were too few differentially open chromatin regions to complete a well-powered analysis.

R2C6. The authors use the concordance between chromatin accessibility and gene expression to validate cell type-specific marker genes. But it is also interesting to describe genes that have open chromatin but are not expressed. These may be genes that are poised to respond to changes in their environment in a cell type-specific fashion. These are also very interesting genes and so it would be useful to look for and highlight these genes in the different chromatin cell types.

Response: We thank the reviewer for this comment and we agree that it would be interesting to highlight some of these poised genes with promiscuous chromatin accessibility, but class-specific gene expression. One such gene is *RTKN2*, which has open chromatin at its promoter peak in all CD4 T cells but is only expressed in Tregs (**Supplementary Fig. 3b**). This class-specific gene expression in Tregs may arise from direct binding of the Treg master regulator FOXP3 as noted in mouse cells (Ferraro et al. PNAS, 2014). Furthermore, using human FOXP3 TF ChIP-seq datasets from both Tregs and conventional T cells (Schmidl et al., Blood, 2014), FOXP3 only bound the *RTKN2* marker peak in the Tregs. Another example is *CCL2*, a leukocyte-recruiting chemokine, in stromal cells, where the promoter is accessible in sublining and mural cell populations, but the gene is primarily expressed in the inflammatory sublining population S_A-0 (**Supplementary Fig. 4b**). Multiple studies have shown that *CCL2* is induced by TNF and/or IFN γ (Koch et al., J Clin Invest., 1992; Mizoguchi et al., Nat Comm, 2018). Moreover, a recent study by Smith et al., (Nature Immunology, 2023) showed cultured RA tissue fibroblast-like synoviocytes (FLSs) had increased *CCL2* expression after TNF and IFN γ stimulation; *CCL2* expression was strongest in activated sublining populations, which is similar to our S_A-0 population. They also stimulated FLS with TNF, IFN γ and IL-1 β and found *CCL2* protein expression there compared to unstimulated FLS was notably weaker than expected in their CD34+ sublining population, mirroring our S_A-2 population's lack of comparable *CCL2* gene expression. Furthermore, Armaka and colleagues suggested *CCL2* is primed to be expressed in inflammatory fibroblasts, but only after NF κ B regulation (Armaka et al., Genome Med, 2022); we also see an RELA motif, which NF κ B TFs bind, in our *CCL2* marker peak. We also see some examples in the B cells, such as *FCER2*, whose promoter peak is open in the naive B_A-3, unswitched memory B_A-4, and switched memory B_A-2 classes, but whose gene is primarily expressed in the initial naive cells. Similarly, *ITGAX* is expressed mainly in ABC B_A-5, but its peak is open in all memory B cells classes (B_A-4, B_A-2, and B_A-5) (**Supplementary Fig. 6b**).

Addition to **Discussion** [in multiome/superstate paragraphs]:

Furthermore, when we did not see class correspondence between chromatin accessibility and gene expression on the individual gene level, we observed more class-specific gene expression in the context of promiscuous chromatin accessibility. This suggested a poised chromatin state that depends on the presence of a specific TF or extracellular signal to give rise to a particular transcriptional outcome. For example, the promoter peak of *RTKN2* was accessible in all CD4 T cells, but the gene was primarily expressed in Tregs (**Supplementary Fig. 3b**), likely because it is a direct target of the Treg master regulator FOXP3 (Ferraro et al., PNAS, 2014). *CCL2* in stromal fibroblasts had an accessible promoter peak in both sublining populations, but was primarily expressed in the inflammatory subset (**Supplementary Fig. 4b**), likely due to stimulation by TNF/IFN γ (Koch et al., J Clin Invest., 1992; Smith et al., Nature Immunology, 2023).

R2C7. The identification of pathogenic RA chromatin classes is a very important finding from these studies. I would move Supplementary Figure 7 to the main body of the manuscript.

Response: We thank the reviewer for their enthusiastic response to this analysis! We thought it was so important that we repeated this analysis with the RA PBMCs we generated to answer R1C3 in **Supplementary Fig. 15c**. However, Reviewer 3 gave us conflicting advice, as articulated in R3C7.

Therefore, we decided to compromise between the differing opinions and keep this original analysis as a supplementary analysis, now in **Supplementary Fig. 9**.

R2C8. The paper is packed with a lot of very valuable information about these chromatin cell types, their gene accessibility biomarkers, with and without correlated gene expression, and their association with clinical metrics of RA, previously defined RA subtypes, and previously identified RA genetic risk variants that is buried throughout the text. I would strongly recommend that the authors compile all of this useful information into a single summary table that could serve as a reference to the community.

Response: We thank the reviewer for this wonderful suggestion! We have compiled the main results for the requested information types in **Supplementary Tables 9-10**. To get a systematic, data-driven list of biomarkers as suggested here as well as in R2C1, we formalized our differential peak and gene analyses for each chromatin class within a cell type and added the results in **Supplementary Table 9**; we separated them out from the other requested information types, found in **Supplementary Table 10**, to keep both tables readable. Briefly, for the differential peak analysis, we used a logistic model on single cells within a cell type to relate the binarized promoter peak counts of minimal accessibility to their chromatin class, sample, and fragment count. While for the differential gene analysis, we used a Wilcoxon Rank-Sum test to relate the normalized gene counts per cell to their assigned chromatin class. Additionally, we added relevant markers for both peaks and genes to the maker heatmaps in **Supplementary Figs. 3-7b**.

Addition to **Discussion**:

In this study, we described 24 chromatin classes across 5 broad cell types in 30 synovial tissue samples assayed with unimodal scATAC and multimodal snATAC along with TFs potentially regulating them. Based on our observation that cells from the same chromatin class corresponded to multiple transcriptional cell states, we proposed that these chromatin classes are putative superstates of related transcriptional cell states. Finally, we assessed these chromatin classes' relationship to RA clinical metrics, subtypes, and genetic risk variants. **Our main findings are summarized in Supplementary Tables 9-10.**

Addition to **Methods**:

Single cell differential peak analysis. We used a logistic model to determine differential promoter peaks across chromatin class identity. We did this at the single cell level for the combined unimodal scATAC and multimodal snATAC cells and took into account the sample's sample ('sample') and overall fragment counts ('nFragments') as covariates. Genome-wide promoter peaks were defined per cell type as in **T cell lineage analysis**. For each peak and cell type combination, we calculated two logistic regressions using lme4::glmer¹²¹ with a nloptwrap optimizer for speed:

Full model: $\text{peak} \sim \text{cellType} + (1|\text{sample}) + \text{scale}(\log_{10}(\text{nFragments}))$

Null model: $\text{peak} \sim (1|\text{sample}) + \text{scale}(\log_{10}(\text{nFragments}))$

The log₂FC was determined as the cell type beta. We calculated significance as a likelihood ratio test (LRT) between the full and null models with multiple hypothesis test correction using FDR. The top 5 peaks per class, defined as having log₂FC>0.5 and -log₁₀(FDR)>5, ordered by FDR, are shown in **Supplementary Table 9**.

Single cell differential gene analysis. For the multiome cells only, we calculated differentially expressed genes between chromatin class identities within a cell type via a Wilcoxon Rank-Sum test using a normalized gene expression matrix input to presto::wilcoxauc. The top 5 genes per class, defined as having $\log_{2}FC > 0.5$ and $-\log_{10}(FDR) > 5$, ordered by FDR and $\log_{2}FC$, are shown in **Supplementary Table 9**. We selected one peak of potentially multiple that overlapped the annotated gene based on the differential peak's significance in the corresponding class.

Addition to **Methods** [in **Co-varying neighborhood analysis (CNA)** section]:

In **Supplementary Table 10**, clinical metrics and cell type abundance phenotypes (CTAPs) were listed if the median abundance correlation of the AMP-RA reference cells within their Symphony-classified chromatin class was more extreme than the FDR threshold for that patient attribute (Zhang et al., Nature, 2023). Classes were considered significantly expanded if that class' cells were positively correlated with that category's per-sample class abundance within a cell type and depleted if negatively correlated.

Modified **Supplementary Figs.**:

Supplementary Fig. 3b:

Supplementary Fig. 4b:

Supplementary Fig. 5b:

Supplementary Fig. 6b:

Supplementary Fig. 7b:

Additional Supplementary Tables 9-10 [the full Supplementary Table 9 is in Excel]:

Cell Type	Class	Promoter Peak	Gene	peak		gene		T5P	T5G	MK
				log2FC	log10padj	logFC	log10padj			
T cells	TA-0	chr11:122838407-122838607	CRTAM	2.09	Inf	0.78	Inf	X		X
T cells	TA-0	chr2:230877807-230878007	ITM2C	1.68	303.66	0.03	2.44	X		
T cells	TA-0	chr2:86790870-86791070	CD8A	1.57	244.96	0.63	280.54	X		X
T cells	TA-0	chr17:35880348-35880548	CCL5	1.35	230.75	1.94	Inf	X	X	X
T cells	TA-0	chr12:122227507-122227707	DIABLO	1.22	162.75	0.00	0.09	X		
T cells	TA-0	chr7:36724400-36724600	AOAH	0.85	95.14	1.26	Inf		X	
T cells	TA-0	chr14:24609770-24609970	GZMH	1.31	87.92	0.49	203.20			X
T cells	TA-0	chr12:68159701-68159901	IFNG	0.73	72.09	0.48	119.99			X
T cells	TA-0	chr5:55022512-55022712	GZMK	1.06	45.30	1.18	Inf		X	X
T cells	TA-0	chr17:36103644-36103844	CCL4	1.41	39.95	1.64	Inf		X	
T cells	TA-0	chr12:8989442-8989642	KLRG1	1.04	31.35	0.28	52.21			X
T cells	TA-0	chr5:55102504-55102704	GZMA	0.34	7.52	0.91	243.59			X
T cells	TA-0	NA	NKG7	NA	NA	1.09	Inf		X	
T cells	TA-4	chr10:70603489-70603689	PRF1	1.83	119.02	0.31	42.91	X		X
T cells	TA-4	chr2:235669409-235669609	AGAP1	2.41	100.05	0.42	54.11	X		
T cells	TA-4	chr9:81689741-81689941	TLE1	1.98	94.03	0.24	84.41	X		X
T cells	TA-4	chr4:15963229-15963429	FGFBP2	1.99	83.56	0.25	123.20	X		X
T cells	TA-4	chr14:24634266-24634466	GZMB	1.31	77.09	0.26	28.39	X		X
T cells	TA-4	chr7:70693914-70694114	AUTS2	1.04	50.60	0.53	53.31		X	
T cells	TA-4	chr20:8132009-8132209	PLCB1	0.77	30.36	1.05	119.11		X	
T cells	TA-4	chr14:24609770-24609970	GZMH	0.84	17.12	0.33	26.60			X
T cells	TA-4	chr2:86790870-86791070	CD8A	0.64	16.70	0.26	11.21			X
T cells	TA-4	chr9:75890527-75890727	PCSK5	0.59	8.91	0.09	4.10			X
T cells	TA-4	chr12:8989442-8989642	KLRG1	0.52	3.07	0.27	13.38			X
T cells	TA-4	chr8:66613381-66613581	MYBL1	0.06	0.40	0.62	54.71		X	
T cells	TA-4	chr5:55102504-55102704	GZMA	0.02	0.04	0.64	33.30			X
T cells	TA-4	NA	GNLV	NA	NA	0.73	132.91		X	X
T cells	TA-4	NA	NKG7	NA	NA	0.84	66.01		X	
T cells	TA-1	chr9:75890527-75890727	PCSK5	1.55	165.37	0.14	46.31	X		X
T cells	TA-1	chr12:56338708-56338908	IL23A	1.00	127.68	0.04	5.68	X		
T cells	TA-1	chr3:42502460-42502660	VIPR1	0.75	104.00	0.03	7.66	X		
T cells	TA-1	chr19:19606050-19606250	PBX4	0.95	102.34	0.22	19.50	X		
T cells	TA-1	chr1:220690190-220690390	C1orf115	1.19	92.39	0.00	0.19	X		
T cells	TA-1	chr9:73151703-73151903	ANXA1	0.82	50.36	0.81	148.75		X	X
T cells	TA-1	chr1:207321460-207321660	CD55	0.43	38.07	0.65	123.38		X	
T cells	TA-1	chr5:35856728-35856928	IL7R	0.52	27.89	1.09	290.49		X	X
T cells	TA-1	chr17:40565375-40565575	CCR7	0.42	26.11	0.18	22.56			X
T cells	TA-1	chr6:90296806-90297006	BACH2	0.36	22.20	0.77	123.42		X	
T cells	TA-1	chr10:60733393-60733593	ANK3	0.39	15.04	0.88	141.58		X	
T cells	TA-1	chr12:6789386-6789586	CD4	0.38	9.66	0.00	0.00			X
T cells	TA-2	chr5:55978070-55978270	IL6ST	1.83	266.30	0.99	204.35	X		X
T cells	TA-2	chr4:107824763-107824963	SGMS2	2.42	255.19	0.17	98.47	X		
T cells	TA-2	chr4:86594169-86594369	PTPN13	1.57	222.38	1.46	Inf	X	X	X
T cells	TA-2	chr6:135851541-135851741	PDE7B	2.00	216.88	1.44	Inf	X	X	X
T cells	TA-2	chr20:52972288-52972488	TSHZ2	1.88	214.11	2.23	Inf	X	X	X
T cells	TA-2	chr5:143434694-143434894	NR3C1	1.04	80.67	1.53	Inf		X	X
T cells	TA-2	chr12:6789386-6789586	CD4	0.99	57.66	0.40	113.69			X
T cells	TA-2	chr2:203867644-203867844	CTLA4	0.86	53.88	0.84	296.15			X
T cells	TA-2	chr2:241859806-241860006	PDCD1	0.77	51.68	0.17	48.30			X
T cells	TA-2	NA	CXCL13	NA	NA	1.02	Inf		X	X
T cells	TA-3	chr19:50329219-50329419	NR1H2	1.02	83.15	0.02	0.25	X		
T cells	TA-3	chr11:111540572-111540772	LAYN	2.05	72.84	0.10	66.33	X		
T cells	TA-3	chr13:109786543-109786743	IRS2	0.86	67.47	0.24	23.89	X		
T cells	TA-3	chr12:68933152-68933352	CPM	0.89	65.85	0.12	8.77	X		
T cells	TA-3	chr12:113135767-113135967	RASAL1	1.15	58.10	0.01	3.26	X		
T cells	TA-3	chr2:213151549-213151749	IKZF2	0.98	42.79	1.06	217.50		X	X
T cells	TA-3	chr2:95025503-95025703	MAL	0.59	10.27	0.10	9.59			X
T cells	TA-3	chr12:6789386-6789586	CD4	0.38	4.31	0.09	3.60			X
T cells	TA-3	chr1:169711607-169711807	SELL	0.33	2.27	0.18	22.89			X
T cells	TA-3	chr10:6062340-6062540	IL2RA	0.19	1.52	0.33	53.75			X
T cells	TA-3	chr17:40565375-40565575	CCR7	-0.14	1.18	0.11	2.51			X
T cells	TA-3	chr10:62268653-62268853	RTKN2	0.12	0.84	0.60	131.76		X	X
T cells	TA-3	chr10:17644005-17644205	STAM	0.04	0.22	0.73	133.97		X	
T cells	TA-3	NA	FOXP3	NA	NA	0.17	105.79			X
T cells	TA-3	NA	AC093865.1	NA	NA	0.74	184.78		X	
T cells	TA-3	NA	AL136456.1	NA	NA	0.97	134.40		X	

Supplementary Table 9. Markers for each chromatin class. The top 5 peaks ('T5P') from a logistic regression model relating class to binary peak distributions over all ATAC cells (Methods), with at least $\log_2FC > 0.5$ and a log-likelihood ratio test $-\log_{10}(FDR) > 5$, ordered by

FDR, are shown. The top 5 genes ('T5G') from a Wilcoxon Rank Sum Test via presto relating class to normalized gene expression over multiome cells (**Methods**), with at least $\log_{2}FC > 0.5$ and an $-\log_{10}(\text{adjusted p-value}) > 5$, ordered by adjusted p-value and $\log_{2}FC$, are shown. We also show some chosen biological markers ('MK'). There are NAs for the peak-associated columns if there was not a promoter peak in our set associated with that gene.

Cell type	Chromatin class	TF motifs	Corresponding transcriptional cell states	RA CNA associations	RA risk variants
T cells	TA-0: CD8A+ GZMK+	EOMES TBX21 IRF4	T-5: CD4+ GZMK+ memory T-13: CD8+ GZMKB+ memory T-14: CD8+ GZMK+ memory T-22: Vdelta1		rs11209051 rs798000
	TA-4: CD8A+ PRF1+ cytotoxic	EOMES TBX21 RUNX3	T-12: CD4+ GNLY T-15: CD8+ GZMB+/TEMRA T-21: Innate-like T-23: Vdelta2	Lymphoid Aggregates (-) Kiern (-) CTAP-TB (-)	rs11209051 rs734094 rs911760
	TA-1: CD4+ IL7R+	TCF7 LEF1 GATA3	T-0: CD4+ IL7R+ memory T-1: CD4+ CD161+ memory T-4: CD4+ naive T-16: CD8+ CD45ROlow/naive		rs798000 rs911760 rs3784099
	TA-2: CD4+ PD-1+ TFH/TFH	BATF JUN FOS	T-3: CD4+ Tfh/Tph T-7: CD4+ Tph T-10: CD4+ OX40+ NR3C1+ T-11: CD4+ CD146+ memory	Kiern (+) CTAP-TB (+)	rs11209051 rs798000 rs3784099 rs7441808
	TA-3: CD4+ IKZF2+ Treg	KLF10 SP3	T-8: CD4+ CD25-high Treg T-9: CD4+ CD25-low Treg		rs11209051 rs798000
Stromal cells	SA-1: PRG4+ lining	FOS JUND BATF	F-0: PRG4+ CLIC5+ lining F-1: PRG4+ lining	CTAP-F (+) CTAP-M (-)	rs72798636
	SA-2: CD34+ MFAP5+ sublining	NFATC4 NFATC3	F-2: CD34+ sublining F-4: DKK3+ sublining F-7: NOTCH3+ sublining	CTAP-F (+)	rs7441808
	SA-0: CXCL12+ HLA-DRhi sublining	TEAD1 STAT1 STAT3	F-6: CXCL12+ SFRP1+ sublining F-5: CD74hiHLAhi sublining F-3: POSTN+ sublining F-8: RSP03+ intermediate		
	SA-3: MCAM+ mural	KLF2 EBF1	Mu-0: Mural	CTAP-F (-) CTAP-M (+) CTAP-TF (-)	
Myeloid cells	MA-0: F13A1+ MARCKS+ TRM	RELA SREBF2	M-1: MERTK+ LYVE1- M-9: DC3 M-8: PLCG2+ M-5: C1QA+ M-6: STAT1+ CXCL10+		rs9927316
	MA-2: LYVE1+ TIMD4+ TRM	KLF4 NFX NFATC3	M-0: MERTK+ LYVE1+ M-2: MERTK+ S100A8+	Lymphoid Density (-) CTAP-EFM (+) CTAP-F (+)	rs9927316
	MA-4: SPP1+ FABP5+ intermediate	JUN FOS	M-3: MERTK+ HBEGF+ M-4: SPP1+	Lymphoid Density (+) Lymphoid Aggregates (+) Kiern (+) CTAP-M (+)	rs9927316
	MA-1: FCN1+ SAMS1+ infiltrating monocytes	JUN FOSL1 CEBPD	M-7: IL1B+ FCN1+ HBEGF+ M-11: CD16+ DC4	Lymphoid Aggregates (-) CTAP-EFM (-)	rs9927316 rs734094
	MA-3: CD1C+ AFF3+ DC	SPI1 IRF1 STAT1/2	M-10: DC2 M-12: DC1 M-14: LAMP3+		rs9927316 rs734094
B/plasma cells	BA-3: FCER2+ IGHG+ naive B	SPIB SPI1	B-2: IgM+IgD+TCL1A+ naive	Lymphoid Aggregates (-)	rs663743 rs3784099 rs4840568
	BA-4: CD24+ MAST4+ unswitched memory B	NFKB1 ZBTB7A	B-3: IgM+IgD+CD1c+ MZ-like B-1: CD24hiCD27+IgM+ unswitched memory		rs911760 rs663743 rs3784099 rs4840568
	BA-2: TOX+ PDE4D+ switched memory B	ETS1 RELA	B-0: CD24+CD27+CD11b+ switched memory	CTAP-TB (+)	rs663743 rs3784099 rs4840568
	BA-5: ITGAX+ ABC	TBX19	B-5: CD11c+LAMP1+ ABC		rs911760 rs663743 rs4840568
	BA-0: CREB3L2+ plasma	KLF2 SP3	B-7: HLA-DR+IgG+ plasmablast B-8: IgG1+IgG3+ plasma	CTAP-TB (-)	
	BA-1: CD27+ plasma	BATF JUN	B-6: IgM+ plasma	CTAP-TB (-)	rs911760
Endothelial cells	EA-2: SEMA3G+ arteriolar	SOX17	E-3: NOTCH4+ arteriolar		rs71508903 rs72798636
	EA-3: PROX1+ lymphatic		E-4: Lymphatic	CTAP-F (+) CTAP-M (-)	rs71508903
	EA-0: SELP+ venular	STAT3 NFIB	E-1: LIFR+ venular E-2: ICAM1+ venular	CTAP-F (+)	rs71508903 rs72798636
	EA-1: RGCC+ capillary	FOS JUN	E-0: SPARC+ capillary	CTAP-F (-) CTAP-M (+)	rs71508903 rs72798636

Supplementary Table 10. Selected main results per chromatin class: TF motifs, transcriptional cell states, RA CNA associations, and RA risk variants. In column “RA CNA associations,” clinical metrics and cell type abundance phenotypes (CTAPs) were listed if the AMP-RA reference cells within their Symphony-classified chromatin class were significantly expanded (+) or depleted (-) in association with that patient attribute (**Methods**).

Reviewer #3 (expert in computational biology, regulation of gene expression and epigenomics):

This study presents a comprehensive analysis of the chromatin landscape of synovial tissues from RA patients at single cell levels using scATAC-seq and multiomic analyses. The data are highly valuable to the community and timely needed for studying RA pathogenesis. Clustering analysis defined 6 broad cell types and 24 chromatin classes were further uncovered within 5 of these cell types, from which marker genes and motifs were uncovered. These chromatin classes were compared with the AMP RA clusters defined from single cell transcriptomic studies and each chromatin class corresponded to multiple transcriptomic classes. The authors therefore referred to the chromatin classes as superstates corresponding to multiple transcriptional cell states. This is an interesting but not surprising observation and this superstate hypothesis needs additional validations.

Overall, this is an interesting study and the authors may want to consider the following points to improve the manuscript. (1) As a data driven study, the scATAC-seq, multiome and CITE-seq data from previous study were analyzed in this work. The clarity of the description on data analysis and integration needs to be improved and workflow chart is recommended to summarize the procedure and elucidate the logic. (2) The superstate hypothesis is intriguing but additional validations are needed.

Detailed comments are the following.

Response: We thank the reviewer for their positive feedback and constructive criticisms.

R3C1. As the large dataset is a valuable resource to the community, it would be helpful to have a summary table of QC such as total reads, mapping rate, percentage of reads falling in peak neighborhoods, reads in promoters, mitochondrial reads, reads falling in the blacklisted regions for the final selected cells.

Response: We thank the reviewer for this excellent suggestion.

We have included a per-cell count of total reads, reads falling in peak neighborhoods, reads in promoters, mitochondrial reads, and reads falling in the blacklisted regions for the final selected cells in two metadata tables (one per ATAC assay) uploaded to SAGE (accession IDs syn53642004 and

syn53642005). Since that was a rather large table, we have also included a mean aggregated version by ATAC assay and cell type as **Supplementary Table 2**. We excluded mapping rate as a criterion because the ATAC read QC excluded all non-mapped reads, resulting in 100% mapped reads at the point of final cell selection.

Addition to Results [QC paragraph]:

Applying stringent ATAC quality control, we retained cells with >10,000 reads, >50% of those reads falling in peak neighborhoods, >10% of reads in promoter regions, <10% of reads in the mitochondrial chromosome, and <10% of reads falling in the ENCODE blacklisted regions²⁴ (**Methods; Supplementary Figs. 1a-b, 2a-b; Supplementary Table 2**).

Additional Supplementary Table:

ATAC Assay	Cell Type	Pre-duplication read counts	Deduplicated read counts	Peak 'neighborhood' read counts	Promoter read counts	Mitochondrial read counts	Blacklisted region read counts
unimodal	Bplasma	38,620	24,476	18,741	8,131	829	1,533
unimodal	endothelial	51,628	31,231	21,040	8,326	1,481	1,775
unimodal	myeloid	38,241	25,243	19,533	6,927	345	1,079
unimodal	NK	41,180	26,042	21,002	9,194	576	1,249
unimodal	stromal	48,412	30,396	20,299	6,943	746	1,843
unimodal	Tcell	40,580	25,079	19,955	8,932	824	1,401
multimodal	Bplasma	124,867	46,437	36,592	15,939	1,336	5,021
multimodal	endothelial	109,231	45,591	31,673	12,514	1,465	4,085
multimodal	myeloid	139,874	58,182	44,318	15,461	1,026	4,085
multimodal	NK	137,460	48,150	38,121	16,850	1,265	4,554
multimodal	stromal	142,374	61,606	42,097	14,588	754	5,811
multimodal	Tcell	131,795	44,828	36,082	16,450	1,264	4,666

Supplementary Table 2. Mean quality control metrics for ATAC cells segregated by ATAC assay and cell type.

R3C2. Access ID for raw and processed data should be provided.

Response: We thank the reviewer for this comment. The raw FASTQs for the unimodal scATAC data and all the processed data can be found on Synapse under accession ID syn53650034 (<https://doi.org/10.7303/syn53650034>). The raw FASTQs for the multimodal snATAC and snRNA for the synovial tissue and superstate PBMC datasets are on dbGaP under accession ID phs003417.v1.p1 (https://www.ncbi.nlm.nih.gov/projects/gap/cgi-bin/study.cgi?study_id=phs003417.v1.p1). The different repositories are due to different funding sources.

Addition to Methods [in Data Availability section]:

The raw FASTQs for the unimodal scATAC data and all the processed data can be found on Synapse under accession ID syn53650034. The raw FASTQs for the multimodal snATAC and snRNA for the synovial tissue and superstate PBMC datasets are on dbGaP under accession ID phs003417.v1.p1.

R3C3. While the consensus open chromatin peaks of scATAC-seq are called from all the cells pooled to a bulk, how are peaks called for snATAC-seq? Line 797-798, "an average of 75% (n=12 datasets; range: 66%-83%) of the 200bp

trimmed snATAC-seq donor-specific peaks overlapping the scATAC-seq consensus peaks”, how are donor-specific snATAC-seq peaks called? More details need to be provided.

Response: We thank the reviewer for calling attention to this under-explained methodological detail. We called peaks the same way for the unimodal scATAC-seq consensus peaks across all 18 samples as the multiome snATAC-seq sample-specific peaks, as we hope is now made clear in the updated text. To further clarify, the only analysis we did with the multiome snATAC-seq sample-specific peaks was the overlap statistic the reviewer mentioned. We wanted to reassure ourselves that the unimodal scATAC-seq consensus peaks were a valid representation of the multiome samples, as we assumed they likely would be since all samples used here were from synovial tissue. We preferred to use the consensus peaks for all downstream analyses, as we were already confident about those peaks called on more cells from more synovial tissue samples. Also, we changed donor-specific to sample-specific, since peaks were called on samples from donors, not donors themselves; each donor contributed one sample.

Addition to Methods [in ATAC peak calling section]:

We wanted to confirm that these unimodal scATAC-seq consensus peaks were reasonable to use for the multiome snATAC-seq datasets, beyond just that the datasets were done on the same tissue type. Therefore, we called peaks, as done above, on the individual sample multimodal snATAC-seq BAM files and found that an average of 75% (n=12 samples; range: 66%-83%) of the 200bp trimmed multimodal snATAC-seq sample-specific peaks overlapped the unimodal scATAC-seq consensus peaks. Furthermore, we used the 5x full consensus peak neighborhoods in the cell QC step for multiome datasets as an added safeguard.

R3C4. “Broad cell type clustering” in Methods does not really describe the procedure. Initial broad cell type clustering is mentioned in “ATAC quality control”. Is there any further broad cell type clustering after the initial one? It’d better present a summary of the procedure with a workflow chart to help readers understand how the open chromatin data are clustered to six cell types.

Response: We appreciate the reviewer calling attention to a confusing part of our Methods section and the great suggestion of a workflow chart. We added the workflow chart as the new **Supplementary Fig. 1** and reorganized the **Methods** section to better correspond to it, by: adding an initial Computational methods section to introduce the workflow chart; splitting up ATAC quality control into ATAC read QC, ATAC cell QC, ATAC clustering, and ATAC doublet cluster removal; reordering the ATAC peak calling section; splitting up RNA quality control into RNA cell QC, RNA clustering, and RNA doublet cluster removal; and moving Symphony classification of transcriptional cell state, now Symphony classification of transcriptional identity, before broad cell type clustering. We kept broad cell type clustering as a section since we needed to discuss the extra filters in the multiome datasets that span both ATAC and RNA clustering, which the new details added to that section and the workflow chart hopefully now clarify.

Specifically, for each modality, we did an initial round of clustering on all post cell-QC cells to determine which clusters were likely doublets. We did this by cluster for each modality (mRNA/ATAC) looking at

four criteria: (1) multiple cell-type-specific marker genes/peaks, (2) intermediate placement between broad cell type clusters in principal component space, (3) high UMI/fragment counts, and (4) high doublet scores determined per cell per sample by Scrublet (Wolock et al., Cell Syst, 2019)/ArchR (Granja et al., Nat Genet, 2021). We do this by cluster, not by cell, as even if an individual cell (e.g., cell A) may not be classified as a doublet using these markers, if it clusters strongly with many other cells that look like doublets, cell A is also likely to be a doublet (Wolock et al., Cell Syst, 2019). Then, using only the non-doublet cells, we do another round of clustering to define the final broad cell type annotations using marker genes/peaks, aka broad cell type clustering. In both rounds of clustering described here, the method per modality is the same; however, the cells being clustered get subsetted between rounds.

Addition to Results [first paragraph]:

Applying stringent ATAC quality control, we retained cells with >10,000 reads, >50% of those reads falling in peak neighborhoods, >10% of reads in promoter regions, <10% of reads in the mitochondrial chromosome, and <10% of reads falling in the ENCODE blacklisted regions²⁴ (**Methods; Supplementary Figs. 1a-b, 2a-b; Supplementary Table 2**). We further required that cells from the multimodal data passed **quality control for the snRNA modality** (**Methods; Supplementary Figs. 1b, 2c**). After additional QC within individual cell types combining both technologies, the final dataset contained 86,994 cells from 30 samples (median of 2,990 cells/sample) (**Supplementary Figs. 1c-d, 2d-e**).

Addition to Methods:

Computational methods. **Supplementary Fig. 1** shows an overview of the computational methodology for cell type/state identification, as many of the methods were reused in different contexts. In the following sections, we explain the core methodology the first time it is used, and then only the ways in which the methodology differs in the different contexts afterwards.

We have not reproduced the fully restructured **Methods** sections here, for the sake of brevity. Please see the revised manuscript.

Additional Supplementary Fig. 1:

Supplementary Fig. 1. Computational methods workflows for cell type, chromatin class, and transcriptional cell state annotations.

a. Unimodal scATAC (light blue) workflow from 10x Genomics Cell Ranger ATAC 1.1.0 to broad cell type clustering.

b. Multiome (purple) workflow for both snATAC (dark blue) and snRNA (red) from 10x Genomics Cell Ranger ARC 2.0.0 to broad cell type clustering. In addition to calling broad cell types within each modality, we used the non-doublet multimodal snRNA cells as a query dataset to map onto the AMP-RA synovial tissue CITE-seq broad cell type reference¹⁴ using Symphony⁹⁷. We removed any cells whose broad cell types did not match for all three annotations (snATAC broad cell type, snRNA broad cell type, and classified AMP-RA broad cell type).

c. Chromatin class (middle blue) workflow from combining unimodal scATAC and multimodal snATAC at the broad cell type level to chromatin class clustering.

d. Transcriptional cell state workflow using multimodal snRNA as a query to map onto the AMP-RA synovial tissue CITE-seq references¹⁴ using Symphony⁹⁷. This is done for each cell type using the cells with chromatin class annotations.

R3C5. scATAC-seq and snATAC-seq data are combined in each broad cell type to define fine-grain chromatin classes within the cell type. Line 841-842, "After subsetting the matrix by PMA peaks, we ran the same clustering pipeline detailed in the broad cell type clustering section with 10 PCs requested." There is no detailed description of the pipeline in the broad cell type clustering section with 10 PCs. Are all the cells from scATAC and snATAC pooled to call consensus peaks for clustering? How to decide the number of clusters (not much discussion on Supplementary Fig. 10)? Need to clarify and provide details.

Response: We thank the reviewer for this comment.

We have restructured the methods section such that there is now an **ATAC clustering** section, which defines how we clustered peaks x cells matrices. To construct each cell type's input peaks x cells matrix for chromatin class clustering, we: 1) concatenated the unimodal scATAC-seq and multimodal snATAC-seq cells into the same matrix using the consensus peak set; 2) split up that matrix by each cell type's cells; 3) determined each cell type's peaks with minimal accessibility (PMA); 4) subset the matrix by PMA peaks. For each cell type, that subsetted peaks x cells matrix was used as input into our **ATAC clustering** pipeline, where the only change is that we asked for 10 PCs instead of 20 PCs as there were notably fewer cells with less expected variation per cell type as opposed to all cell types combined.

We did not re-call peaks on the pooled scATAC and snATAC cells per cell type. We decided to use a consensus peak set across downstream analyses for consistency, interpretability, and confidence in high-quality peak calls across many cells.

As we originally said in our manuscript about the new **Supplementary Fig. 13** (old **Supplementary Fig. 10**):

We tried a number of clustering resolutions (see **Supplementary Fig. 13** for a subset) and chose the resolution at which we could define clusters biologically with known markers that tracked in both chromatin accessibility and gene expression spaces.

By biologically meaningful clusters, we meant clustering resolutions that largely respected the gene expression and promoter peak chromatin accessibility of known cell-state-specific markers, such as *PRF1* in T_A-4: CD4⁺ PRF1⁺ cytotoxic (**Fig. 2b**) or *SPP1* in M_A-4: SPP1⁺ FABP5⁺ intermediate (**Fig. 4b**). If we cluster even one more cluster deep, those markers and the identities more generally are split over too many clusters, like T cell clusters 4 and 5 in **Supplementary Fig. 13a, left** and myeloid clusters 4 and 0 in **Supplementary Fig. 13c, left**.

Addition to Methods:

ATAC clustering. We did multiple rounds of clustering with different inputs. Generally, we did: binarize peaks x cells matrix, log(TF_xIDF) normalization using Seurat::TF.IDF¹³³, most variable peak feature selection using Symphony::vargenes_vst⁹⁷, center/scale features to mean 0 and variance 1 across cells using base::scale, PCA dimensionality reduction using irlba::prcomp_irlba, batch correction by sample using Harmony::HarmonyMatrix²⁷, shared nearest neighbor creation using RANN::nn2 and Seurat::ComputeSNN¹³³, Louvain clustering using Seurat::RunModularityClustering¹³³, and cluster visualization using UMAP coordinates via umap::umap. For the unimodal scATAC-seq feature selection, we chose peaks that had at least one fragment in at least five percent of cells and TF_xIDF normalization using Seurat::TF.IDF¹³³ before continuing in the above steps. We used 20 PCs for the broad cell type clustering and 10 PCs for the chromatin class clustering since there was less variation within a cell type.

Fine-grain chromatin class clustering. To define chromatin classes within broad cell types (**Supplementary Fig. 1c**), we made peaks x cells matrices for each broad cell type concatenating unimodal scATAC-seq and multimodal snATAC-seq cells of that type across the consensus peaks. Since peaks were called on all unimodal scATAC-seq cells regardless of cell type, we first subset each consensus peaks x broad cell type cells matrix by "peaks with minimal accessibility" (PMA). We defined minimal accessibility as consensus peaks that had a fragment

in at least 0.5% of cells of that type, except for endothelial cells which we increased to a minimum of 50 cells. After subsetting the matrix by PMA peaks, we ran the same clustering pipeline detailed in ATAC clustering. For endothelial cells, due to small cell counts, we batch-corrected on both sample and assay and updated Harmony's sigma parameter to 0.2. We did another round of QC to exclude cells that clustered primarily due to relatively fewer total fragments per cell and fewer peaks with at least one 1 fragment per cell, and then re-clustered. We tried a number of clustering resolutions (see Supplementary Fig. 13 for a subset) and chose the resolution at which known cell-state-specific markers' promoter peak chromatin accessibility and gene expression largely respected cluster boundaries, such as PRF1 in T_A-4: CD4+ PRF1+ cytotoxic (Fig. 2b) or SPP1 in M_A-4: SPP1+ FABP5+ intermediate (Fig. 4b).

R3C6. Line 828-831, "We also classified the multiome snRNA cells into the AMP-RA CITE-seq study¹² broad cell types using Symphony (see Symphony classification of transcriptional cell state). The small minority of cells (2%) with discordant cell types defined in the snATAC, snRNA, and CITE seq modalities for the multiome datasets were removed.". snRNA-seq were classified to the broad cell types defined by AMP-RA CITE-seq study using Symphony.

Response: We thank the reviewer for this comment, though it might have gotten cutoff. We assume it was asking for more clarification on the broad cell type annotation using the AMP RA tissue CITE-seq reference. We have moved the **Methods** section **Symphony classification of transcriptional cell state**, renamed **Symphony classification of transcriptional identity**, before broad cell type clustering and clarified that the same procedure was used for the broad cell type and fine-grain cell state Symphony references. We also clarified that the non-doublet cells that passed cell QC were used to do the Symphony classification of the broad cell types in both the **Methods** text and **Supplementary Fig. 1b**.

Addition to Methods:

Symphony classification of transcriptional identity. To determine the RA transcriptional cell types/states within our multimodal data, we used Symphony⁹⁷ to map the multimodal snRNA profiles into the AMP-RA reference synovial tissue transcriptional cell types/states¹⁴ (Supplementary Fig. 1b,d). We used one Symphony reference object from that study for the broad cell types together and one for each broad cell type we tested (T cell, stromal, myeloid, B/plasma, and endothelial) for the fine-grain cell state identities. The broad cell types and lymphocyte states were defined using both gene and surface protein expression while the others were defined using gene expression only. In each case, we mapped the multimodal snRNA gene x cells matrix into the appropriate Symphony reference object using the mapQuery function, accounting for sample as a batch variable. Using the knnPredict function with k=5, each multiome cell was classified into a reference transcriptional cell type/state by the most common annotation of its five nearest AMP-RA reference neighbors in the harmonized embedding. We considered it a high confidence mapping if at least 3 out of the 5 nearest reference neighbors were the same cell type/state, though the number of cell types/states will affect this as more cell types/states means more boundary regions between cell types/states.

Broad cell type clustering. For non-doublet cells passing cell QC, we subsetted the feature x cells matrices and performed broad cell type clustering within modalities as described above in **ATAC clustering** for the unimodal scATAC and multimodal snATAC datasets separately and **RNA clustering** for the multimodal snRNA datasets (**Supplementary Fig. 1a-b**). We also classified the multiome snRNA cells into the AMP-RA CITE-seq study¹⁴ broad cell types using Symphony⁹⁷ (see **Symphony classification of transcriptional identity**). The small minority of cells (2%) with discordant cell types defined in the snATAC, snRNA, and CITE-seq modalities for the multiome datasets were removed (**Supplementary Fig. 1b**).

R3C7. It is weird to compare the RA tissue with healthy PBMC. It is hard to draw any solid conclusion because the tissue chromatin classes can be due to the difference between synovial tissue and blood.

Response: We concur with the reviewer that there were competing contrasts in the RA tissue and healthy PBMC analysis, namely the tissue and blood comparison and the RA and healthy comparison, that defied solid conclusions. We have reworded the original caveat in the results section to say that in a more straightforward manner.

Addition to Results:

However, there were some tissue chromatin classes that did not have clear counterparts in PBMCs, such as T_A-2: CD4+ PD-1+ TFH/TPH, M_A-2: LYVE1+ TIMD4+ TRM, M_A-4: SPP1+ FABP5+ intermediate, and B_A-5: ITGAX+ ABC (**Supplementary Fig. 9**). **With the current dataset, we cannot conclusively determine whether these disparities reflect tissue and blood or RA and healthy differences.**

We were expecting some differences to come from the synovial tissue and blood comparison since it is known that some populations are enriched in tissue more so than blood, as referenced in the original text. But, to our knowledge, the differences in chromatin accessibility had not been articulated. The chromatin classes that did not have blood counterparts were those populations that are known to be tissue-enriched, but as they were also the populations implicated in RA pathogenesis, it is still not entirely clear if the difference being highlighted here is tissue vs blood or RA vs healthy. In either case, the conclusion that further study of these populations should be done in diseased tissue is valid.

Addition to Results:

However, prior studies have shown both that these cell states are tissue-enriched^{12,95,96} and implicated in RA pathogenesis^{11-13,16,61}, suggesting that the study of disease tissue is necessary for well-powered analyses of these populations.

To remove one of the contrasts, a better comparison would be RA tissue and RA PBMCs. In light of that, we redid this *de novo* ATAC combined clustering analysis using RA tissue and sorted RA PBMCs generated as part of the superstate experiment for R1C3. We saw general concordance with the healthy PBMC analysis done originally (new **Supplementary Fig. 9**). As noted in R1C3, the RA tissue/PBMCs TFH/TPH and RA tissue/PBMC Tregs were largely grouped within the same combined clusters. However, the relatively small proportions of TFH/TPH cells sorted from 4 RA PBMC samples (**Supplementary Fig. 15a**) suggests that synovial tissue is a better source of these cells.

Given Reviewer 2's opposite opinion in their seventh comment, we decided to compromise between the differing opinions and keep the original analysis as a supplementary analysis.

R3C8. It is common that scRNA-seq identifies more clusters than scATAC-seq. For the "superstate" found from scATAC-seq data that represent multiple transcriptional states, is it possible that it is due to the different coverages of scRNA-seq and scATAC-seq? To rule out this possibility and validate the superstate hypothesis, cells in the same superstate need to be sorted out and their transcriptomic profiles need to be analyzed at single cell level.

Response: We thank the reviewer for this really important recommendation. With the proposed superstate model being such a central conclusion within our paper, we were very excited to experimentally validate it. Based on our results, we consider it unlikely that our observations were primarily driven by technical factors.

Using valuable PBMC samples from four RA patients and FACS of surface protein markers, we sorted four populations spanning two chromatin classes and four transcriptional states: CD4⁺CD127⁻CD25^{hi} Treg, CD4⁺CD127⁻CD25^{int} Treg, CD4⁺CD25⁻PD1⁺CXCR5⁺ TFH, and CD4⁺CD25⁻PD1⁺CXCR5⁻ TPH. We then isolated nuclei and hashtagged each population before pooling for a single multiome experiment, thus allowing us to get ATAC and RNA information for each gold standard sorted population via FACS in a cost-efficient manner. We also used this data to address R1C3.

Additional Supplementary Fig. 15:

Legend: Supplementary Fig. 15. Multiome experimental support for the hypothesized superstate model using RA PBMCs sorted for Treg, TFH, TPH populations via FACS.

a. FACS plots of pooled PBMCs from 4 RA patients sorted for: CD4⁺CD127⁻CD25^{hi} Treg, CD4⁺CD127⁻CD25^{int} Treg, CD4⁺CD25⁻PD1⁺CXCR5⁺ TFH, and CD4⁺CD25⁻PD1⁺CXCR5⁻ TPH.

b. Quality control steps ending in the final cell counts for the FACS cell state hashtags in snATAC (**left**) and snRNA (**right**). allQC refers to cells passing ATAC, RNA, and HTO quality control measures (**Methods**).

We used this strategy since it is not possible to directly sort superstates as they are defined via chromatin accessibility peaks. However, based on our analysis comparing chromatin profiles of established protein markers to our superstates in R1C3, we feel that we have sorted a good approximation of the same superstate. Additionally, we used RA PBMCs instead of RA tissue because tissue biopsies are rare and logistically challenging to acquire. Moreover, we found concordance between tissue and blood Tregs in our analysis in **Supplementary Fig. 9**.

We also wanted to assess whether the two cell states within a chromatin class defined via cell surface proteins (e.g., CD4⁺CD25⁺PD1⁺CXCR5⁺ TFH and CD4⁺CD25⁺PD1⁺CXCR5⁻ TPH) were transcriptionally distinct. By clustering the cells from the four sorted populations based on gene expression, we successfully distinguished between the pairs of transcriptomic states from each chromatin class (**Supplementary Fig. 15d**). Moreover, we observed that each gold-standard FACS-defined population had a distinct mRNA cluster identity.

Addition to **Supplementary Fig. 15**:

d. Clustering sorted RA multimodal PBMC snRNA cells visualized on UMAP (**left**) and the natural log of the Odds Ratio between these clusters and the RA PBMC FACS determined cell states (**right**). Non-significant (FDR>0.05) OR values are white. The colors of the y-axis on the **right** correspond to RA PBMC FACS determined cell states as in the UMAPs on the **left**.

Next, we assessed whether the two transcriptional cell states within a chromatin class had differential features in each modality. We calculated differential gene expression using a Wilcoxon test and differential promoter peak accessibility using logistic regression, accounting for number of fragments within cells. While we found many differentially expressed genes between transcriptional cell states, we largely did not observe similar differential accessibility of those genes' promoter peaks (**Fig. 7d-e**). We saw this phenomenon in both the Treg superstate and the TFH/TPH superstate, the latter of which having no significant differentially accessible promoter peaks at FDR=0.10. This finding corroborates our proposed superstate model of distinct transcriptional cell states sharing similar open chromatin landscapes.

Addition to **Fig. 7**:

Legend: **d-e.** Using genes and promoter peak pairs with at least minimal signal, the Wilcoxon FDR of normalized gene expression (x-axis) and the logistic regression FDR of binary promoter peak accessibility (y-axis) between (**d.**) RA PBMC CD25^{hi} and CD25^{int} Treg populations (n=7,208 pairs) and (**e.**) RA PBMC TFH and TPH populations (n=5,264 pairs) (**Methods**). Color was determined by the state with the higher gene expression and the shape denotes whether the state with the higher accessibility agreed. The dotted lines correspond to FDR=0.10, calculated separately within modalities.

For example, the *PDE4D* gene, inhibitors for which are used as a RA treatment (McCann et al., Arthritis Research & Therapy, 2010), had significantly higher expression in TPH than TFH cells (unadjusted P=4.64e-19), but a non-significant change in the promoter peak accessibility (unadjusted P=0.913) (**Supplementary Fig. 15e**). On the other hand, *ZBTB10*, a telomere associated transcription factor (Bluhm et al., NAR, 2019), was a rare example where the chromatin accessibility and gene expression concurred across Treg states (**Supplementary Fig. 15f**). However, globally, the lack of these examples contributed to the lack of 1-1 concordance between transcriptional cell states and chromatin classes.

Addition to **Supplementary Fig. 15:**

e.-f. Normalized gene expression (**left**) and proportion of cells with or without chromatin accessibility in the promoter peak (**right**) for (**e.**) *PDE4D*, segregated by RA PBMC TFH and TPH populations and (**f.**) *ZBTB10*, segregated by RA PBMC CD25^{hi} and CD25^{int} Treg populations. Nominal Wilcoxon (**left**) and logistic regression (**right**) P-values above (**Methods**).

Addition to **Results** [last paragraph in superstate section; including R1C3 response]:

Finally, to more thoroughly investigate the validity of the chromatin superstate model, we profiled the chromatin accessibility and transcriptomes of select cell states known to be functionally distinct and defined by well-characterized surface markers (Rao et al., *Nature*, 2017; Bonelli et al., *J Immunol*, 2009). We generated a multiome dataset of sorted RA PBMC subsets via FACS of four populations spanning two chromatin classes and four transcriptional states: CD4⁺CD127⁻CD25^{hi} Treg, CD4⁺CD127⁻CD25^{int} Treg, CD4⁺CD25⁺PD1⁺CXCR5⁺ TFH, and CD4⁺CD25⁺PD1⁺CXCR5⁻ TPH (**Supplementary Fig. 15a; Methods**). We performed quality control steps in all three modalities and identified FACS cell state labels before doing any downstream analysis (**Supplementary Fig. 15b; Methods**). When we *de novo* clustered the ATAC modalities of the combined PBMC and tissue cells (**Supplementary Fig. 15c; Methods**), we found that the sorted RA PBMC TFH/TPH cells were most enriched in combined cluster 2 (OR=4), which was most highly enriched for RA tissue TFH/TPH cells (OR=32). Similarly, sorted RA PBMC Tregs were most enriched for combined cluster 4 (OR=3), which was most highly enriched for RA tissue Tregs (OR=24). This confirmed that our tissue class annotations agreed with well-known subclasses of T cells sorted using established protein markers.

We also wanted to assess whether the two cell states within a chromatin class defined via cell surface proteins (e.g., CD4⁺CD25⁺PD1⁺CXCR5⁺ TFH and CD4⁺CD25⁺PD1⁺CXCR5⁻ TPH) were transcriptionally distinct. By clustering the cells from the four sorted populations based on gene expression, we successfully distinguished between the pairs of transcriptomic states from each chromatin class (**Supplementary Fig. 15d; Methods**). Moreover, we observed that each gold-standard FACS-defined population had a distinct mRNA cluster identity. Next, we calculated the differentially expressed genes and differentially accessible promoter peaks between the transcriptional states within the same class. While we found significant transcriptional differences, we largely did not observe similar accessibility differences in the corresponding genes' promoter peaks (**Fig. 7d-e; Methods**). This was consistent with the model of transcriptional cell states from a common superstate sharing open chromatin landscapes. For example, the *PDE4D* gene, which encodes an RA treatment target (McCann et al., *Arthritis Research & Therapy*, 2010), had significantly more expression in TPH than TFH cells (unadjusted P=4.64e-19), but a non-significant change in the promoter peak accessibility (unadjusted P=0.913) (**Supplementary Fig. 15e**). On the other hand, *ZBTB10*, a telomere-associated transcription factor (Bluhm et al., *NAR*, 2019), was a rare example where the chromatin accessibility and gene expression concurred across Treg states (**Supplementary Fig. 15f**). However, globally, the lack of these examples likely contributed to the lack of fully distinguished state-specific chromatin classes.

Addition to **Discussion** [in multiome/superstate paragraph]:

Indeed, when expanding genome-wide, we saw a similar pattern of class-specific transcriptional cell states but chromatin classes encompassing multiple related states in our proposed superstate model (**Fig. 7a-c; Supplementary Fig. 10g-h**). To validate this model, we conducted an RA PBMC multiome experiment of FAC-sorted populations. While we saw differentially expressed genes between transcriptional cell states within a chromatin class, there

was an almost complete lack of differentially accessible promoter peaks corresponding to those genes (**Fig. 7d-e**).

Addition to **Methods**:

Superstate multiome experimental protocol. From PBMC samples from 4 RA patients, we enriched for CD4 T cells using the MACS protocol and sorted for 4 populations using FACS (CD4⁺CD127⁻CD25^{hi} Tregs, CD4⁺CD127⁻CD25^{int} Tregs, CD4⁺CD25⁻PD1⁺CXCR5⁺ TFH, and CD4⁺CD25⁻PD1⁺CXCR5⁻ TPH). We used the following antibodies: CD3-FITC, CD4-BV421, CD25-PE-Cy7, CD127-BV650, CXCR5-PE, PD1-APC, Live/Dead-7AAD. All antibodies were purchased by BioLegend except the Live/Dead antibody purchased from ThermoFisher Scientific. After nuclei isolation, each sorted population was tagged with a nuclear hashing antibody before pooling across populations. Total-SeqTM-A hashtag antibodies were purchased from BioLegend. We performed a multiome experiment as described in **Multiome experimental protocol**, with the additional step of producing cDNA from Hashtag oligos (for Protein Antibody Hashtags) during GEM incubation, generating the Hashtag library alongside the Gene Expression library. The Hashtag library was sequenced at approximately five thousand reads per cell.

Superstate multiome quality control. Quality control steps for the superstate multiome experiment were the same as the RA tissue multiome experiments, up to and not including the doublet step in both ATAC and RNA modalities (**Supplementary Fig. 1a-b**). To better account for doublets between these very similar cell states, we only included cells with a single identity determined by running Seurat::HTODemux (Stuart et al., Cell, 2019) on the normalized hashtag library. Those cell state identities were strictly used as a label. Cells needed to pass QC in all three modalities to be included in the downstream analysis. We kept 402 CD4⁺CD127⁻CD25^{hi} Tregs, 1,690 CD4⁺CD127⁻CD25^{int} Tregs, 535 CD4⁺CD25⁻PD1⁺CXCR5⁺ TFH, and 371 CD4⁺CD25⁻PD1⁺CXCR5⁻ TPH cells.

TFH/TPH/Treg differential feature analysis. For the sorted RA PBMCs, we determined differential genes and peaks between each pair of states within one chromatin class: (1) CD4⁺CD127⁻CD25^{hi} Tregs and CD4⁺CD127⁻CD25^{int} Tregs; (2) CD4⁺CD25⁻PD1⁺CXCR5⁺ TFH and CD4⁺CD25⁻PD1⁺CXCR5⁻ TPH. We calculated differential genes as in **Single cell differential gene analysis**. Differential promoter peaks were calculated similarly to **Single cell differential peak analysis**, but we excluded sample as a covariate since there was a single pooled RA PBMC sample and used stats::glm instead of lme4::glmer since we removed the random effect of sample, thus negating the need for a mixed effect model. If a gene had multiple promoter peaks, we chose the peak with the max normalized peak accessibility summed across cells in that pair of states. Furthermore, we only included peak/gene pairs with at least 1 fragment/UMI in at least 50 cells in that pair of states. We corrected p-values using FDR separately within modalities.

Addition to **Methods** [in **RNA clustering**. section]:

We used 20 PCs for the broad cell type clustering and 10 PCs for the sorted RA PBMC mRNA clustering since there was less variation within a cell type.

R3C9. Another analysis missing from the manuscript is to compare the pooled snATAC profiles of the multiome cells in different AMP RA transcriptional states to which snRNA were assigned. As snRNA cells were assigned to the AMP RA transcriptomic clusters, pooling snATAC in the same cluster can improve the coverage and may detect the differences between the transcriptional states or confirm there is no difference.

Response: We thank the reviewer for this comment, and we have added the requested analysis. We summed the multiome snATAC reads by sample and transcriptional cell state to get a peaks x pseudobulks matrix. As quality control measures, we only included promoter peaks with at least minimal accessibility in the cell type and sample/state combinations with at least minimal cell counts (**Methods**). We ran this analysis for all states in a cell type across classes and the states within a class. For each peak for each set of states, we calculated two negative binomial models of that peak's sample/state pseudobulk distribution: a full model that accounted for state, sample, and fragment count, and a null model that removed state as the variable of interest. We conducted a log-likelihood test between the full and null models to get a p-value that we then adjusted for multiple hypotheses via FDR within each analysis run.

We expected to find differential peaks between transcriptional cell states within different chromatin classes since their chromatin was sufficiently different to be labeled as separate classes. Indeed, when we ran this analysis for all states within each cell type, we found many differential peaks at FDR<0.10, with a median of 8717 peaks across cell types (**Supplementary Fig. 12a**).

Additional Supplementary Fig. 12:

a

Supplementary Fig. 12. Differential promoter peaks between pseudobulk by sample and transcriptional cell states within or across chromatin classes.

a. The number of differential promoter peaks between different transcriptional cell states within or across classes, as determined by an ANOVA LRT FDR<0.10 (**Methods**), colored by class or

cell type; peaks with $FDR \geq 0.10$ are colored in gray. A dashed line separates the within and across class analyses.

However, if our proposed superstate hypothesis is correct, we would expect many fewer differential peaks between states within a chromatin class superstate. We found very few differential peaks at $FDR < 0.10$ for states within a class, with a median of 23 peaks across classes (**Supplementary Fig. 12a**). For example, T_{A-1} found 218 differentially accessible peaks across the sample/state pseudobulks (1.3% of peaks tested). That included the promoter peaks for *CD4* and *CD8A*, as expected given that T_{A-1} encompasses both T-4: CD4+ naive and T-16: CD8+ CD45ROlow/naive states based on their shared naive functions, as further evidenced by their shared *SELL* promoter peaks (**Supplementary Fig. 12b**). This suggests that there are very few differences between transcriptional cell states in the same chromatin class at the peak level. Furthermore, this corroborates the similarity of pooled ATAC reads by transcriptional cell state for states in the same chromatin class we found at the loci highlighted in **Supplementary Figs. 11 and 18**.

Addition to **Supplementary Fig. 12**:

b. For T_A-1: CD4+ IL7R+ chromatin class, the Z-score of the transcriptional cell state beta for each differential peak, labeled by the gene associated with that promoter (**Methods**). T-2: CD4+ IL7R+ CCR5+ memory was used as the reference cell type in the model. We highlighted *CD4*, *CD8A*, and *SELL* promoter peaks with bolded gene names and a black box.

Addition to **Results**:

Indeed, when we aggregated the snATAC reads by states, we observed shared openness between transcriptional cell states within the same class (*i.e.*, superstate), as seen with the cytotoxic T_A-4 grouped cell states T-12/T-15 at the cytotoxicity-associated³² *FGFBP2* gene, lining fibroblast S_A-1 grouped cell states F-0/F-1 at the lining-associated¹¹ *CLIC5* gene, and intermediary myeloid M_A-4 grouped cell states M-3/M-4 at bone marrow macrophage-associated⁶⁰ *SPP1* gene (**Supplementary Fig. 11**). Furthermore, we found very few differential peaks between transcriptional states in the same chromatin class even after pseudobulking by sample and state to decrease sparsity (**Methods**; **Supplementary Fig. 12a**). T_A-1: CD4+ IL7R+ had one of the higher numbers of differential peaks within a class, but still only found 1.3% of the peaks tested as differential. Among those was the expected *CD4* and *CD8A* promoter peaks since both the T-16: CD8+ CD45ROlow/naive state and T-4: CD4+ naive state corresponded to T_A-1 (**Supplementary Fig. 12b**; **Fig. 7a**). These populations likely mapped together since they shared naïve T cell transcriptional profiles, consistent with a highly accessible *SELL* promoter peak. This contrasted sharply to the number of differential peaks found between states across classes within a cell type (median of 8,717 within a cell type vs 23 within a single class; **Supplementary Fig. 12a**), suggesting that the chromatin landscape in states within a class is much more homogeneous than across classes, as proposed by our superstate model.

Addition to **Methods**:

ATAC pseudobulk differential peak analysis. For T, stromal, and myeloid cell types, we summed the non-binary ATAC peaks x cells matrix by sample and transcriptional cell state combinations. We subset the summed matrix to include only samples with more than 150 cells, states with more than 130 cells, and combinations with more than 10 cells. For the within class analysis, we split the matrix by the transcriptional cell states that belonged to the same chromatin class (e.g., 5 T cell matrices); we excluded any class with only 1 state passing our QC thresholds. We also kept the full matrix per cell type for the across classes analysis. We subset peaks by each cell type's promoter PMA peaks (see **T cell lineage analysis**) that had at least 5 reads across the pseudobulks within that analysis. For each peak for each set of states (either within or across classes), we calculated two negative binomial models of that peak's sample/state pseudobulk distribution using MASS::glm.nb, accounting for covariates of sample identity and the number of fragments in the sample and cell state combination and differing by the inclusion of transcriptional cell state:

Full model: peak ~ cell state + sample + scale(log10(nFragments))

Null model: peak ~ sample + scale(log10(nFragments))

Cell state and sample were represented by a 1-hot encoded matrix. We calculated an ANOVA log-likelihood ratio test (LRT) p-value between these two models and reconciled multiple hypothesis test correction within each analysis separately via FDR. Peaks were considered differential if they had FDR<0.10.

R3C10. In Discussion Line 605-608, any evidence to support that non-pathogenic transcriptional cell states are able to transition to pathogenic transcriptional cell state if they correspond to the same chromatin class?

Response: We thank the reviewer for this fascinating question. It would be very useful within the field to know if non-pathogenic transcriptional cell states can transition to pathogenic states within a similar chromatin context both to understand the biological mechanisms at play and to guide treatment strategies (e.g., cell state depletion or signaling disruption). However, the experimental validation of such a question is beyond the scope of our study, though we do suggest them as part of our discussion. We also give an example of one such study that showed ILCs in a mouse model of psoriasis with chromatin accessibility in a disease-relevant population of ILC3s even before disease induction, particularly at ILC3 TFs; after addition of disease-inducing IL-23, the chromatin accessibility in ILC3s further increased (Bielecki et al., Nature, 2021).

Addition to Discussion:

Defining the relationship between transcriptional cell states and chromatin classes may have important therapeutic implications. One effective RA treatment strategy is the deletion of a pathogenic cell state: the use of B-cell depleting antibodies (e.g., rituximab¹⁰) is an example. However, if one chromatin class corresponds to multiple transcriptional cell states, then deleting very specific pathogenic populations may be ineffective as other non-pathogenic states may transition into the pathogenic state in response to the same pathogenic tissue environment. As an example, a recent study (Bielecki et al., Nature, 2021) of ILCs in a mouse model of psoriasis showed chromatin accessibility in a disease-relevant population of ILC3s even before disease induction using IL-23, particularly at ILC3 TFs, that then increased further after induction. In that case, altering the environment or removing exogenous factors (e.g., TFs, cytokines) might be a more effective treatment. Within RA, the S_A-0: CXCL12+ HLA-DR^{hi} sublining fibroblast class, with its four related transcriptional states in our superstate model, may merit further investigation in this regard. S_A-0 accessible peaks were enriched for STAT motifs, suggesting potential regulation by the JAK/STAT signaling pathway. Indeed, JAK inhibition via tofacitinib and upadacitinib has been shown to prevent pro-inflammatory HLA-DR induction in RA synovial fibroblasts¹²⁹. Additional experiments would be required to determine if the F-3: POSTN+ sublining transcriptional cell state could transform into the RA-expanded (Zhang et al., Nature, 2023) F-5: CD74^{hi}HLA^{hi} sublining or F-6: CXCL12+ SFRP1+ sublining fibroblast populations under JAK/STAT stimulation.

REVIEWERS' COMMENTS

Reviewer #1 (Remarks to the Author):

The authors have addressed all my concerns. Congratulations to a very nice paper.

Reviewer #2 (Remarks to the Author):

The authors have done an outstanding job responding to all of the reviewers' comments. The revisions included in the revised manuscript, have made an excellent paper even better.

Reviewer #3 (Remarks to the Author):

The authors have conducted additional experimental and computational analyses to thoroughly address all the comments and concerns. In particular, the additional analysis on the sorted superstate cells and additional details of data QC and analysis details make the study much more solid.

Reviewer #3 (Remarks on code availability):

The code doe provide README files that provide sufficient information to run the scripts.